# A hitchhiker's guide to Poisson gradient estimation

**Michael Ibrahim** [1] [*]  **Hanqi Zhao** [2] [*]  **Eli Sennesh** [3]  **Zhi Li** [4]  **Anqi Wu** [2]  **Jacob L. Yates** [1]  **Chengrui Li** [5] [†]
**Hadi Vafaii** [1] [†]

## Abstract

Poisson-distributed latent variable models are widely used in computational neuroscience, but differentiating through discrete stochastic samples remains challenging. Two approaches address this: *Exponential Arrival Time* simulation (EAT; Vafaii et al., 2024) and *Gumbel-SoftMax* relaxation (GSM; Li et al., 2024a). We provide the first systematic comparison of these methods, along with practical guidance for practitioners. Our main technical contribution is a modification to the EAT method that theoretically guarantees an unbiased first moment (exactly matching the firing rate), and reduces second-moment bias. We evaluate these methods on their distributional fidelity, gradient quality, and performance on two tasks: (1) variational autoencoders with Poisson latents, and (2) partially observable generalized linear models, where latent neural connectivity must be inferred from observed spike trains. Across all metrics, our modified EAT method exhibits better overall performance (often comparable to exact gradients), and substantially higher robustness to hyperparameter choices. These results extend to over-dispersed Negative Binomial latents, where modified EAT again performs best. However, only GSM generalizes to arbitrary non-Poisson distributions, including the under-dispersed regime. Together, our results clarify the trade-offs between these methods and offer concrete recommendations for practitioners working with Poisson latent variable models. Our code, data, and model checkpoints are available here: github.com/hadivafaii/PoissonGradientEstimation

[*]Equal contribution † Co-senior authors [1]Redwood Center for Theoretical Neuroscience, UC Berkeley [2]Georgia Institute of Technology [3]VERSES AI Research Lab, Los Angeles, USA [4]Aerospace Information Research Institute, Chinese Academy of Sciences [5]Reality Labs, Meta. Correspondence to: Chengrui Li <cnlichengrui@meta.com>, Hadi Vafaii <vafaii@berkeley.edu>.

*Proceedings of the 43rd International Conference on Machine Learning*, Seoul, South Korea. PMLR 306, 2026. Copyright 2026 by the author(s).

## 1. Introduction

Brains communicate via discrete *action potentials*, or "spikes" (Kandel et al., 2021; du Bois-Reymond, 1848), that are often noisy (Faisal et al., 2008). The *rate coding hypothesis* (Adrian & Zotterman, 1926; Barlow, 1972; Perkel & Bullock, 1968) states that information is encoded in *firing rates* (number of spikes per unit time), motivating the use of Poisson distributions to model stochastic spike counts.

The Poisson distribution is a principled choice, for both theoretical and empirical reasons. Theoretically, Poisson arises as the maximum-entropy limit of sums of independent Bernoulli events with fixed mean (Harremoës, 2002), interpretable as spikes arising from many weak synaptic inputs. Empirically, this is supported by classic evidence that cortical spike counts often exhibit approximately Poisson variability (Tomko & Crapper, 1974; Tolhurst et al., 1983; Shadlen & Newsome, 1998), together with recent mechanistic evidence that Poisson-like irregularity can emerge from weakly synchronous network drive (Pattadkal et al., 2025).

Motivated by this connection, Vafaii et al. (2024) developed the *Poisson Variational Autoencoder* ($\mathcal{P}$-VAE), a generative model with Poisson-distributed latent variables that encodes inputs as discrete spike counts. Subsequent work demonstrated that iterative inference in the $\mathcal{P}$-VAE yields a spiking implementation of *sparse coding* (Vafaii et al., 2025; Olshausen & Field, 1996; Rozell et al., 2008). Further work investigated the information geometry of the Poisson family, showing that the $\mathcal{P}$-VAE objective naturally links information *coding rate* to neural *firing rate*, yielding an emergent metabolic cost that promotes sparse, energy-efficient representations under resource constraints (Vafaii & Yates, 2026). Separately, Johnson et al. (2025) showed that the $\mathcal{P}$-VAE captures key psychophysical phenomena in perceptual decision-making, such as speed–accuracy trade-offs.

From a machine learning perspective, the $\mathcal{P}$-VAE model family demonstrates improved sample efficiency in downstream classification tasks (Vafaii et al., 2024), as well as strong out-of-distribution generalization performance (Vafaii et al., 2025), surpassing hybrid iterative-amortized VAE alternatives (Kim et al., 2018; Marino et al., 2018). Take together, these results position the $\mathcal{P}$-VAE family as a strong candidate for NeuroAI applications.

*Table 1. A hitchhiker's guide to Poisson gradient estimation.* We summarize the relative strengths and limitations of each relaxation method across five key axes based on our theoretical and empirical findings. **Distributional Fidelity:** adherence to true Poisson moments (Fig. 1) and Wasserstein distance (Fig. 2). **Gradient Bias/Variance:** quality of the gradient estimator (Fig. 3). **Temperature Robustness:** stability of performance across temperature $\tau$ selection (Fig. 4). **Generalizability:** ability to model any non-Poisson discrete distribution.

| Method | Dist. Fidelity | Gradient Bias | Gradient Variance | Temp. Robustness | General-izability | Recommendation |
|---|---|---|---|---|---|---|
| **EAT$_{cubic}$** (Ours) | ★★★★★ *Unbiased* | ★★★★★ *Low* | ★★★★☆ *Moderate* | ★★★★★ *Excellent* | ★☆☆☆☆ *Poisson only* | **Default Choice** |
| **EAT$_{sigmoid}$** (Vafaii et al., 2024) | ★☆☆☆☆ *High bias* | ★★★★☆ *High at high $\tau$* | ★★★★★ *Lowest* | ★★☆☆☆ *Sensitive* | ★☆☆☆☆ *Poisson only* | **Max Smoothness** |
| **GSM** (Li et al., 2024a) | ★★★☆☆ *Moderate* | ★★☆☆☆ *High at low $\tau$* | ★★☆☆☆ *Poor* | ★★☆☆☆ *Sensitive* | ★★★★★ *Universal* | **Non-Poisson Data** |

However, gradient-based training of discrete latent variable models is difficult. Unlike Gaussian VAEs (Kingma & Welling, 2013), discrete models generally lack low-variance pathwise estimators, forcing practitioners to rely on high-variance score function estimators (Paisley et al., 2012; Bengio et al., 2013; Schulman et al., 2015; Li et al., 2024b). To overcome this challenge, Vafaii et al. (2024) introduced a differentiable relaxation of Poisson based on point processes, which we call the *Exponential Arrival Time* (EAT) method. This enabled end-to-end training of $\mathcal{P}$-VAE models using standard autograd libraries.

Independently, Li et al. (2024a) developed an alternative Poisson relaxation based on the *Gumbel-SoftMax* (GSM) approach (Jang et al., 2017; Maddison et al., 2017). They applied this to the *Partially Observable Generalized Linear Model* (POGLM) for neural population data (Pillow & Latham, 2007). POGLMs are important for neural data analysis as they enable inference of network connectivity and latent dynamics from partially observed recordings (a common scenario in neuroscience where not all neurons can be simultaneously measured). While both EAT and GSM address the same fundamental challenge—differentiating through stochastic, integer-valued Poisson samples—no systematic comparison between them exists. Consequently, practitioners lack guidance on method selection.

**Our contributions.** In this paper, we provide this missing comparison. Crucially, we identify a limitation in the standard EAT method: it exhibits high sensitivity to the temperature hyperparameter. We trace this instability to the use of a sigmoid approximation, whose infinite support leads to poor distributional fidelity. Our key technical contribution is a principled correction: we replace the sigmoid in EAT with a cubic Hermite interpolant (smoothstep) possessing compact support. We compare our modified algorithm, EAT$_{cubic}$, against the original EAT$_{sigmoid}$, GSM, score-based, and exact gradients (when available), across both $\mathcal{P}$-VAE and POGLM settings, finding that EAT$_{cubic}$ consistently outperforms all alternatives across distributional fidelity, gradient quality,

and downstream task performance, while remaining significantly more robust to temperature choices. We synthesize our findings into practical recommendations in Tab. 1.

**Paper outline.** The paper is organized as follows:

- **Section 2** (with **sections A.1, B and C**): Pedagogical introduction to variational inference, gradient estimation, and the EAT and GSM relaxation methods.

- **Sections 3.1 and 3.2** (with **section D**): We introduce EAT$_{cubic}$ and demonstrate its substantially improved distributional fidelity via Campbell's theorem.

- **Section 3.3** (with **sections E and F**): Systematic evaluation of gradient quality against exact gradients.

- **Section 3.4** (with **sections G and H**): Empirical validation. While all pathwise methods succeed with careful tuning, only EAT$_{cubic}$ is stable across temperatures.

- **Section 4**: Conclusions and directions for future work.

## 2. Background and related work

We use red / blue to indicate the encoder (inference) / decoder (generative) components of each model, respectively.

**Probabilistic latent variable models.** Consider a generative model $p(\boldsymbol{x}, \boldsymbol{z}; \boldsymbol{\theta})$, where $\boldsymbol{x}$ are observed data, $\boldsymbol{z}$ are unobserved latent variables, and $\boldsymbol{\theta}$ are the parameters of the generative model. Since $\boldsymbol{z}$ is unknown, learning $\boldsymbol{\theta}$ via maximum likelihood $\arg\max_{\boldsymbol{\theta}} \int p(\boldsymbol{x}, \boldsymbol{z}; \boldsymbol{\theta}) \, d\boldsymbol{z}$ is challenging due to the often intractable marginalization over $\boldsymbol{z}$. This presents two coupled tasks: (i) learning the model parameters $\boldsymbol{\theta}$, and (ii) inferring the latents $\boldsymbol{z}$ given observations $\boldsymbol{x}$ (i.e., finding the optimal posterior $p(\boldsymbol{z}|\boldsymbol{x}; \boldsymbol{\theta})$).

**Variational inference and ELBO.** Variational inference (VI; Blei et al. (2017)) is a powerful framework for learning latent variable models, including VAEs (Kingma & Welling,

$$\mathrm{ELBO}(\boldsymbol{x}; \boldsymbol{\theta}, \boldsymbol{\phi}) \; := \; \ln p(\boldsymbol{x}; \boldsymbol{\theta}) - \mathcal{D}_{\mathrm{KL}}\Big(q(\boldsymbol{z}|\boldsymbol{x}; \boldsymbol{\phi}) \,\|\, p(\boldsymbol{z}|\boldsymbol{x}; \boldsymbol{\theta})\Big) \; = \; \mathbb{E}_{q(\boldsymbol{z}|\boldsymbol{x};\boldsymbol{\phi})}\Big[\ln p(\boldsymbol{x}, \boldsymbol{z}; \boldsymbol{\theta}) - \ln q(\boldsymbol{z}|\boldsymbol{x}; \boldsymbol{\phi})\Big] \qquad (1)$$

2013; Vafaii et al., 2024) and POGLMs (Li et al., 2024a). In VI, we introduce a *variational posterior* $q(\boldsymbol{z}|\boldsymbol{x}; \boldsymbol{\phi})$ with parameters $\boldsymbol{\phi}$, and address both inference and learning tasks by optimizing the *Evidence Lower BOund* (ELBO; Eq. (1)). Since the KL divergence is always non-negative, maximizing ELBO with respect to the generative model parameters $\boldsymbol{\theta}$ improves the marginal likelihood $p(\boldsymbol{x}; \boldsymbol{\theta})$; while maximizing ELBO with respect to the inference model parameters $\boldsymbol{\phi}$ minimizes the KL between the approximate posterior $q(\boldsymbol{z}|\boldsymbol{x}; \boldsymbol{\phi})$ and the true posterior $p(\boldsymbol{z}|\boldsymbol{x}; \boldsymbol{\theta})$, which improves the quality of inference.

**Optimizing ELBO: score-based vs. pathwise.** When applying stochastic gradient ascent to Eq. (1), the gradient w.r.t. $\boldsymbol{\phi}$ is non-trivial because the expectation over $q$ depends on $\boldsymbol{\phi}$. The *score function* gradient estimator (REINFORCE; Williams (1992)) handles this by using the log-derivative identity $\nabla_{\boldsymbol{\phi}} q = q \, \nabla_{\boldsymbol{\phi}} \ln q$ to bring the derivative inside the expectation, which is then approximated via Monte Carlo sampling. We provide details in section A.1, with the final score-based gradient formula in Eq. (14).

An alternative is the *pathwise* gradient estimator, which requires a *reparameterization trick*. The canonical example is the Gaussian VAE (Kingma & Welling, 2013), where a latent sample is written as $\boldsymbol{z} = \boldsymbol{\mu} + \boldsymbol{\sigma} \odot \boldsymbol{\epsilon}$ with $\boldsymbol{\epsilon} \sim \mathcal{N}(\boldsymbol{0}, \mathbf{I})$. This moves the dependence on $\boldsymbol{\phi}$ inside the expectation, enabling direct backpropagation. We discuss this further in section A.2, with the final formula in Eq. (15).

The score function estimator suffers from high variance compared to pathwise estimators (Paisley et al., 2012; Bengio et al., 2013; Schulman et al., 2015). For Poisson-distributed latent variables, no reparameterization trick analogous to the Gaussian was previously known. To address this, Vafaii et al. (2024) and Li et al. (2024a) independently introduced methods for relaxing the Poisson distribution into differentiable approximations, enabling pathwise gradient estimation. We now briefly review these methods.

**The Exponential Arrival Time (EAT) relaxation.** Motivated by the spiking nature of neural computation, Vafaii et al. (2024) introduced the Poisson VAE ($\mathcal{P}$-VAE) by replacing Gaussian latents with Poisson-distributed spike counts. In a $\mathcal{P}$-VAE, both the prior and approximate posterior are Poisson: $q(\boldsymbol{z}|\boldsymbol{x}; \boldsymbol{\phi}) = \mathcal{P}\mathrm{ois}(\boldsymbol{z}; \boldsymbol{\lambda}(\boldsymbol{x}))$, where $\boldsymbol{\lambda}$ is a vector of firing rates. The encoder network outputs log-rates $\boldsymbol{u}(\boldsymbol{x}) = \mathrm{enc}(\boldsymbol{x}; \boldsymbol{\phi})$, which are exponentiated to ensure positivity: $\boldsymbol{\lambda}(\boldsymbol{x}) = \exp(\boldsymbol{u}(\boldsymbol{x}))$. As in standard VAEs (Kingma & Welling, 2013), the ELBO (Eq. (1)) further decomposes

into a reconstruction term minus a KL regularizer:

$$\begin{aligned}\mathrm{ELBO}(\boldsymbol{x}; \boldsymbol{\theta}, \boldsymbol{\phi}) \; = \; & \mathbb{E}_{\boldsymbol{z} \sim q(\boldsymbol{z}|\boldsymbol{x};\boldsymbol{\phi})}[\ln p(\boldsymbol{x}|\boldsymbol{z}; \boldsymbol{\theta})] \\ & - \mathcal{D}_{\mathrm{KL}}(q(\boldsymbol{z}|\boldsymbol{x}; \boldsymbol{\phi}) \,\|\, p(\boldsymbol{z}; \boldsymbol{\theta})) .\end{aligned} \qquad (2)$$

The main challenge in training $\mathcal{P}$-VAE is differentiating through stochastic, integer spike count samples $\boldsymbol{z}$. To address this, the authors developed the *Exponential Arrival Time* (EAT) relaxation. The key insight is that a Poisson random variable $z \sim \mathcal{P}\mathrm{ois}(\lambda)$ can be viewed as counting events in a Poisson point process: $z = \sum_m \mathbf{1}[t_m < 1]$, where $t_m = \sum_{i=1}^{m} \Delta t_i$ are arrival times generated by cumulating exponential inter-arrival times. The hard indicator $\mathbf{1}[t_m < 1]$ is non-differentiable, but can be relaxed using a smooth approximation $f_{\mathrm{approx}}((1 - t_m)/\tau)$ (e.g., sigmoid):

$$z = \sum_{m=1}^{M} f_{\mathrm{approx}}\left(\frac{1 - \sum_{i=1}^{m} \Delta t_i}{\tau}\right), \quad \Delta t_i \overset{\mathrm{i.i.d.}}{\sim} \mathrm{Exp}(\lambda),$$

where the number of exponential samples $M$ should be large enough and $\tau > 0$ is a temperature parameter controlling the sharpness of the threshold. As $\tau \to 0$, the relaxation recovers the true Poisson distribution. We review the EAT method in detail in section B, summarize it in Algorithm 1, and provide a visual illustration in Fig. 6.

The choice of $f_{\mathrm{approx}}$ and $\tau$ are critical modeling decisions. Vafaii et al. (2024) used the sigmoid function without a clear justification, and did not systematically explore the temperature parameter. Their results reveal high sensitivity to $\tau$, with only a narrow range yielding good performance ($\tau \lesssim 0.05$). This sensitivity suggests that at larger $\tau$, the relaxed distribution deviates substantially from the true Poisson, degrading optimization. Understanding and mitigating this distributional mismatch is a central goal of our paper.

**The Gumbel-Softmax (GSM) relaxation.** A critical problem in neural recording is that the recorded neurons are only a small part of the entire population in a particular brain region of interest. A partially observable generalized linear model (POGLM) (Pillow & Latham, 2007; Jimenez Rezende & Gerstner, 2014; Linderman et al., 2017) studies the mutual interactions between all neurons underlying the observed neural spike trains $\boldsymbol{x}_1, \ldots, \boldsymbol{x}_T$, while assuming the existence of latent spike trains $\boldsymbol{z}_1, \ldots, \boldsymbol{z}_T$ from hidden neurons. Unlike VAEs, POGLM is inherently a sequential model with a time dimension $t \in \{1, \ldots, T\}$ and its generative model cannot be decomposed into a prior-

conditional-product explicitly. Specifically,

$$\begin{bmatrix} \boldsymbol{x}_t \\ \boldsymbol{z}_t \end{bmatrix} \sim \mathcal{P}\text{ois}\left( f\left( \begin{bmatrix} \boldsymbol{x}_{t-1} \\ \boldsymbol{z}_{t-1} \end{bmatrix}, \begin{bmatrix} \boldsymbol{x}_{t-2} \\ \boldsymbol{z}_{t-2} \end{bmatrix}, \dots; \boldsymbol{\theta} \right) \right),$$

where $f(\cdot; \boldsymbol{\theta})$ can be arbitrary but is usually a linear mapping in neuroscience modeling settings. To learn such a complicated model through VI efficiently, a stable gradient estimation for $\boldsymbol{\phi}$ is necessary on the variational distribution $\boldsymbol{z}_t \sim \mathcal{P}\text{ois}\left( g\left( \boldsymbol{x}_1, \dots, \boldsymbol{x}_T; \boldsymbol{\phi} \right) \right)$, where $g(\cdot; \boldsymbol{\phi})$ can be viewed as an encoder.

The *Gumbel-SoftMax* (GSM, Li et al. (2024a)) Poisson relaxation method uses the Gumbel-Softmax/Concrete distribution (Jang et al., 2017; Maddison et al., 2017) that relaxes each $z \sim \mathcal{P}\text{ois}(\lambda)$ as:

$$z = \sum_{m=0}^{M-1} m \cdot \tilde{z}_m, \quad \tilde{z}_0, \dots, \tilde{z}_{M-1} \sim \text{GS}_\tau(\pi_0, \dots, \pi_{M-1}),$$

where $\text{GS}_\tau$ is the Gumbel-Softmax distribution with temperature $\tau$; $\pi_m = \mathcal{P}\text{ois}(m; \lambda)$ is the Poisson PMF at $m$.

**The difficulty of auto-tuning temperature.** Both EAT and GSM relaxations require setting a critical hyperparameter: the temperature $\tau > 0$, which controls the smoothness of the continuous relaxation. However, neither Vafaii et al. (2024) nor Li et al. (2024a) addressed how $\tau$ should be chosen. We investigated whether a principled approach could determine $\tau$ automatically, but found no simple heuristic (section H.1). Analyzing gradient estimation as a regression task revealed that the optimal $\tau$ depends non-linearly on both the firing rate $\lambda$ and the specific objective function (Fig. 9). This lack of a universal scaling law forces practitioners to perform costly grid searches, with suboptimal choices leading to poor convergence or biased gradients. This motivates our central contribution: *a modified* EAT *relaxation that is substantially more robust to temperature selection, reducing the need for careful tuning.*

## 3. Results

Our contributions are threefold. First, we improve both the EAT and GSM methods as our core technical contribution (section 3.1). Second, we systematically compare these methods across multiple axes—*distributional fidelity*, *gradient quality*, and *empirical performance* (sections 3.2 to 3.4))—revealing their respective strengths and weaknesses. Finally, we synthesize these insights into concrete guidance for practitioners, supporting Tab. 1.

### 3.1. Improving the distributional fidelity of EAT via cubic Hermite interpolation

The EAT method provides a continuous relaxation of the Poisson distribution (section B). For non-zero temperatures

($\tau > 0$), this relaxed distribution deviates from the true Poisson. But how severe is this deviation? In this section, we answer this question theoretically by deriving closed-form expressions for the first and second moments. Our analysis reveals that EAT$_\text{sigmoid}$ exhibits substantial bias, motivating our proposed modification: EAT$_\text{cubic}$, which replaces the sigmoid with a cubic Hermite polynomial.

**Point processes and Campbell's theorem.** The EAT method exploits the fact that a Poisson sample can be viewed as counting events in a homogeneous point process with rate $\lambda$. Campbell's theorem (Campbell, 1909; Kingman, 1992; Daley & Vere-Jones, 2003) allows us to analytically compute the moments of the relaxed distribution as a function of temperature. We provide an intuitive derivation and formal treatment in section D.

**Closed-form expressions for the EAT moments.** Given a soft indicator $f_\text{approx}(\cdot)$, temperature $\tau$, and a sample $z$ drawn via Algorithm 1 with firing rate $\lambda$, we define the *mean factor* $c(\tau)$ and *variance factor* $v(\tau)$ as the ratios of the relaxed distribution's mean and variance to those of the true Poisson (both equal to $\lambda$). Applying Campbell's theorem (section D.2), we obtain:

$$\mathbb{E}[z] = \lambda\, c(\tau), \quad c(\tau) := \tau \int_{-\infty}^{1/\tau} f_\text{approx}(u)\, du, \quad (3)$$

$$\text{Var}(z) = \lambda\, v(\tau), \quad v(\tau) := \tau \int_{-\infty}^{1/\tau} f_\text{approx}(u)^2\, du. \quad (4)$$

A faithful relaxation requires $c(\tau) = 1$ (unbiased mean) and $v(\tau) = 1$ (correct variance).

**EAT$_\text{sigmoid}$ exhibits substantial bias.** Evaluating these integrals for the sigmoid $f_\text{approx}(u) = 1/(1 + e^{-u})$ (Section D.3) yields:

$$\begin{aligned} c_\text{sigmoid}(\tau) &= \tau \ln(1 + e^{1/\tau}), \\ v_\text{sigmoid}(\tau) &= \tau \left[ \ln(1 + e^{1/\tau}) - \frac{1}{1 + e^{-1/\tau}} \right]. \end{aligned} \quad (5)$$

As $\tau \to 0$, both factors approach 1, recovering the true Poisson. However, the deviations become substantial as the temperature increases. For example, at $\tau = 1.0$, we have $c_\text{sigmoid} \approx 1.31$ (31% mean inflation) and $v_\text{sigmoid} \approx 0.58$ (42% variance reduction).

**The EAT$_\text{cubic}$ approximation is unbiased.** Why is the sigmoid so biased? The root cause is its infinite support: the sigmoid tails extend to $\pm\infty$, allowing events far outside the unit interval to contribute fractional amounts: "ghost spikes" that inflate the mean while smoothing out variance.

This suggests using a soft indicator with *compact support*. The simplest option, like a hard ramp (ReLU-style), is compact but not differentiable at the boundaries, which can

cause gradient issues. We instead use the *cubic smoothstep* (Hermite interpolation), which is $C^1$ continuous (Fig. 7):

$$f_{\text{cubic}}(u) = 3w^2 - 2w^3, \quad w = \text{clamp}\left(\frac{u+1}{2}, 0, 1\right)$$

This function transitions smoothly from 0 to 1 over the interval $[-1, 1]$, with zero derivatives at both boundaries.

Applying our moment formulas (Eqs. (3) and (4)), we solve the integrals (Section D.4), and find that for $\tau \leqslant 1$ [1]:

$$
\begin{aligned}
c_{\text{cubic}}(\tau) &= 1, \\
v_{\text{cubic}}(\tau) &= 1 - \frac{9\tau}{35}.
\end{aligned}
\tag{6}
$$

Remarkably, the cubic achieves $c(\tau) = 1$ *exactly* for all $\tau \leqslant 1$: the mean is unbiased by construction. The variance factor also remains relatively closer to 1. For example, at $\tau = 1.0$, we have $v_{\text{cubic}} \approx 0.74$ compared to $v_{\text{sigmoid}} \approx 0.58$.

**EAT_cubic has a more biologically-plausible Fano factor.** The Fano factor $F = \text{Var}(z)/\mathbb{E}[z] = v(\tau)/c(\tau)$ characterizes the dispersion of a distribution. For Poisson, $F = 1$ exactly. This quantity is of particular interest in neuroscience, where cortical neurons typically exhibit Fano factors in the range 0.5–1.5 (Churchland et al., 2010; Pattadkal et al., 2025). While both EAT_sigmoid and EAT_cubic relaxations are sub-Poisson ($F < 1$), the cubic remains closer to the physiological range across all temperatures. For example, at $\tau = 1.0$: $F_{\text{cubic}} \approx 0.74$ compared to $F_{\text{sigmoid}} \approx 0.44$.

Figure 8 visualizes the analytical expressions for sigmoid (Eq. (5)) and cubic (Eq. (6)). See Tab. 2 for a summary.

**Empirical validation: EAT_cubic substantially improves the moment biases.** Our theoretical analysis provides closed-form expressions for the moment biases, but do these predictions hold in practice? We validate them empirically and extend the comparison to include GSM.

We constructed EAT_sigmoid, EAT_cubic, and GSM relaxations across temperatures $\tau \in [0.02, 0.5]$ and rates $\lambda \in \{2, 20, 100\}$. For each configuration, we drew $N_{\text{samples}} = 50{,}000$ samples from these relaxed distributions and computed the empirical mean and variance. We repeated this for $N_{\text{trials}} = 100$ to obtain error bars, then divided by the true Poisson moments (both equal to $\lambda$).

Figure 1 shows the results. For the mean (top row), EAT_cubic remains essentially unbiased across all conditions, confirming our theoretical prediction that $c_{\text{cubic}}(\tau) = 1$ for $\tau \leqslant 1$. The GSM method also maintains low mean bias. In contrast, EAT_sigmoid exhibits substantial bias that worsens with both

---

[1]For $\tau > 1$, see Eq. (52).

temperature and rate: at $\lambda = 100$ and $\tau = 0.5$, the mean drops to $\sim 85\%$ of the true value.

For the variance (bottom row), the differences are even more striking. EAT_cubic consistently outperforms both alternatives, retaining approximately $73\%$ of the true variance at $\lambda = 100$, $\tau = 0.5$. The GSM method retains roughly $60\%$, while EAT_sigmoid collapses to below $20\%$.

Overall, EAT_sigmoid performs poorly on both moments, with degradation accelerating at higher rates and temperatures. While GSM matches EAT_cubic on mean bias, EAT_cubic achieves substantially better variance fidelity, making it the preferred choice for distributional accuracy.

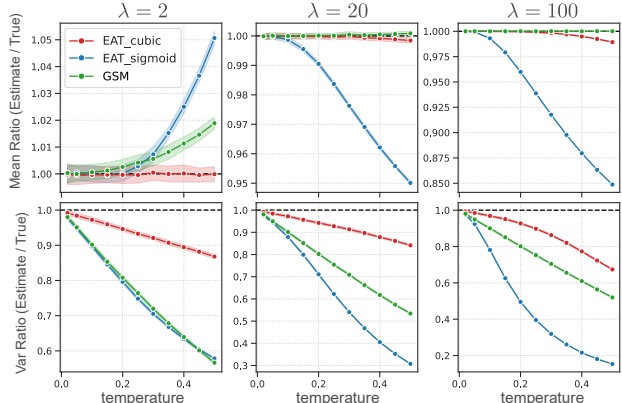

*Figure 1.* **Empirical validation of moment biases across relaxation methods.** We compare EAT_sigmoid (blue), EAT_cubic (red), and GSM (green) across temperatures $\tau \in [0.02, 0.5]$ and firing rates $\lambda \in \{2, 20, 100\}$ (columns). **Top row:** Ratio of empirical mean to true Poisson mean ($\lambda$). **Bottom row:** Ratio of empirical variance to true Poisson variance ($\lambda$). Dashed lines indicate ideal Poisson fidelity (ratio = 1). Shaded regions show $\pm 1$ standard error over 100 trials. EAT_cubic (red) maintains near-perfect mean fidelity across all conditions and substantially better variance fidelity than both alternatives, confirming our theoretical predictions. EAT_sigmoid (blue) exhibits severe bias, particularly at higher rates, with variance collapsing to below 20% of the true value at $\lambda = 100$, $\tau = 0.5$. See Fig. 14 for results across all tested $\lambda$.

### 3.2. EAT_cubic achieves superior distributional fidelity

The moment analysis captures mean and variance, but distributions can differ in higher-order statistics while sharing the same first two moments. For a more complete picture, we compare the full distributions using the *Wasserstein-1* distance (also known as the *earth mover's distance*):

$$W_1(P, Q) := \int_{-\infty}^{\infty} |F_P(x) - F_Q(x)| \, dx,$$

where $F_P$ and $F_Q$ are the cumulative distribution functions of $P$ and $Q$, respectively. Intuitively, $W_1$ measures the minimum "work" required to transform one distribution into another, where work is the amount of probability mass moved times the distance traveled. For one-dimensional

distributions, $W_1$ can be efficiently computed from empirical samples by sorting and summing absolute differences between order statistics.

We computed the $W_1$ distance between each relaxation and the true Poisson using the same experimental settings as before ($\tau \in [0.02, 0.5]$, $\lambda \in \{2, 20, 100\}$). Figure 2 shows the results. Across all conditions, $\text{EAT}_{\text{cubic}}$ achieves substantially lower $W_1$ distance than both alternatives. The advantage becomes more pronounced at higher rates. At $\lambda = 100$ and $\tau = 0.5$, $\text{EAT}_{\text{cubic}}$ achieves a $7\times$ improvement over the original method: $W_1 \approx 0.2$, compared to $\approx 1.5$ for $\text{EAT}_{\text{sigmoid}}$. These results confirm that $\text{EAT}_{\text{cubic}}$ provides the most faithful approximation to the true Poisson distribution across the full range of temperatures and rates.

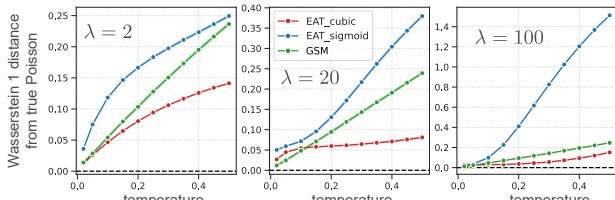

*Figure 2.* **Wasserstein-1 distance from true Poisson across relaxation methods.** We compare $\text{EAT}_{\text{sigmoid}}$ (blue), $\text{EAT}_{\text{cubic}}$ (red), and GSM (green) across temperatures $\tau \in [0.02, 0.5]$ and firing rates $\lambda \in \{2, 20, 100\}$ (columns). Lower values indicate better distributional fidelity; the dashed line at zero represents a perfect match. $\text{EAT}_{\text{cubic}}$ consistently achieves the lowest $W_1$ distance across all conditions, with the gap widening substantially at higher rates. At $\lambda = 100$, $\tau = 0.5$, $\text{EAT}_{\text{cubic}}$ achieves $7\times$ lower distance than $\text{EAT}_{\text{sigmoid}}$. See Fig. 15 for results across all tested $\lambda$.

**Interim conclusion.** Both theoretically and empirically, $\text{EAT}_{\text{cubic}}$ demonstrates superior distributional fidelity compared to $\text{EAT}_{\text{sigmoid}}$ and GSM. But does distributional fidelity predict downstream performance? To answer this, we should also examine gradient quality, which is a critical factor in optimization.

### 3.3. Gradient analysis: direction, bias, and variance

Distributional fidelity measures how well a relaxation approximates the true Poisson, but optimization ultimately depends on gradient quality. In this section, we develop a principled framework for comparing gradient estimates across relaxation methods.

**Linear decoders enable exact gradients.** Evaluating gradient quality requires ground-truth gradients to compare against. How can we obtain these for an intractable ELBO? Linear decoders provide a rare opportunity. When the decoder is linear, $\text{dec}(\boldsymbol{z}; \boldsymbol{\Phi}) = \boldsymbol{\Phi}\boldsymbol{z}$, the expectation over the approximate posterior involves only first and second moments, yielding a closed-form ELBO. For a linear $\mathcal{P}$-VAE, we derive the exact reconstruction loss in section E

(Eq. (59)), along with its gradient (Eq. (64)) and Hessian (Eq. (69)). We will now use these analytical expressions as our ground truth to evaluate the Poisson relaxation methods.

**Decomposing gradient estimates.** In stochastic optimization, we estimate gradients from finite samples. A gradient estimate $\hat{\boldsymbol{g}}$ can be decomposed as:

$$\hat{\boldsymbol{g}} = \boldsymbol{g}^* + \boldsymbol{b} + \boldsymbol{\epsilon},$$

where $\boldsymbol{g}^*$ is the true gradient, $\boldsymbol{\epsilon}$ is zero-mean noise with covariance $\boldsymbol{\Sigma} = \mathbb{E}[\boldsymbol{\epsilon}\boldsymbol{\epsilon}^\top]$, and $\boldsymbol{b}$ is a systematic bias:

$$\boldsymbol{b} := \lim_{N \to \infty} \frac{1}{N} \sum_{i=1}^{N} (\hat{\boldsymbol{g}}_i - \boldsymbol{g}^*),$$

where $N$ is the number of Monte Carlo samples.

**Curvature-aware metrics.** Low bias and variance are desirable, but their impact on optimization depends on the loss landscape. In section F, we derive the optimization dynamics (Eq. (78)) via a second-order Taylor expansion of the loss $\mathcal{L} = -\text{ELBO}$, keeping up to the Hessian $\mathbf{H}$ (Eq. (69)). This reveals that the relevant quantities are *curvature-weighted*:

$$\text{BiasEnergy} := \boldsymbol{b}^\top \mathbf{H} \boldsymbol{b}, \tag{7}$$
$$\text{NoiseEnergy} := \text{Tr}(\mathbf{H}\boldsymbol{\Sigma}), \tag{8}$$

These metrics weight errors by local curvature: bias or noise along high-curvature directions harms optimization more than along flat directions.

**Cosine similarity metrics.** We also measure directional alignment between estimated and true gradients:

$$\text{CosMean} := \cos(\mathbb{E}[\hat{\boldsymbol{g}}], \boldsymbol{g}^*), \tag{9}$$
$$\text{CosSample} := \mathbb{E}[\cos(\hat{\boldsymbol{g}}, \boldsymbol{g}^*)]. \tag{10}$$

CosMean measures alignment of the expected gradient (including bias) with the ground truth, while CosSample measures the average alignment of individual samples.

**Both EAT methods provide reliable gradient estimates.** We empirically evaluate gradient quality using the linear $\mathcal{P}$-VAE framework. Following Vafaii et al. (2024), we trained $\mathcal{P}$-VAE models with linear encoders and decoders ($K = 512$ latent dimensions) on whitened natural image patches. We trained separate models using $\text{EAT}_{\text{sigmoid}}$, $\text{EAT}_{\text{cubic}}$, and GSM as the reparameterization method. As a baseline, we also included score function estimators with a simple variance reduction scheme (batch-average baseline subtraction). See Section G.1 for details.

After training, we estimated gradients of the ELBO reconstruction loss (Eq. (2)) with respect to the encoder log-rates

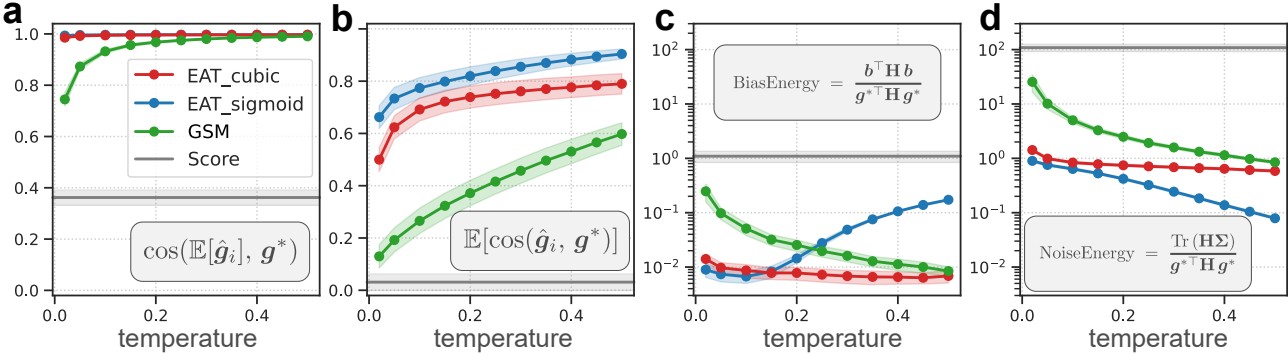

*Figure 3.* **Gradient quality analysis across relaxation methods.** We compare gradient estimates from $\text{EAT}_\text{sigmoid}$ (blue), $\text{EAT}_\text{cubic}$ (red), GSM (green), and score function with baseline (gray) across temperatures $\tau \in [0.02, 0.5]$ at firing rate $\lambda = 20$. **(a)** CosMean (Eq. (9)): cosine similarity between expected gradient and ground truth. **(b)** CosSample (Eq. (10)): average cosine similarity of individual gradient samples. **(c)** BiasEnergy (Eq. (7)): curvature-weighted squared bias. **(d)** NoiseEnergy (Eq. (8)): curvature-weighted noise variance. Both energy metrics are normalized by $g^{*\top}Hg^*$, providing an interpretable scale: values $\lesssim 0.1$ are negligible, while values $\gtrsim 1$ indicate errors dominate the true gradient. Both EAT methods achieve near-perfect directional alignment and low bias ($\sim 10^{-2}$) and noise ($\sim 10^0$) across all temperatures (but $\text{EAT}_\text{sigmoid}$ bias degrades as temperature increases). GSM shows elevated BiasEnergy at low temperatures and higher NoiseEnergy throughout. The score function baseline performs poorly across all metrics. Shaded regions indicate $\pm 1$ standard deviation across $N = 100$ Monte Carlo samples. See Fig. 16 for results across other rates and Fig. 17 for raw bias and variance.

using $N_\text{samples} = 100$ Monte Carlo samples, batch size $B = 90$, and firing rate $\lambda = 20$, repeated across all latent dimensions. We then computed the metrics introduced above (Eqs. (7) to (10)), utilizing the exact analytical expressions for linear decoders (Eqs. (59), (64) and (69)).

We report BiasEnergy and NoiseEnergy normalized by $g^{*\top}Hg^*$, the curvature-weighted energy of the true gradient itself (see Eq. (78)). This provides a natural interpretable scale: values $\ll 1$ indicate that bias or noise is negligible relative to the true gradient signal, values $\approx 1$ indicate comparable magnitude, and values $\gg 1$ indicate that errors dominate the gradient signal entirely.

Figure 3 shows the results. For CosMean (Fig. 3a), both EAT methods maintain near-perfect alignment ($> 0.99$) across all temperatures. In contrast, GSM starts poorly at low temperatures ($\sim 0.75$ at $\tau = 0.02$) and improves to $\sim 0.95$ as temperature increases ($\tau \approx 0.2$) until it matches the EAT methods at $\tau \geqslant 0.5$. Score method achieves $\sim 0.35$.

For CosSample (Fig. 3b), all pathwise methods improve with temperature as gradients become smoother. $\text{EAT}_\text{sigmoid}$ performs best, reaching $\sim 0.9$ at $\tau = 0.5$, followed by $\text{EAT}_\text{cubic}$ ($\sim 0.8$) and GSM ($\sim 0.6$). The score function baseline is barely above zero, indicating individual gradient samples are nearly orthogonal to the ground truth.

The curvature-aware metrics (Fig. 3c,d) reveal striking differences. $\text{EAT}_\text{cubic}$ achieves BiasEnergy $\sim 10^{-2}$ across all temperatures, well below the threshold where bias would impact optimization. At $\tau \lesssim 0.15$, $\text{EAT}_\text{sigmoid}$ matches $\text{EAT}_\text{cubic}$, while GSM is an order of magnitude worse. But as temperature increases, the reverse happens: GSM starts to improve, while $\text{EAT}_\text{sigmoid}$ degrades.

For NoiseEnergy (Fig. 3d), $\text{EAT}_\text{sigmoid}$ performs the best, starting at $\sim 10^0$ and decreasing to $\sim 10^{-1}$ at higher temperatures, while $\text{EAT}_\text{cubic}$ hovers around $\sim 10^0$. In contrast, GSM is an order of magnitude worse at low temperatures, but catches up with $\text{EAT}_\text{cubic}$ as the temperature increases. The score baseline suffers catastrophically at $\sim 10^2$: noise overwhelms the gradient by two orders of magnitude.

These results reveal an important dissociation: while $\text{EAT}_\text{sigmoid}$ showed the worst distributional fidelity (sections 3.1 and 3.2), its gradient estimates are excellent, comparable to or better than $\text{EAT}_\text{cubic}$, particularly in their NoiseEnergy. This suggests that distributional fidelity and gradient quality capture complementary aspects of relaxation quality that should be studied independently.

### 3.4. $\text{EAT}_\text{cubic}$ performs on par with exact gradients

So far, we have explored two distinct metrics— *distributional fidelity* and *gradient quality*—each revealing complementary insights into Poisson relaxation methods. $\text{EAT}_\text{cubic}$ provides the best distributional fidelity, while $\text{EAT}_\text{sigmoid}$ provides the best gradient quality. We now turn to the ultimate test: which method works best in practice?

For a conclusive answer, we evaluate across two paradigms: $\mathcal{P}$-VAE (Vafaii et al., 2024) and POGLM (Li et al., 2024a). These are well-motivated problems with relevance to both neuroscience and machine learning. While they differ in model structure and goals, they share two key aspects: both are latent variable models trained via ELBO maximization, and both require differentiating through stochastic integer Poisson samples. A reliable Poisson relaxation method should perform well on both tasks.

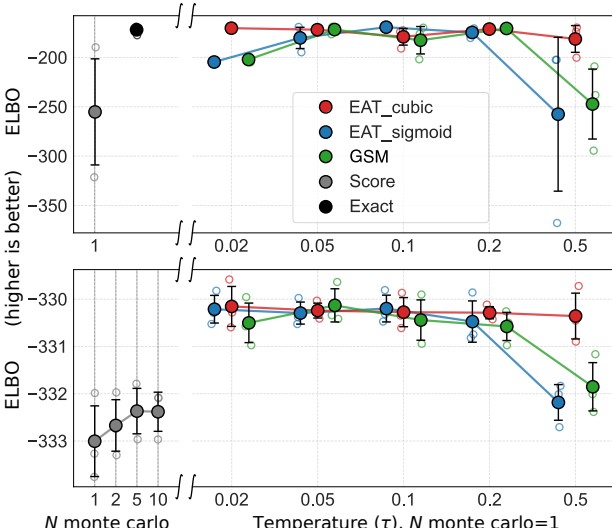

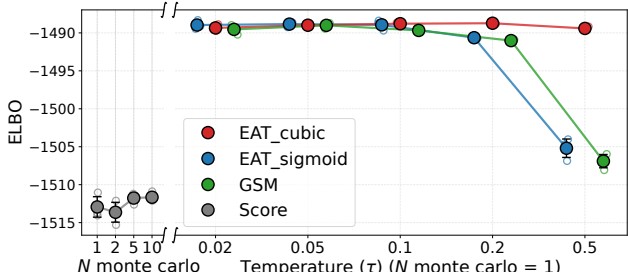

*Figure 5.* **POGLM validation ELBO on a retinal ganglion cell (RGC) dataset, assuming three hidden neurons.** *Left columns:* Score function baseline across MC sample sizes. *Right columns:* Pathwise methods across temperatures $\tau \in \{0.02, 0.05, 0.1, 0.2, 0.5\}$. The resulting curves match the patterns observed in the POGLM panel of Fig. 4 (generated using synthetic data). Across the whole temperature range, $\text{EAT}_{\text{cubic}}$ (red) achieves near-optimal performance, while $\text{EAT}_{\text{sigmoid}}$ (blue) and GSM (green) begin to degrade beyond $\tau = 0.2$. The score function baseline (gray) shows high variance and poor performance. Among all methods in the table, the $\text{EAT}_{\text{cubic}}$ method exhibits the most stable performance. See Fig. 13 for POGLM results under the assumptions of one and two hidden neurons.

GSM degrade substantially at $\tau = 0.5$, dropping to ELBO $\approx -250$ with increased variance across seeds, approaching the score function baseline. The same pattern holds for POGLM (bottom): $\text{EAT}_{\text{cubic}}$ achieves a stable performance (ELBO $\approx -330$) across all temperatures, while $\text{EAT}_{\text{sigmoid}}$ and GSM suffer at $\tau = 0.5$ with performance dropping to ELBO $\approx -332$. Figure 11 repeats the POGLM results across a more densely sampled $\tau \in [0.02, 0.5]$, confirming that only $\text{EAT}_{\text{cubic}}$ performs robustly across all temperatures.

**POGLM on retinal ganglion cells (RGC).** Following Li et al. (2024a), we applied all methods to a real retinal ganglion cell (RGC) dataset (Pillow & Scott, 2012) to evaluate the performance of the methods on real neural data. The real spike train in this dataset is recorded from 27 RGC neurons in a mouse performing a visual task for approximately 20 minutes. Neurons 1–16 are OFF cells, and neurons 17–27 are ON cells.

We first divided the spike train into 1,440 segments, each containing 100 time bins, and then shuffled these segments. The processed sequence was split into training and test sets, with the first 2/3 used for training and the remaining 1/3 for testing. We treated all 27 neurons as visible neurons and assumed the number of hidden neurons $H \in \{1, 2, 3\}$. Each method was trained for 30 epochs. All other training settings the same as previous POGLM experiments.

Across all values of $H$, $\text{EAT}_{\text{cubic}}$ is the only method robust to temperature. Specifically, $\text{EAT}_{\text{sigmoid}}$ and GSM start to degrade sharply at $\tau = 0.1$ (e.g., $H = 3$ shown in Fig. 5) while $\text{EAT}_{\text{cubic}}$ holds for larger $\tau$. This pattern also holds for $H = 1$ and $H = 2$, shown in Fig. 13. Overall, the POGLM results on real neural data closely mirror those obtained

*Figure 4.* **Validation ELBO across relaxation methods and temperatures.** *Top:* Linear $\mathcal{P}$-VAE trained on whitened natural image patches. *Bottom:* POGLM on synthetic neural data. *Left columns:* Score function baseline across MC sample sizes. *Right columns:* Pathwise methods across temperatures $\tau \in \{0.02, 0.05, 0.1, 0.2, 0.5\}$. For $\mathcal{P}$-VAE, the black marker indicates exact gradients (closed-form ELBO for linear decoders), representing an upperbound. $\text{EAT}_{\text{cubic}}$ (red) achieves near-optimal performance consistently across all temperatures, while $\text{EAT}_{\text{sigmoid}}$ (blue) and GSM (green) degrade substantially at $\tau = 0.5$. The score function baseline (gray) exhibits high variance and poor performance despite variance reduction; we can also see increased performance when the number of Monte Carlo sample sizes gets bigger. Error bars indicate $\pm 1$ std across three random seeds; open circles show individual runs. See Fig. 10 for the learned representations, and Fig. 11 for POGLM results across a denser sampling of $\tau \in [0.02, 0.5]$, including weight recovery results.

For $\mathcal{P}$-VAE, we use linear decoders which yield a closed-form ELBO, enabling training with *exact* gradients as an upperbound; for POGLM, we follow Li et al. (2024a). We include score function estimators (with batch-average baseline) as a lower bound, and explore $\tau \in \{0.02, 0.05, 0.1, 0.2, 0.5\}$ with three random seeds per condition. Both EAT and GSM require an upperbound $M$ (the number of exponential samples or categorical truncation limit, respectively), which we select adaptively per batch using the inverse CDF of the maximum rate to ensure computational efficiency without sacrificing accuracy. See sections G.1 to G.3 for details.

**ELBO performance.** Figure 4 shows the results. In $\mathcal{P}$-VAE (top), the exact condition achieves ELBO $\approx -169$, representing the best possible performance. All relaxation methods can reach this level at certain temperatures: $\text{EAT}_{\text{cubic}}$ at $\tau \in \{0.02, 0.05, 0.2\}$, GSM at $\tau \in \{0.05, 0.2\}$, and $\text{EAT}_{\text{sigmoid}}$ at $\tau = 0.1$. Crucially, only $\text{EAT}_{\text{cubic}}$ maintains this high performance consistently across all temperatures, including $\tau = 0.5$. In contrast, both $\text{EAT}_{\text{sigmoid}}$ and

from the synthetic experiments in Fig. 4, leading to the same conclusion: $\text{EAT}_{\text{cubic}}$ is the only method that performs reliably and robustly across the tested temperatures.

**The difficulty of adaptive temperature selection.** While the upperbound $M$ can be set adaptively using the inverse CDF (section G.3), no analogous solution exists for the temperature $\tau$. We explored this question (section H.1), analyzing gradient estimation as a regression task, and found that the optimal $\tau$ depends non-linearly on both the firing rate $\lambda$ and the specific objective function (Fig. 9). This lack of a universal scaling law forces practitioners using $\text{EAT}_{\text{sigmoid}}$ or GSM to perform costly grid searches, and as our results demonstrate, suboptimal choices can severely degrade performance. In sharp contrast, $\text{EAT}_{\text{cubic}}$ achieves strong performance across the full range we tested. For practitioners, this robustness translates directly into reduced engineering effort and more reliable training.

**Additional empirical results.** To test whether the preference for $\text{EAT}_{\text{cubic}}$ remains consistent across a range of settings, we conducted a broader set of experiments. We report these additional results in section H.3, and summarize the most important findings here.

- Section H.3.1: at a larger scale ($40 \times 40$ ImageNet patches), only $\text{EAT}_{\text{cubic}}$ closes the gap to exact gradients, while $\text{EAT}_{\text{sigmoid}}$ plateaus $\sim 35$ nats below regardless of $\tau$. This is a stronger statement than the main paper, where the gap was small enough to be plausibly closable by tuning $\tau$.
- Section H.3.2: the temperature robustness of $\text{EAT}_{\text{cubic}}$ transfers to over-dispersed Negative Binomial latents.
- Section H.3.3: on a downstream task (macaque V1 neural data prediction via linear readout from $\mathcal{P}$-VAE features), both relaxations outperform exact gradients, consistent with beneficial implicit regularization.
- Section H.3.4: in a limited-data regime, exact gradients overfit while EAT estimators remain stable.

The rest of section H.3 reports POGLM weight recovery, RGC results for $H \in \{1, 2\}$ hidden neurons, and wider sweeps over $\lambda$ and $\tau$ for distributional fidelity and gradient analysis. One exception is the $\mathcal{P}$-VAE decoder weights (Fig. 10), where $\text{EAT}_{\text{sigmoid}}$ produces the cleanest features, likely due to its $C^\infty$ smoothness compared to the $C^1$ cubic.

## 4. Conclusions and discussion

We systematically investigated two existing approaches to Poisson relaxations—$\text{EAT}_{\text{sigmoid}}$ and GSM—and identified a key limitation of the former: the sigmoid's infinite support causes substantial divergence between the relaxed and true Poisson distributions. Leveraging the theory of Poisson point processes, we derived closed-form expressions for the moments of the relaxed distribution, revealing that this divergence worsens with temperature. We addressed this issue by replacing the sigmoid with a cubic smoothstep function, whose compact support eliminates spurious contributions from events far outside the counting interval.

Our modified $\text{EAT}_{\text{cubic}}$ consistently outperformed both $\text{EAT}_{\text{sigmoid}}$ and GSM across distributional fidelity, gradient quality, temperature robustness, and downstream task performance, often matching or exceeding results obtained with exact gradients. We distill these findings into practical recommendations in Tab. 1.

**Distributional fidelity also matters.** Our results highlight a factor often overlooked in stochastic optimization for variational inference: *distributional fidelity matters*. While much attention has focused on reducing gradient variance, the distributional shift introduced by relaxation methods can be equally consequential. We encourage future work to evaluate distributional fidelity alongside traditional metrics.

**Limitations and future directions.** Our experiments focused on variational inference via ELBO optimization. Alternative approaches such as wake-sleep algorithms (Hinton et al., 1995; Bornschein & Bengio, 2015), reweighted wake-sleep (Le et al., 2020), or score-ascent methods (Kim et al., 2022) sidestep the need for custom gradient estimators entirely, and may offer complementary advantages in certain settings. While we demonstrated benefits in linear $\mathcal{P}$-VAE and POGLM settings, further validation on larger-scale models and diverse application domains would strengthen the generality of our conclusions.

In the main paper, we focused exclusively on Poisson latents, but the GSM method readily generalizes to arbitrary discrete distributions, including overdispersed alternatives such as the Negative Binomial (section H.2). This is particularly relevant for cortical data, where spike counts often exhibit overdispersion (Goris et al., 2014). Future work should extend our comparative analysis to these settings.

Conversely, cortical neurons sometimes also display sub-Poisson Fano factors (Churchland et al., 2010), suggesting an intriguing connection to the underdispersion inherent in our relaxation methods: the mathematical structure enabling gradient-based learning may inadvertently capture aspects of biological neural variability.

More broadly, the tension between discrete neural codes and the differentiability requirements of modern machine learning represents a fertile ground for NeuroAI research, where biological constraints may inspire algorithmic innovations.

# Acknowledgements

**Author contributions:**

**H.V.** and **C.L.** conceived the project and directed the research. **H.V.** developed the theoretical framework and the proofs, designed the validation protocols, and wrote the manuscript. **C.L.** contributed to experimental design, writing, editing, and structuring of the manuscript. **M.I.** and **H.Z.** performed the extensive experimental validation and produced the figures, under the supervision of **H.V.** and **C.L.**, respectively. **E.S.** contributed code and provided key feedback on manuscript organization. **Z.L.** assisted with experiments. **A.W.** provided senior supervision and funding. **J.L.Y.** provided senior supervision, funding, and critical feedback on the manuscript. All authors reviewed and approved the final manuscript.

# Impact Statement

This paper presents work whose goal is to advance the field of Machine Learning. There are many potential societal consequences of our work, none of which we feel must be specifically highlighted here.

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

# Appendix: A hitchhiker's guide to Poisson gradient estimation

## Contents

## A. Gradient estimators for variational inference

Variational inference optimizes the evidence lower bound (ELBO) with respect to two sets of parameters: the decoder parameters $\theta$ of the generative model $p(\boldsymbol{x}, \boldsymbol{z}; \theta)$, and the encoder parameters $\phi$ of the variational distribution $q(\boldsymbol{z}|\boldsymbol{x}; \phi)$. While the gradient with respect to $\theta$ is straightforward, the gradient with respect to $\phi$ requires more care since $\phi$ affects the distribution from which we sample. In this section, we derive the standard gradient estimators used in practice.

**Gradient with respect to decoder parameters.** Recall the ELBO definition from Eq. (1):

$$\text{ELBO}(\boldsymbol{x}; \theta, \phi) \;=\; \int q(\boldsymbol{z}|\boldsymbol{x}; \phi) \Big[ \ln p(\boldsymbol{x}, \boldsymbol{z}; \theta) - \ln q(\boldsymbol{z}|\boldsymbol{x}; \phi) \Big] \, d\boldsymbol{z}. \tag{11}$$

The gradient with respect to $\theta$ passes directly through the integrand:

$$
\begin{aligned}
\frac{\partial \text{ELBO}(\boldsymbol{x}; \theta, \phi)}{\partial \theta} \;&=\; \int q(\boldsymbol{z}|\boldsymbol{x}; \phi) \frac{\partial}{\partial \theta} \Big[ \ln p(\boldsymbol{x}, \boldsymbol{z}; \theta) - \ln q(\boldsymbol{z}|\boldsymbol{x}; \phi) \Big] \, d\boldsymbol{z} \\[2mm]
&\approx\; \frac{1}{N} \sum_{i=1}^{N} \frac{\partial}{\partial \theta} \Big[ \ln p(\boldsymbol{x}, \boldsymbol{z}^{(i)}; \theta) - \ln q(\boldsymbol{z}^{(i)}|\boldsymbol{x}; \phi) \Big] \\[2mm]
&=\; \frac{\partial}{\partial \theta} \widehat{\text{ELBO}}(\boldsymbol{x}; \theta, \phi),
\end{aligned}
\tag{12}
$$

where $\boldsymbol{z}^{(i)} \sim q(\boldsymbol{z}|\boldsymbol{x}; \phi)$ are samples from the variational distribution. Since the samples do not depend on $\theta$, we can simply evaluate the Monte Carlo estimate $\widehat{\text{ELBO}}$ and use automatic differentiation to compute the gradient.

### A.1. Score function estimator for encoder parameters

The gradient with respect to $\phi$ is more subtle because $\phi$ appears both in the sampling distribution and in the integrand. We cannot simply interchange differentiation and integration.

The *score function estimator* (also known as REINFORCE or the likelihood ratio method) handles this by using the identity $\nabla_\phi q = q \nabla_\phi \ln q$. Differentiating the ELBO at a fixed point $\phi_0$:

$$
\begin{aligned}
\frac{\partial \text{ELBO}(\boldsymbol{x}; \theta, \phi)}{\partial \phi} \;=\; & \int \Big[ \ln p(\boldsymbol{x}, \boldsymbol{z}; \theta) - \ln q(\boldsymbol{z}|\boldsymbol{x}; \phi_0) \Big] \frac{\partial q(\boldsymbol{z}|\boldsymbol{x}; \phi)}{\partial \phi} \, d\boldsymbol{z} \\[2mm]
& + \int q(\boldsymbol{z}|\boldsymbol{x}; \phi_0) \frac{\partial}{\partial \phi} \Big[ \ln p(\boldsymbol{x}, \boldsymbol{z}; \theta) - \ln q(\boldsymbol{z}|\boldsymbol{x}; \phi) \Big] \, d\boldsymbol{z}.
\end{aligned}
\tag{13}
$$

For the first term, we apply the log-derivative trick: $\partial_\phi q = q \, \partial_\phi \ln q$. For the second term, note that $\partial_\phi \ln p(\boldsymbol{x}, \boldsymbol{z}; \theta) = 0$ and that $\int \partial_\phi q \, d\boldsymbol{z} = \partial_\phi \int q \, d\boldsymbol{z} = \partial_\phi 1 = 0$. This yields:

$$
\begin{aligned}
\frac{\partial \text{ELBO}}{\partial \phi} \;&=\; \int q(\boldsymbol{z}|\boldsymbol{x}; \phi) \Big[ \ln p(\boldsymbol{x}, \boldsymbol{z}; \theta) - \ln q(\boldsymbol{z}|\boldsymbol{x}; \phi_0) \Big] \frac{\partial \ln q(\boldsymbol{z}|\boldsymbol{x}; \phi)}{\partial \phi} \, d\boldsymbol{z} \\[2mm]
&\approx\; \frac{1}{N} \sum_{i=1}^{N} \Big[ \ln p(\boldsymbol{x}, \boldsymbol{z}^{(i)}; \theta) - \ln q(\boldsymbol{z}^{(i)}|\boldsymbol{x}; \phi_0) \Big] \frac{\partial \ln q(\boldsymbol{z}^{(i)}|\boldsymbol{x}; \phi)}{\partial \phi}.
\end{aligned}
\tag{14}
$$

The term in brackets acts as a *reward signal* that weights the score function $\nabla_\phi \ln q$. In implementation, we compute $\ln q(\boldsymbol{z}^{(i)}|\boldsymbol{x}; \phi)$ and multiply by the `stop_gradient` (or `detach`) of the reward before backpropagation.

### A.2. Pathwise estimator for encoder parameters

When the latent variable can be expressed via the *reparameterization trick*—that is, $\boldsymbol{z} = r(\boldsymbol{\varepsilon}|\boldsymbol{x}; \phi)$ for some deterministic function $r$ and noise $\boldsymbol{\varepsilon} \sim \mathcal{E}(\boldsymbol{\varepsilon})$ independent of $\phi$—we can use the *pathwise estimator* instead:

$$
\begin{aligned}
\frac{\partial\,\mathrm{ELBO}(\boldsymbol{x}; \theta, \phi)}{\partial \phi} &= \frac{\partial}{\partial \phi} \int q(\boldsymbol{z}|\boldsymbol{x}; \phi) \Big[ \ln p(\boldsymbol{x}, \boldsymbol{z}; \theta) - \ln q(\boldsymbol{z}|\boldsymbol{x}; \phi) \Big] \, d\boldsymbol{z} \\
&= \frac{\partial}{\partial \phi} \int \mathcal{E}(\boldsymbol{\varepsilon}) \Big[ \ln p(\boldsymbol{x}, r(\boldsymbol{\varepsilon}|\boldsymbol{x}; \phi); \theta) - \ln q(r(\boldsymbol{\varepsilon}|\boldsymbol{x}; \phi)|\boldsymbol{x}; \phi) \Big] \, d\boldsymbol{\varepsilon} \\
&\approx \frac{\partial}{\partial \phi} \frac{1}{N} \sum_{i=1}^{N} \Big[ \ln p(\boldsymbol{x}, \boldsymbol{z}^{(i)}; \theta) - \ln q(\boldsymbol{z}^{(i)}|\boldsymbol{x}; \phi) \Big] \\
&= \frac{\partial}{\partial \phi} \widehat{\mathrm{ELBO}}(\boldsymbol{x}; \theta, \phi),
\end{aligned}
\tag{15}
$$

where $\boldsymbol{z}^{(i)} = r(\boldsymbol{\varepsilon}^{(i)}|\boldsymbol{x}; \phi)$ with $\boldsymbol{\varepsilon}^{(i)} \sim \mathcal{E}(\boldsymbol{\varepsilon})$. The key insight is that we have moved the $\phi$-dependence *inside* the integrand via the reparameterization, so differentiation and integration can be interchanged.

The pathwise estimator typically has much lower variance than the score function estimator, which is why reparameterization-based methods (such as the VAE) have been so successful.

Both the *Exponential Arrival Time* (EAT, Vafaii et al. (2024)) and *Gumbel-Softmax* (GSM, Li et al. (2024a)) methods extend this approach to Poisson latent variables by constructing differentiable relaxations of the Poisson distribution. We explore these methods next.

## B. The Exponential Arrival Time (EAT) relaxation method

Variational inference with discrete latent variables poses a fundamental challenge: we cannot backpropagate gradients through integer-valued samples. The reparameterization trick, which has proven so effective for continuous distributions like Gaussians, does not directly apply. The *Exponential Arrival Time* (EAT, (Vafaii et al., 2024)) method circumvents this obstacle by exploiting a beautiful connection between Poisson random variables and continuous-time point processes.

**Poisson as event counting.** The key insight is that a Poisson random variable can be viewed as counting events in a stochastic process. Consider events—neural spikes, photon arrivals, customer visits—occurring randomly in time at rate $\lambda$. The number of events $N$ falling within the unit interval $[0, 1]$ follows a Poisson distribution with parameter $\lambda$:

$$
N = |\{t_m : t_m \in [0, 1]\}| \sim \mathrm{Poisson}(\lambda)
\tag{16}
$$

where $\{t_1, t_2, \ldots\}$ are the arrival times of the events, and $|\cdot|$ indicates the cardinality (size) of a set.

**The inter-arrival time representation.** Where do these arrival times come from? For a Poisson process, they emerge from a simple generative procedure. Let $\Delta t_1, \Delta t_2, \ldots$ be independent draws from an $\mathrm{Exponential}(\lambda)$ distribution. We interpret each $\Delta t_i$ as the time duration between consecutive events. The arrival times are then cumulative sums:

$$
t_m = \sum_{i=1}^{m} \Delta t_i.
\tag{17}
$$

This representation is fully differentiable: we can sample $\Delta t_i$ using the reparameterization $\Delta t_i = -\log(u_i)/\lambda$ where $u_i \sim \mathrm{Uniform}(0, 1)$, and gradients flow through both the rate parameter $\lambda$ and the cumulative sum.

**The discreteness problem.** The count $N$ in (16) is computed by summing hard indicator functions:

$$
N = \sum_m \mathbf{1}[t_m < 1].
\tag{18}
$$

The indicator $\mathbf{1}[t_m < 1]$ is 1 if event $m$ arrives before time 1, and 0 otherwise. This is where discreteness enters: the indicator is a step function with zero gradient almost everywhere.

---

**Algorithm 1** Exponential arrival time (EAT) relaxation of the Poisson distribution

---

**Input:** Rate parameter $\lambda \in \mathbb{R}_{>0}$, number of exponential samples $M$, temperature $\tau \geqslant 0$

**Output:** Event counts $z \in \mathbb{R}_{\geqslant 0}$     {Non-negative integers only when $\tau = 0$, dense floats otherwise}

$\text{Exp} \leftarrow \text{Exponential}(\lambda)$     {Create exponential distribution}

$\Delta t_1, \ldots, \Delta t_M \leftarrow \text{Exp.rsample}((M,))$     {Sample inter-event times, $\Delta t \in \mathbb{R}_{>0}$}

$t_1, \ldots, t_M \leftarrow \text{cumsum}(\Delta t_1, \ldots, \Delta t_M)$     {Compute arrival times $t_m = \sum_{i=1}^{m} \Delta t_i$}

$\text{indicator}_m \leftarrow \boldsymbol{f_{\mathbf{approx}}}\left(\frac{1-t_m}{\tau}\right)$     {Soft indicator for events within unit time, e.g., $f_{\text{approx}}(\cdot) = \text{sigmoid}(\cdot)$}

$z \leftarrow \sum_{m=1}^{M} \text{indicator}_m$     {Sum contributions to get event counts}

**Return** $z$

---

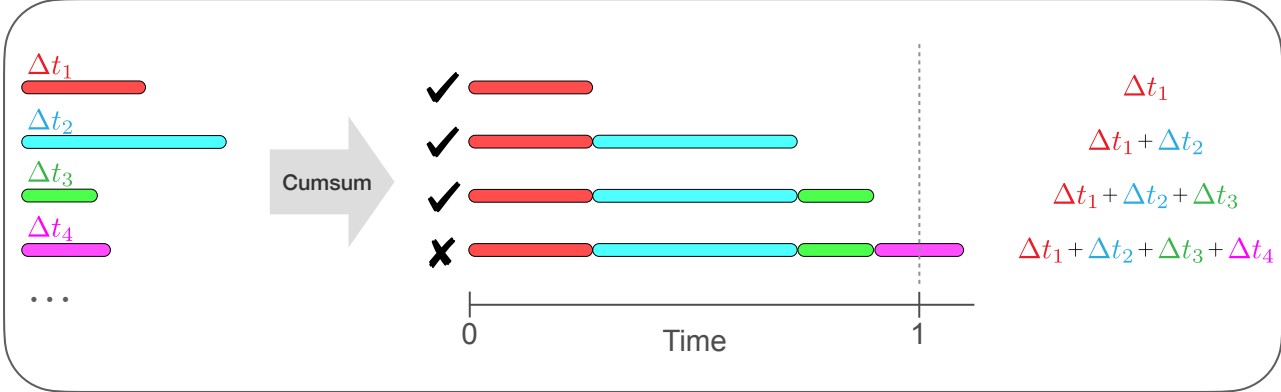

*Figure 6.* **Schematic of the EAT sampling process.** Given rate $\lambda$, independent inter-arrival times $\Delta t_i \sim \text{Exp}(\lambda)$ are cumulatively summed to determine absolute arrival times (right). The Poisson sample $z$ is defined as the count of events arriving before the unit time horizon (vertical dashed line). In this example, the first three events fall within the window, while the fourth exceeds it, resulting in a generated sample of $z = 3$. This process visualizes the steps detailed in Algorithm 1.

**Soft relaxation.**   The EAT method replaces the hard indicator with a smooth approximation. Let $f_{\text{approx}} : \mathbb{R} \to [0, 1]$ be a differentiable function that approximates the step function, and let $\tau > 0$ be a temperature parameter controlling the sharpness of the transition. The relaxed count is:

$$z = \sum_m f_{\text{approx}}\left(\frac{1 - t_m}{\tau}\right). \tag{19}$$

When $t_m \ll 1$ (event well inside the interval), the argument $(1 - t_m)/\tau$ is large and positive, so $f_{\text{approx}} \approx 1$. When $t_m \gg 1$ (event outside), the argument is large and negative, so $f_{\text{approx}} \approx 0$. Events near the boundary $t_m \approx 1$ contribute fractional values that interpolate smoothly.

The standard choice, introduced by Vafaii et al. (2024), is the logistic sigmoid: $f_{\text{approx}}(u) = \sigma(u) = 1/(1 + e^{-u})$.

**Temperature annealing.**   As $\tau \to 0$, the soft indicator sharpens toward the hard step function, and $z$ converges in distribution to the true Poisson count $N$. In practice, one often anneals the temperature during training: starting with a larger $\tau$ for smoother gradients, then decreasing $\tau$ to approach the discrete distribution as optimization progresses.

**The algorithm.**   Algorithm 1 summarizes the complete procedure, and Fig. 6 illustrates it. The EAT method samples $M$ inter-arrival times (chosen large enough to capture essentially all events that could fall within the unit interval), computes arrival times via cumulative summation, applies the soft indicator, and sums to obtain the relaxed count. All operations are differentiable, enabling gradient-based optimization of the rate parameter $\lambda$ and any upstream parameters that determine it.

## C. The Gumbel-Softmax (GSM) relaxation method

**Poisson as infinite categorical distribution.** Rather than thinking of a Poisson variable $z \sim \mathcal{P}\text{ois}(\lambda)$ from a point process counting perspective, we can expand it explicitly as an infinite categorical distribution, i.e., its probability mass function (PMF), and hence:

$$z \sim \text{Cat}(\pi_0, \pi_1, \dots), \quad \pi_m = \mathbb{P}[z = m] = \frac{\lambda^m e^{-\lambda}}{m!} \tag{20}$$

**Truncation.** The first step of the relaxation is to truncate the categorical distribution with a large enough upperbound $M$. Now, the approximated sample becomes

$$z \sim \text{Cat}(\pi_0/C, \pi_1/C, \dots, \pi_{M-1}/C), \tag{21}$$

where $C = \sum_{m=0}^{M-1} \pi_m$ is the normalizing factor.

**Gumbel-Softmax reparametrization.** The second step is to use the Gumbel-Softmax trick from (Jang et al., 2017; Maddison et al., 2017) to get a soft and gradient-enabled Gumbel-Softmax sample

$$\tilde{z} \sim \text{GS}\left(\frac{\pi_0}{C}, \frac{\pi_1}{C}, \dots, \frac{\pi_{M-1}}{C}\right). \tag{22}$$

Specifically, we sample $g_m \overset{\text{i.i.d.}}{\sim} \text{Gumbel}(0,1)$ for all $m \in \{0, \dots, M-1\}$, e.g., via $g_m = -\ln(-\ln u_m)$ with $u_m \overset{\text{i.i.d.}}{\sim} \text{Uniform}(0,1)$. Then, we compute the Softmax:

$$\tilde{z}_m = \frac{e^{\frac{\ln \pi_m - \ln C + g_m}{\tau}}}{\sum_{i=0}^{M-1} e^{\frac{\ln \pi_i - \ln C + g_i}{\tau}}} \tag{23}$$

so that $\tilde{z}_m$ is a soft one-hot representation over the simplex $\Delta^{M-1}$, i.e., $\tilde{z}_m > 0$ and $\sum_{m=0}^{M-1} \tilde{z}_m = 1$. Finally,

$$z = \sum_{m=0}^{M-1} m \tilde{z}_m. \tag{24}$$

Similar to the EAT relaxation method, in the zero-temperature limit $\tau \to 0$, $\tilde{z}_m$ becomes a hard one-hot sample from $\text{Cat}(\pi_0/C, \pi_1/C, \dots, \pi_{M-1}/C)$, so that exactly one component $\tilde{z}_k = 1$ for some $k \in \{0, \dots, M-1\}$, and the relaxed count becomes $z = k$.

**Numerical stability and the likelihood function.** We now show that we GSM has the property of normalization invariance, which can improve its numerical stability and efficiency. Observing that for each category $m$ in the GS distribution,

$$\frac{\pi_m}{C} = \frac{\lambda^m e^{-\lambda}}{m! C} \tag{25}$$

contains the common factor $\frac{e^{-\lambda}}{C}$ for all $m \in \{1, \dots, M\}$. Hence, we can define

$$\ln \pi'_m := m \ln \lambda - \ln(m!) = \ln \pi_m + \lambda, \tag{26}$$

so that

$$\tilde{z}_m = \frac{e^{\frac{\ln \pi_m - \ln C + g_m}{\tau}}}{\sum_{i=0}^{M-1} e^{\frac{\ln \pi_i - \ln C + g_i}{\tau}}} = \frac{e^{\frac{\ln \pi'_m + g_m}{\tau}}}{\sum_{i=0}^{M-1} e^{\frac{\ln \pi'_i + g_i}{\tau}}}. \tag{27}$$

This implies that we can directly feed the simplified $\pi'_m$ into the Gumbel-Softmax during sampling, and remove the unnecessary factors of the $C$ and $\lambda$ in its evaluation. Besides, this refactor also simplifies the log-likelihood function of the

---

**Algorithm 2** Gumbel-Softmax (GSM) relaxation of the Poisson distribution

---

**Input:** Rate parameter $\lambda \in \mathbb{R}_{>0}$, upperbound truncation number $M$, temperature $\tau \geqslant 0$

**Output:** Event counts $z \in \mathbb{R}_{\geqslant 0}$ {Non-negative integers only when $\tau = 0$, dense floats otherwise}

$\ln \pi_m \leftarrow m \ln \lambda - \ln(m!), \ \forall m \in \{0, \ldots, M-1\}$ {Compute GS logits}

$\mathrm{GS} \leftarrow \mathrm{Gumbel\text{-}Softmax}_\tau(\ln \pi_0, \ldots, \ln \pi_{M-1})$ {Create GS distribution}

$\tilde{z}_0, \ldots, \tilde{z}_{M-1} \leftarrow \mathrm{GS.\,rsample}()$ {Sample soft one-hot}

$z \leftarrow \sum_{m=0}^{M-1} m \tilde{z}_m$ {Aggregate the soft one-hot representation into a count}

**Return** $z$

---

GS distribution:

$$\ln p(\tilde{z}_0, \ldots, \tilde{z}_{M-1}; \pi'_0, \ldots, \pi'_{M-1})$$

$$= \ln \Gamma(M) + (M-1)\ln\tau - M \cdot \ln\left(\sum_{m=0}^{M-1} \frac{\pi_m \mathrm{e}^\lambda}{\tilde{z}_m^\tau}\right) + \sum_{m=1}^{M}\left((\ln \pi_m + \lambda) - (\tau+1)\ln\tilde{z}_m\right)$$

$$= \ln \Gamma(M) + (M-1)\ln\tau - M \cdot \left[\lambda + \ln\left(\sum_{m=0}^{M}\frac{\pi_m}{\tilde{z}_m^\tau}\right)\right] + \sum_{m=0}^{M-1}\left(\ln\pi_m - (\tau+1)\ln\tilde{z}_m\right) + \sum_{m=0}^{M-1}\lambda \qquad (28)$$

$$= \ln \Gamma(M) + (M-1)\ln\tau - M \cdot \ln\left(\sum_{m=0}^{M-1}\frac{\pi_m}{\tilde{z}_m^\tau}\right) \cancel{-M\cdot\lambda} + \sum_{m=0}^{M-1}\left(\ln\pi_m - (\tau+1)\ln\tilde{z}_m\right) \cancel{+M\cdot\lambda}$$

$$= \ln p(\tilde{z}_0, \ldots, \tilde{z}_{M-1}; \pi_0, \ldots, \pi_{M-1})$$

**The algorithm.** Algorithm 2 summarizes the complete procedure. The method samples Gumbel-Softmax soft one-hot samples $\tilde{z}_0, \ldots, \tilde{z}_{M-1}$ with a large enough truncation $M$ and then forms the final relaxed count by the category-weighted aggregation $z = \sum_{m=0}^{M-1} m\tilde{z}_m$. All operations are differentiable, enabling gradient-based optimization of the rate parameter $\lambda$ and any upstream parameters that determine it.

## D. Distributional fidelity of the EAT Poisson relaxation method

Both the EAT (section B) and GSM (section C) methods produce continuous relaxations of Poisson random variables, but how faithfully do these relaxations preserve the statistical properties of the true distribution? We can address this theoretically for EAT, through the application of well-known results in the theory of point processes.

In this section, we derive exact expressions for the first and second moments of the EAT relaxed count $z$ as a function of the temperature parameter $\tau$ and the choice of soft indicator $f_{\mathrm{approx}}$ (see Algorithm 1). These results allow us to quantify the bias in mean and variance introduced by the relaxation, and to compare different choices of $f_{\mathrm{approx}}$.

We proceed as follows:

- **Section D.1**: We introduce Campbell's theorem, a classical result from point process theory that converts expectations of sums over random points into deterministic integrals.

- **Section D.2**: We apply Campbell's theorem to the relaxed count $z$, deriving general expressions for the mean and variance in terms of integrals of $f_{\mathrm{approx}}$.

- **Section D.3**: We evaluate these integrals explicitly for sigmoid function, obtaining closed-form moment expressions.

- **Section D.4**: We derive analogous expressions for the cubic smoothstep function, which improves distributional fidelity.

- **Section D.5**: We analyze the Fano factor (variance-to-mean ratio) and discuss its implications.

- **Section D.6**: We summarize all results in a unified table.

## D.1. Campbell's theorem

Campbell's theorem provides a powerful tool for computing expectations of sums over Poisson point processes. The result dates back to Campbell (1909), with modern treatments available in standard references on point processes (Kingman, 1992; Daley & Vere-Jones, 2003).

**Intuition.** Consider a Poisson process $\Pi = \{t_1, t_2, \ldots\}$ with rate $\lambda$ on $[0, \infty)$, and suppose we want to compute the expected value of $\sum_{t \in \Pi} f(t)$ for some function $f$. The key insight is to think infinitesimally: partition the time axis into tiny intervals of width $dt$. In each interval $[t, t + dt]$:

- The probability of having exactly one point is $\lambda \, dt$ (plus higher-order terms).
- If there is a point, it contributes $f(t)$ to the sum.

Thus, the expected contribution from the interval $[t, t + dt]$ is approximately $f(t) \cdot \lambda \, dt$. Summing (integrating) over all intervals gives $\lambda \int_0^\infty f(t) \, dt$.

**Formal statement.** Let $\Pi$ be a homogeneous Poisson point process on $[0, \infty)$ with rate $\lambda > 0$, and let $f : [0, \infty) \to \mathbb{R}$ be a measurable function. Campbell's theorem states:

$$\mathbb{E}\left[\sum_{t \in \Pi} f(t)\right] = \lambda \int_0^\infty f(t) \, dt, \tag{29}$$

provided the integral on the right-hand side is well-defined (i.e., $\int_0^\infty |f(t)| \, dt < \infty$).

For the second moment, we expand the square and use the independence property of Poisson processes to handle cross-terms:

$$\mathbb{E}\left[\left(\sum_{t \in \Pi} f(t)\right)^2\right] = \lambda \int_0^\infty f(t)^2 \, dt + \lambda^2 \left(\int_0^\infty f(t) \, dt\right)^2. \tag{30}$$

The first term arises from "diagonal" contributions (a single point squared), while the second term arises from "off-diagonal" contributions (pairs of distinct points). A direct consequence is the variance formula:

$$\mathrm{Var}\left(\sum_{t \in \Pi} f(t)\right) = \lambda \int_0^\infty f(t)^2 \, dt. \tag{31}$$

## D.2. Applying Campbell's theorem to EAT

Recall from Eq. (19) that the relaxed count is:

$$z = \sum_{t \in \Pi} f_{\mathrm{approx}}\left(\frac{1 - t}{\tau}\right),$$

where $\Pi$ is a Poisson process with rate $\lambda$. This is exactly the form required by Campbell's theorem, with $f(t) = f_{\mathrm{approx}}\left(\frac{1-t}{\tau}\right)$.

**First moment.** Applying Eq. (29):

$$\mathbb{E}[z] = \lambda \int_0^\infty f_{\mathrm{approx}}\left(\frac{1 - t}{\tau}\right) dt. \tag{32}$$

To evaluate this integral, we substitute $u = (1 - t)/\tau$, which gives $t = 1 - \tau u$ and $dt = -\tau \, du$. The limits transform as: $t = 0 \Rightarrow u = 1/\tau$ and $t = \infty \Rightarrow u = -\infty$. Thus:

$$\mathbb{E}[z] = \lambda \int_{1/\tau}^{-\infty} f_{\mathrm{approx}}(u) \cdot (-\tau) \, du$$

$$= \lambda \tau \int_{-\infty}^{1/\tau} f_{\mathrm{approx}}(u) \, du \tag{33}$$

We define the *mean factor*:

$$c(\tau) := \tau \int_{-\infty}^{1/\tau} f_{\mathrm{approx}}(u)\,du, \qquad \text{so that } \ \mathbb{E}[z] = \lambda \cdot c(\tau) \tag{34}$$

For the relaxation to be unbiased, we would need $c(\tau) = 1$.

**Variance.** Applying Eq. (31) with the same substitution:

$$\mathrm{Var}(z) = \lambda \int_0^\infty f_{\mathrm{approx}}\left(\frac{1-t}{\tau}\right)^2 dt$$

$$= \lambda\tau \int_{-\infty}^{1/\tau} f_{\mathrm{approx}}(u)^2\,du. \tag{35}$$

We define the *variance factor*:

$$v(\tau) := \tau \int_{-\infty}^{1/\tau} f_{\mathrm{approx}}(u)^2\,du, \qquad \text{so that } \ \mathrm{Var}(z) = \lambda \cdot v(\tau) \tag{36}$$

For a true Poisson distribution, we have $\mathrm{Var}(N) = \lambda$, so matching Poisson variance requires $v(\tau) = 1$.

**Underdispersion is unavoidable.** Since $f_{\mathrm{approx}}$ takes values in $[0,1]$, we have $f_{\mathrm{approx}}(u)^2 \leqslant f_{\mathrm{approx}}(u)$ for all $u$, with equality only when $f_{\mathrm{approx}}(u) \in \{0,1\}$. This immediately implies $v(\tau) \leqslant c(\tau)$, with strict inequality whenever $f_{\mathrm{approx}}$ takes values strictly between 0 and 1. As we will see, this means the relaxed distribution is always *underdispersed* relative to the true Poisson distribution.

### D.3. Explicit formulas: sigmoid

We now evaluate the integrals in Eqs. (34) and (36) for the sigmoid function $\sigma(u) = (1 + e^{-u})^{-1}$.

**Antiderivatives.** For the first moment, we need $\int \sigma(u)\,du$. Noting that $\sigma(u) = e^u/(1+e^u)$, we recognize this as the derivative of $\ln(1+e^u)$:

$$\int \sigma(u)\,du = \ln(1+e^u) = \mathrm{softplus}(u). \tag{37}$$

For the variance, we need $\int \sigma(u)^2\,du$. We use the identity:

$$\sigma(u)^2 = \sigma(u)\cdot\sigma(u) = \sigma(u) - \sigma(u)(1-\sigma(u)) = \sigma(u) - \sigma'(u),$$

where we used $\sigma'(u) = \sigma(u)(1-\sigma(u))$. Integrating both sides:

$$\int \sigma(u)^2\,du = \int \sigma(u)\,du - \int \sigma'(u)\,du = \mathrm{softplus}(u) - \sigma(u). \tag{38}$$

**Evaluating the definite integrals.** For the mean factor, using Eq. (37):

$$c(\tau) = \tau\,[\mathrm{softplus}(u)]_{-\infty}^{1/\tau}$$

$$= \tau\left[\mathrm{softplus}(1/\tau) - \lim_{u\to-\infty}\mathrm{softplus}(u)\right]$$

$$= \tau\cdot\mathrm{softplus}(1/\tau), \tag{39}$$

since $\text{softplus}(u) \to 0$ as $u \to -\infty$.

For the variance factor, using Eq. (38):

$$
\begin{aligned}
v(\tau) &= \tau\left[\text{softplus}(u) - \sigma(u)\right]_{-\infty}^{1/\tau} \\
&= \tau\left[\text{softplus}(1/\tau) - \sigma(1/\tau) - (0 - 0)\right] \\
&= \tau\left[\text{softplus}(1/\tau) - \sigma(1/\tau)\right],
\end{aligned}
\tag{40}
$$

since both $\text{softplus}(u) \to 0$ and $\sigma(u) \to 0$ as $u \to -\infty$.

**Final expressions.** Letting $a = \text{softplus}(1/\tau) = \ln(1 + e^{1/\tau})$ and $b = \sigma(1/\tau) = 1/(1 + e^{-1/\tau})$, we have:

$$
c_{\text{sigmoid}}(\tau) = \tau \cdot a = \tau \ln(1 + e^{1/\tau})
\tag{41}
$$

$$
v_{\text{sigmoid}}(\tau) = \tau(a - b) = \tau\left[\ln(1 + e^{1/\tau}) - \frac{1}{1 + e^{-1/\tau}}\right]
\tag{42}
$$

**Limiting behavior.** As $\tau \to 0$, we have $1/\tau \to \infty$, so $a \to 1/\tau$ and $b \to 1$. Thus:

$$
c_{\text{sigmoid}}(\tau) \to \tau \cdot \frac{1}{\tau} = 1, \qquad v_{\text{sigmoid}}(\tau) \to \tau\left(\frac{1}{\tau} - 1\right) \to 1.
$$

Both factors approach 1 as the temperature decreases, recovering the true Poisson statistics in the limit.

### D.4. Explicit formulas: cubic smoothstep

The cubic smoothstep (also known as Hermite interpolation) offers an alternative soft indicator with *compact support*:

$$
f_{\text{cubic}}(u) = \begin{cases}
0, & (u < -1) \\
3w^2 - 2w^3, \quad w = \dfrac{u+1}{2}, & (-1 \leqslant u \leqslant 1) \\
1, & (u > 1)
\end{cases}
\tag{43}
$$

This function is $C^1$ continuous (with zero derivatives at $u = \pm 1$) and transitions smoothly from 0 to 1 over the interval $[-1, 1]$. Figure 7 visualizes the cubic smoothstep function alongside sigmoid.

The compact support leads to qualitatively different behavior depending on whether the temperature is small ($\tau \leqslant 1$) or large ($\tau > 1$).

**Case $\tau \leqslant 1$: Full transition captured.** When $\tau \leqslant 1$, the upper integration limit $1/\tau \geqslant 1$ lies at or beyond the transition region. We split the integral:

$$
\int_{-\infty}^{1/\tau} f_{\text{cubic}}(u)\, du = \underbrace{\int_{-\infty}^{-1} 0\, du}_{=0} + \underbrace{\int_{-1}^{1} f_{\text{cubic}}(u)\, du}_{=I_1} + \underbrace{\int_{1}^{1/\tau} 1\, du}_{=1/\tau - 1}.
\tag{44}
$$

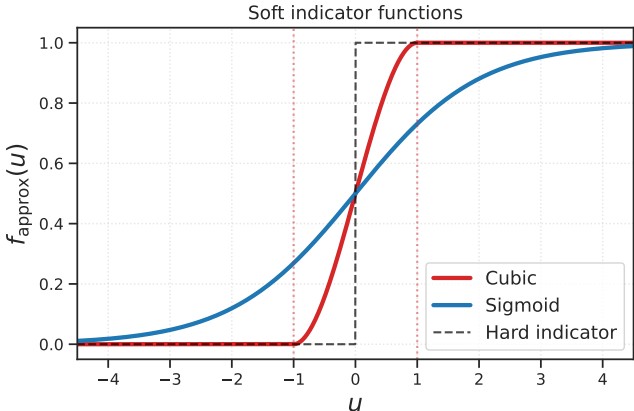

*Figure 7.* **Soft indicator functions for the EAT relaxation.** The hard indicator $1[u > 0]$ (black dashed) is non-differentiable. The sigmoid (blue) provides a $C^\infty$ smooth approximation but has infinite support, with tails extending to $\pm\infty$. The cubic smoothstep (red) is $C^1$ smooth with compact support on $[-1, 1]$ (dotted lines): it equals exactly 0 for $u < -1$ and exactly 1 for $u > 1$. This compact support eliminates "ghost spike" contributions from events far outside the counting interval, yielding superior distributional fidelity (section 3.1).

To evaluate $I_1$, we substitute $w = (u+1)/2$, so $du = 2\,dw$, and the limits become $w = 0$ to $w = 1$:

$$
\begin{aligned}
I_1 &= \int_0^1 (3w^2 - 2w^3) \cdot 2\,dw \\
&= 2\left[w^3 - \frac{w^4}{2}\right]_0^1 \\
&= 2\left(1 - \frac{1}{2}\right) = 1.
\end{aligned}
\tag{45}
$$

Therefore, from Eq. (44):

$$
\int_{-\infty}^{1/\tau} f_{\mathrm{cubic}}(u)\,du = 0 + 1 + \left(\frac{1}{\tau} - 1\right) = \frac{1}{\tau},
$$

and the mean factor is:

$$
\boxed{c_{\mathrm{cubic}}(\tau) = \tau \cdot \frac{1}{\tau} = 1 \qquad \text{for } \tau \leqslant 1.}
\tag{46}
$$

*The cubic smoothstep has no mean bias for $\tau \leqslant 1$.*

For the variance factor, we need $\int_{-\infty}^{1/\tau} f_{\mathrm{cubic}}(u)^2\,du$. Using the same decomposition:

$$
\int_{-\infty}^{1/\tau} f_{\mathrm{cubic}}(u)^2\,du = 0 + I_2 + \left(\frac{1}{\tau} - 1\right),
\tag{47}
$$

where $I_2 = \int_{-1}^1 f_{\mathrm{cubic}}(u)^2\,du$. Expanding $(3w^2 - 2w^3)^2 = 9w^4 - 12w^5 + 4w^6$:

$$
\begin{aligned}
I_2 &= \int_0^1 (9w^4 - 12w^5 + 4w^6) \cdot 2\,dw \\
&= 2\left[\frac{9w^5}{5} - 2w^6 + \frac{4w^7}{7}\right]_0^1 \\
&= 2\left(\frac{9}{5} - 2 + \frac{4}{7}\right) \\
&= 2 \cdot \frac{63 - 70 + 20}{35} = \frac{26}{35}.
\end{aligned}
\tag{48}
$$

Therefore:

$$\int_{-\infty}^{1/\tau} f_{\text{cubic}}(u)^2 \, du \;=\; \frac{26}{35} + \frac{1}{\tau} - 1 \;=\; \frac{1}{\tau} - \frac{9}{35},$$

and the variance factor is:

$$v_{\text{cubic}}(\tau) \;=\; \tau\left(\frac{1}{\tau} - \frac{9}{35}\right) \;=\; 1 - \frac{9\tau}{35} \qquad \text{for } \tau \leqslant 1. \tag{49}$$

**Case $\tau > 1$: Partial transition.** When $\tau > 1$, the upper limit $1/\tau < 1$ falls inside the transition region. Define $\beta = (1/\tau + 1)/2 = (1+\tau)/(2\tau)$, which lies in $(1/2, 1)$. The integrals become:

$$\int_{-\infty}^{1/\tau} f_{\text{cubic}}(u) \, du \;=\; \int_0^\beta (3w^2 - 2w^3) \cdot 2 \, dw$$

$$= 2\left[\beta^3 - \frac{\beta^4}{2}\right] \;=\; \beta^3(2 - \beta). \tag{50}$$

Similarly:

$$\int_{-\infty}^{1/\tau} f_{\text{cubic}}(u)^2 \, du \;=\; 2\left[\frac{9\beta^5}{5} - 2\beta^6 + \frac{4\beta^7}{7}\right]. \tag{51}$$

The moment factors for $\tau > 1$ are:

$$c_{\text{cubic}}(\tau) \;=\; \tau\beta^3(2 - \beta), \qquad v_{\text{cubic}}(\tau) \;=\; \tau\left(\frac{18\beta^5}{5} - 4\beta^6 + \frac{8\beta^7}{7}\right) \qquad \text{for } \tau > 1, \tag{52}$$

where $\beta = (1+\tau)/(2\tau)$. These expressions are continuous at $\tau = 1$ (where $\beta = 1$).

### D.5. Fano factor analysis

The **Fano factor** is defined as the ratio of variance to mean:

$$F \;=\; \frac{\text{Var}(z)}{\mathbb{E}[z]} \;=\; \frac{\lambda \cdot v(\tau)}{\lambda \cdot c(\tau)} \;=\; \frac{v(\tau)}{c(\tau)}. \tag{53}$$

For a true Poisson distribution, $F = 1$ exactly. Distributions with $F < 1$ are called *sub-Poisson* or underdispersed.

Since the $\lambda$-dependence cancels, the Fano factor depends only on the temperature and the choice of $f_{\text{approx}}$.

**Sigmoid.**

$$F_{\text{sigmoid}}(\tau) \;=\; \frac{\tau(a - b)}{\tau a} \;=\; 1 - \frac{\sigma(1/\tau)}{\text{softplus}(1/\tau)} \tag{54}$$

**Cubic ($\tau \leqslant 1$).** Since $c_{\text{cubic}}(\tau) = 1$:

$$F_{\text{cubic}}(\tau) \;=\; v_{\text{cubic}}(\tau) \;=\; 1 - \frac{9\tau}{35} \qquad \text{for } \tau \leqslant 1. \tag{55}$$

**Comparison.** Both relaxations are sub-Poisson ($F < 1$) for all $\tau > 0$, as predicted by the general argument in Section D.2. However, the cubic preserves the Fano factor substantially better: at $\tau = 0.5$, the cubic has $F = 0.87$ while the sigmoid drops to $F = 0.59$.

This difference arises from the compact support of the cubic: events far from the boundary contribute exactly 0 or 1, with only a narrow region of fractional contributions. The sigmoid's infinite tails create a diffuse "haze" of partial contributions that inflates the mean while reducing variance.

### D.6. Summary

Tab. 2 collects all moment formulas derived in this section, and Fig. 8 visualizes them as a function of temperature $\tau$.

*Table 2.* Summary of moment factors for EAT relaxations. Ideal Poisson fidelity corresponds to $c = v = F = 1$. **Bold** indicates a closer approximation of the true Poisson. Notation: $a \coloneqq \mathrm{softplus}(1/\tau) = \ln(1 + e^{1/\tau})$, $b \coloneqq \mathrm{sigmoid}(1/\tau) = (1 + e^{-1/\tau})^{-1}$.

| | Mean Factor | Variance Factor | Fano Factor |
|---|---|---|---|
| | $c(\tau)$ (34) | $v(\tau)$ (36) | $F(\tau)$ (53) |
| | $\mathbb{E}[z] = \lambda \cdot c$ | $\mathrm{Var}(z) = \lambda \cdot v$ | $F(\tau) = v/c$ |
| **Formulas** | | | |
| Sigmoid | $\tau a$ | $\tau(a - b)$ | $1 - b/a$ |
| Cubic ($\tau \leqslant 1$) | $1$ | $1 - \dfrac{9\tau}{35}$ | $1 - \dfrac{9\tau}{35}$ |
| **$\tau = 0.1$** | | | |
| Sigmoid | **1.00** | 0.90 | 0.90 |
| Cubic | **1.00** | **0.97** | **0.97** |
| **$\tau = 0.5$** | | | |
| Sigmoid | 1.06 | 0.62 | 0.59 |
| Cubic | **1.00** | **0.87** | **0.87** |
| **$\tau = 1.0$** | | | |
| Sigmoid | 1.31 | 0.58 | 0.44 |
| Cubic | **1.00** | **0.74** | **0.74** |

The cubic smoothstep achieves superior distributional fidelity across all metrics:

- **Mean:** The cubic has $c(\tau) = 1$ exactly for $\tau \leqslant 1$, while the sigmoid overshoots ($c > 1$).

- **Variance:** Both are underdispersed, but the cubic retains substantially more variance (e.g., 87% vs. 59% at $\tau = 0.5$).

- **Fano factor:** The cubic's Fano factor remains closer to the Poisson value of 1.

The only tradeoff is smoothness: the sigmoid is $C^\infty$ while the cubic is only $C^1$. In practice, $C^1$ continuity is typically sufficient for gradient-based optimization.

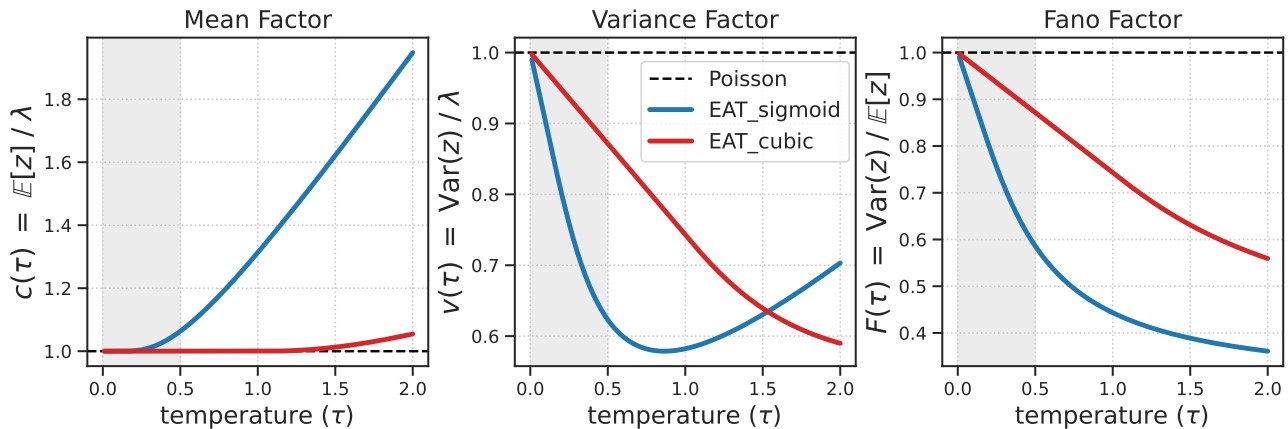

*Figure 8.* **Theoretical moment factors as a function of temperature.** Analytical plots of the mean factor $c(\tau)$ (Eq. (34)), variance factor $v(\tau)$ (Eq. (36)), and Fano factor $F(\tau)$ (Eq. (53)) for $\text{EAT}_{\text{sigmoid}}$ (blue) and $\text{EAT}_{\text{cubic}}$ (red). **Left:** $\text{EAT}_{\text{cubic}}$ is theoretically unbiased ($c = 1$) for $\tau \leq 1$, whereas $\text{EAT}_{\text{sigmoid}}$ inflates the mean as temperature increases. **Middle:** Both methods are underdispersed ($v < 1$), but $\text{EAT}_{\text{cubic}}$ retains significantly more variance than $\text{EAT}_{\text{sigmoid}}$. **Right:** The Fano factor of $\text{EAT}_{\text{cubic}}$ remains closer to the ideal Poisson value of 1. The gray shaded area indicates the range of temperatures explored in our experiments. Related to Fig. 1 (empirical validation).

## E. Poisson VAE analytical deviations: loss, gradients, and Hessian

Let the input data be a vector $\boldsymbol{x} \in \mathbb{R}^M$. We define the latent log-rates as $\boldsymbol{u} \in \mathbb{R}^K$ and the decoder weights as $\boldsymbol{\Phi} \in \mathbb{R}^{M \times K}$. The firing rates are given by the element-wise exponential, which we use to construct the approximate posterior:

$$\boldsymbol{\lambda} = \exp(\boldsymbol{u}) \in \mathbb{R}_+^K, \qquad q_{\boldsymbol{u}}(\boldsymbol{z}|\boldsymbol{x}) = \mathcal{P}\text{ois}(\boldsymbol{z}; \boldsymbol{\lambda}) = \prod_{i=1}^K \frac{e^{-\lambda_i} \lambda_i^{z_i}}{z_i!}. \tag{56}$$

The reconstruction loss is given by:

$$\mathcal{L}_{\text{recon.}}(\boldsymbol{u}, \boldsymbol{\Phi}) = \mathbb{E}_{\boldsymbol{z} \sim q_{\boldsymbol{u}}(\boldsymbol{z}|\boldsymbol{x})} \left[ \|\boldsymbol{x} - \boldsymbol{\Phi}\boldsymbol{z}\|_2^2 \right]. \tag{57}$$

Because the objective is quadratic with respect to the latent variable $\boldsymbol{z}$, we can decompose the expected loss into the loss evaluated at the mean plus a trace term accounting for the posterior variance. We utilize the canonical identity for the expectation of a quadratic form of a random variable $\boldsymbol{z}$ with mean $\boldsymbol{\lambda}$ and covariance $\boldsymbol{\Sigma}$:

$$\mathbb{E}\left[ \|\boldsymbol{x} - \boldsymbol{\Phi}\boldsymbol{z}\|_2^2 \right] = \|\boldsymbol{x} - \boldsymbol{\Phi}\boldsymbol{\lambda}\|_2^2 + \text{Tr}\left( \boldsymbol{\Phi}^\top \boldsymbol{\Phi} \boldsymbol{\Sigma} \right). \tag{58}$$

Using the fact that the Poisson distribution has equal mean and variance (implying $\boldsymbol{\Sigma} = \text{diag}(\boldsymbol{\lambda})$), we can analytically compute $\mathcal{L}_{\text{recon.}}(\boldsymbol{u}, \boldsymbol{\Phi})$. We find that it consists of a Mean Squared Error (MSE) term and a variance penalty term:

$$\boxed{\mathcal{L}_{\text{recon.}}(\boldsymbol{u}, \boldsymbol{\Phi}) = \underbrace{\|\boldsymbol{x} - \boldsymbol{\Phi}\boldsymbol{\lambda}\|_2^2}_{\text{MSE}} + \underbrace{\boldsymbol{\lambda}^\top \boldsymbol{d}}_{\substack{\text{variance} \\ \text{penalty}}}} \tag{59}$$

Here, $\boldsymbol{d} \in \mathbb{R}^K$ is the vector containing the squared norms of the dictionary elements:

$$\boldsymbol{d} = \text{diag}(\boldsymbol{\Phi}^\top \boldsymbol{\Phi}). \tag{60}$$

We now derive the gradient and Hessian of the reconstruction loss $\mathcal{L}_{\text{recon.}}$ with respect to the log-rates $\boldsymbol{u}$. For notational clarity, in the remainder of this section, we drop the explicit dependence on the decoder weights $\boldsymbol{\Phi}$ and use: $\mathcal{L}(\boldsymbol{u}) \equiv \mathcal{L}_{\text{recon.}}(\boldsymbol{u}, \boldsymbol{\Phi})$.

**Gradient derivation.** First, we define the Gram matrix of the decoder weights as $\mathbf{G} = \Phi^\top\Phi$. Expanding the MSE term:

$$
\begin{aligned}
\|\boldsymbol{x} - \Phi\boldsymbol{\lambda}\|_2^2 &= (\boldsymbol{x} - \Phi\boldsymbol{\lambda})^\top(\boldsymbol{x} - \Phi\boldsymbol{\lambda}) \\
&= \boldsymbol{x}^\top\boldsymbol{x} - 2\boldsymbol{x}^\top\Phi\boldsymbol{\lambda} + \boldsymbol{\lambda}^\top\Phi^\top\Phi\boldsymbol{\lambda} \\
&= \boldsymbol{x}^\top\boldsymbol{x} - 2(\Phi^\top\boldsymbol{x})^\top\boldsymbol{\lambda} + \boldsymbol{\lambda}^\top\mathbf{G}\boldsymbol{\lambda}.
\end{aligned}
\tag{61}
$$

The variance penalty is simply $\boldsymbol{d}^\top\boldsymbol{\lambda}$ and $\boldsymbol{d} \equiv \mathrm{diag}(\mathbf{G})$.

We first take the derivative with respect to the rates $\boldsymbol{\lambda}$. Note that $\nabla_{\boldsymbol{\lambda}}(\boldsymbol{\lambda}^\top\mathbf{G}\boldsymbol{\lambda}) = 2\mathbf{G}\boldsymbol{\lambda}$ because $\mathbf{G}$ is symmetric.

$$
\nabla_{\boldsymbol{\lambda}}\mathcal{L} = -2\Phi^\top\boldsymbol{x} + 2\mathbf{G}\boldsymbol{\lambda} + \boldsymbol{d}.
\tag{62}
$$

Next, we apply the chain rule to find the gradient with respect to $\boldsymbol{u}$. Since $\boldsymbol{\lambda} = \exp(\boldsymbol{u})$, the Jacobian is diagonal:

$$
\frac{\partial\boldsymbol{\lambda}}{\partial\boldsymbol{u}} = \mathrm{diag}(\boldsymbol{\lambda})
\tag{63}
$$

Thus, the gradient $\boldsymbol{g} = \nabla_{\boldsymbol{u}}\mathcal{L} = \mathrm{diag}(\boldsymbol{\lambda})\nabla_{\boldsymbol{\lambda}}\mathcal{L}$ is:

$$
\boxed{\boldsymbol{g} = \boldsymbol{\lambda} \odot (2\mathbf{G}\boldsymbol{\lambda} - 2\Phi^\top\boldsymbol{x} + \boldsymbol{d})}
\tag{64}
$$

The term $\boldsymbol{\lambda} \odot (\dots)$ acts as a multiplicative gate. If a unit is inactive ($\lambda_k \approx 0$), its gradient vanishes, leading to the "dead neuron" problem.

**Hessian derivation.** Let $\mathbf{v} = \nabla_{\boldsymbol{\lambda}}\mathcal{L} = 2\mathbf{G}\boldsymbol{\lambda} - 2\Phi^\top\boldsymbol{x} + \boldsymbol{d}$. Then the gradient can be written as $\boldsymbol{g} = \boldsymbol{\lambda} \odot \mathbf{v}$. To find the Hessian $\mathbf{H} = \nabla_{\boldsymbol{u}}^2\mathcal{L}$, we differentiate $\boldsymbol{g}$ with respect to $\boldsymbol{u}$ using the product rule:

$$
\mathbf{H} = \frac{\partial}{\partial\boldsymbol{u}}(\boldsymbol{\lambda} \odot \mathbf{v}) = \mathrm{diag}(\mathbf{v})\frac{\partial\boldsymbol{\lambda}}{\partial\boldsymbol{u}} + \mathrm{diag}(\boldsymbol{\lambda})\frac{\partial\mathbf{v}}{\partial\boldsymbol{u}}
\tag{65}
$$

Analyzing the first term:

$$
\mathrm{diag}(\mathbf{v})\mathrm{diag}(\boldsymbol{\lambda}) = \mathrm{diag}(\boldsymbol{\lambda} \odot \mathbf{v}) = \mathrm{diag}(\boldsymbol{g})
\tag{66}
$$

Analyzing the second term:

$$
\begin{aligned}
\frac{\partial\mathbf{v}}{\partial\boldsymbol{u}} &= \frac{\partial\mathbf{v}}{\partial\boldsymbol{\lambda}}\frac{\partial\boldsymbol{\lambda}}{\partial\boldsymbol{u}} \\
&= (2\mathbf{G})\mathrm{diag}(\boldsymbol{\lambda})
\end{aligned}
\tag{67}
$$

Therefore, defining $\boldsymbol{\Lambda} := \mathrm{diag}(\boldsymbol{\lambda})$, the second term becomes:

$$
\mathrm{diag}(\boldsymbol{\lambda})(2\mathbf{G})\mathrm{diag}(\boldsymbol{\lambda}) = 2\boldsymbol{\Lambda}\mathbf{G}\boldsymbol{\Lambda}.
\tag{68}
$$

The exact Hessian is:

$$
\boxed{\mathbf{H} = \mathrm{diag}(\boldsymbol{g}) + 2\boldsymbol{\Lambda}(\Phi^\top\Phi)\boldsymbol{\Lambda}}
\tag{69}
$$

where the gradient $\boldsymbol{g}$ is given in Eq. (64). This expression reveals that the curvature is dominated by the correlation matrix of the features ($\Phi^\top\Phi$), scaled quadratically by the firing rates.

**Conclusion.** We have derived the exact analytical gradient and Hessian for the Poisson VAE loss function. These derivations allow us to bypass noisy numerical differentiation.

## F. Statistical analysis of stochastic gradients

In stochastic optimization, we do not have access to the true gradient $\boldsymbol{g}^*$. Instead, we use an estimator $\hat{\boldsymbol{g}}$ such that:

$$
\hat{\boldsymbol{g}} = \boldsymbol{g}^* + \boldsymbol{b} + \boldsymbol{\epsilon},
\tag{70}
$$

where $\boldsymbol{b} = \mathbb{E}[\hat{\boldsymbol{g}}] - \boldsymbol{g}^*$ is the bias, and $\boldsymbol{\epsilon}$ is zero-mean noise with covariance $\boldsymbol{\Sigma} = \mathbb{E}[\boldsymbol{\epsilon}\boldsymbol{\epsilon}^\top]$.

We wish to define metrics that quantify the quality of this estimator not just by its raw variance, but by its impact on the optimization process. We achieve this by analyzing the expected change in loss after a single gradient step.

## F.1. Second-order Taylor expansion

Let the parameter update rule be $\boldsymbol{u}_{t+1} = \boldsymbol{u}_t - \eta\hat{\boldsymbol{g}}$. We approximate the reconstruction loss (Eq. (59)) at the new location using a second-order Taylor expansion around $\boldsymbol{u}_t$:

$$\mathcal{L}(\boldsymbol{u}_{t+1}) \approx \mathcal{L}(\boldsymbol{u}_t) + (\boldsymbol{u}_{t+1} - \boldsymbol{u}_t)^\top \nabla\mathcal{L}(\boldsymbol{u}_t) + \frac{1}{2}(\boldsymbol{u}_{t+1} - \boldsymbol{u}_t)^\top \mathbf{H}(\boldsymbol{u}_{t+1} - \boldsymbol{u}_t). \tag{71}$$

Substituting the update rule:

$$\mathcal{L}(\boldsymbol{u}_{t+1}) \approx \mathcal{L}(\boldsymbol{u}_t) - \eta\hat{\boldsymbol{g}}^\top \boldsymbol{g}^* + \frac{\eta^2}{2}\hat{\boldsymbol{g}}^\top \mathbf{H}\hat{\boldsymbol{g}}. \tag{72}$$

Taking the expectation over the randomness of the estimator $\hat{\boldsymbol{g}}$:

$$\mathbb{E}[\mathcal{L}_{t+1}] - \mathcal{L}_t \approx -\eta\,\mathbb{E}[\hat{\boldsymbol{g}}]^\top \boldsymbol{g}^* + \frac{\eta^2}{2}\,\mathbb{E}[\hat{\boldsymbol{g}}^\top \mathbf{H}\hat{\boldsymbol{g}}]. \tag{73}$$

The left-hand side, $\mathbb{E}[\mathcal{L}_{t+1}] - \mathcal{L}_t$, quantifies the expected one-step change in loss induced by an update driven by the stochastic gradient $\hat{\boldsymbol{g}}$. From the perspective of optimization, we want this quantity to be as negative as possible, which is why it serves as a central object for evaluating the quality of a gradient estimator.

## F.2. Decomposing the quadratic penalty

We focus on the quadratic term $\mathbb{E}[\hat{\boldsymbol{g}}^\top \mathbf{H}\hat{\boldsymbol{g}}]$. Substituting $\hat{\boldsymbol{g}} = (\boldsymbol{g}^* + \boldsymbol{b}) + \boldsymbol{\epsilon}$:

$$\begin{aligned}
\mathbb{E}[\hat{\boldsymbol{g}}^\top \mathbf{H}\hat{\boldsymbol{g}}] &= \mathbb{E}\left[((\boldsymbol{g}^* + \boldsymbol{b}) + \boldsymbol{\epsilon})^\top \mathbf{H}((\boldsymbol{g}^* + \boldsymbol{b}) + \boldsymbol{\epsilon})\right] \\
&= (\boldsymbol{g}^* + \boldsymbol{b})^\top \mathbf{H}(\boldsymbol{g}^* + \boldsymbol{b}) + 2(\boldsymbol{g}^* + \boldsymbol{b})^\top \mathbf{H}\underbrace{\mathbb{E}[\boldsymbol{\epsilon}]}_{0} + \mathbb{E}[\boldsymbol{\epsilon}^\top \mathbf{H}\boldsymbol{\epsilon}]
\end{aligned} \tag{74}$$

The term $\boldsymbol{\epsilon}^\top \mathbf{H}\boldsymbol{\epsilon}$ is a scalar, so it is equal to its trace. Using the cyclic property of the trace ($\mathrm{Tr}(ABC) = \mathrm{Tr}(BCA)$) and linearity of expectation:

$$\begin{aligned}
\mathbb{E}[\boldsymbol{\epsilon}^\top \mathbf{H}\boldsymbol{\epsilon}] &= \mathbb{E}[\mathrm{Tr}(\boldsymbol{\epsilon}^\top \mathbf{H}\boldsymbol{\epsilon})] \\
&= \mathbb{E}[\mathrm{Tr}(\mathbf{H}\boldsymbol{\epsilon}\boldsymbol{\epsilon}^\top)] \\
&= \mathrm{Tr}(\mathbf{H}\,\mathbb{E}[\boldsymbol{\epsilon}\boldsymbol{\epsilon}^\top]) \\
&= \mathrm{Tr}(\mathbf{H}\boldsymbol{\Sigma})
\end{aligned} \tag{75}$$

## F.3. Definition of energy metrics

Grouping the terms from the expected loss change, we identify two distinct "energy" terms that penalize the optimization based on the local curvature $\mathbf{H}$.

**BiasEnergy.** The quadratic cost of the systematic error is:

$$\boxed{\text{BiasEnergy} = \boldsymbol{b}^\top \mathbf{H}\boldsymbol{b}} \tag{76}$$

This measures whether the gradient bias points in a direction of high curvature. A biased estimator is acceptable if the bias lies in the null space of the Hessian (flat directions), but catastrophic if it aligns with steep eigenvectors.

**NoiseEnergy.** The quadratic cost of the variance is:

$$\boxed{\text{BiasEnergy} = \mathrm{Tr}(\mathbf{H}\boldsymbol{\Sigma})} \tag{77}$$

This measures the interaction between noise covariance and curvature. High variance is permissible in flat directions, but noise in stiff directions (high eigenvalues of $\mathbf{H}$) will cause the optimizer to bounce between valley walls, increasing the expected loss.

## F.4. The optimization equation of motion

By combining the linear approximation and the quadratic penalty terms derived above, we arrive at the final expression for the expected change in loss after a single optimization step. This equation serves as the "equation of motion" for the stochastic gradient descent process:

$$
\mathbb{E}[\mathcal{L}_{t+1}] - \mathcal{L}_t \approx \underbrace{-\eta\|\boldsymbol{g}^*\|^2}_{①} + \underbrace{-\eta\boldsymbol{b}^\top\boldsymbol{g}^*}_{②} +
$$

$$
\underbrace{\frac{\eta^2}{2}\boldsymbol{g}^{*\top}\mathbf{H}\boldsymbol{g}^*}_{③} + \underbrace{\eta^2\boldsymbol{g}^{*\top}\mathbf{H}\boldsymbol{b}}_{④} +
$$

$$
\underbrace{\frac{\eta^2}{2}\boldsymbol{b}^\top\mathbf{H}\boldsymbol{b}}_{⑤} + \underbrace{\frac{\eta^2}{2}\operatorname{Tr}(\mathbf{H}\boldsymbol{\Sigma})}_{⑥}.
$$

(78)

Since our goal is to minimize loss, we desire this total quantity to be as negative as possible. We interpret each term below:

① **The Primary Descent** ($-\eta\|\boldsymbol{g}^*\|^2$): This is the main driver of optimization. It represents the reduction in loss achieved by moving purely in the direction of the steepest descent. It is always negative (beneficial), scaling linearly with the learning rate.

② **Directional Alignment** ($-\eta\boldsymbol{b}^\top\boldsymbol{g}^*$): This term measures whether the systematic bias of our estimator works with us or against us. If $\boldsymbol{b}^\top\boldsymbol{g}^* > 0$ (the bias aligns with the true gradient), this term reduces loss further. If $\boldsymbol{b}^\top\boldsymbol{g}^* < 0$ (the bias points uphill), this term fights the descent. If the bias is strong enough, it can completely negate the Primary Descent, causing divergence.

③ **The Curvature Speed Limit** ($+\frac{\eta^2}{2}\boldsymbol{g}^{*\top}\mathbf{H}\boldsymbol{g}^*$): This is the penalty for moving too fast in a steep valley. Even with a perfect gradient, if the curvature $\mathbf{H}$ is large along the direction of travel, taking a large step $\eta$ will overshoot the minimum and land on the opposite wall, increasing the loss. This term dictates the maximum stable learning rate.

④ **Drift-Curvature Coupling** ($+\eta^2\boldsymbol{g}^{*\top}\mathbf{H}\boldsymbol{b}$): This is a second-order interaction term. It represents the complexity of moving through a curved landscape while being pushed off-course by bias. It quantifies how the steepness of the terrain amplifies the misalignment errors.

⑤ **BiasEnergy** ($+\frac{\eta^2}{2}\boldsymbol{b}^\top\mathbf{H}\boldsymbol{b}$): This is the "static cost" of being wrong. Even if the true gradient were zero (we are at a stationary point), a non-zero bias would constantly push the parameters up the walls of the loss landscape. In high-curvature directions (large eigenvalues of $\mathbf{H}$), even a tiny bias incurs a massive penalty here.

⑥ **NoiseEnergy** ($+\frac{\eta^2}{2}\operatorname{Tr}(\mathbf{H}\boldsymbol{\Sigma})$): This is the "thermal cost" of variance. In a convex valley, random jittering does not average out to zero cost; it effectively heats up the system. Because the valley walls curve upward, random steps to the left and right result in a net increase in average height (loss). This term sets the fundamental *noise floor* of the optimization, setting the limit of how close to the optimum we can get with a fixed learning rate.

**Conclusion.** By analyzing the second-order Taylor expansion of the expected loss, we identified BiasEnergy (Eq. (76)) and NoiseEnergy (Eq. (77)) as the theoretically justified metrics for evaluating stochastic gradient estimators. These metrics properly weight gradient errors by the geometry of the optimization landscape, offering a more nuanced view than simple Mean Squared Error.

# G. Methods: architecture, dataset, training, and hyperparameter details

Here we provide additional details about our experiments. We trained a total of 96 $\mathcal{P}$-VAE models and 528 POGLM models in our sweep. The section is organized as follows:

- Section G.1 describes the $\mathcal{P}$-VAE architecture, dataset, training, and hyperparameter details,

- Section G.2 describes those details for the POGLM,

- Section G.3 describes how we adaptively set the truncating upperbound $M$, required for both EAT and GSM,

- Section G.4 provides methodological details for the stochastic gradient analyses.

## G.1. Poisson variational autoencoder ($\mathcal{P}$-VAE) methods

We conducted a comprehensive evaluation of the Poisson Variational Autoencoder ($\mathcal{P}$-VAE, Vafaii et al. (2024)) using multiple gradient estimation methods and hyperparameter configurations. This section details the experimental setup, model architecture, training procedures, and evaluation protocols.

**Experimental design.** Our experiments evaluated five distinct gradient estimation methods for training the $\mathcal{P}$-VAE:

- **EAT$_{\text{sigmoid}}$**: Monte Carlo (MC) estimation with sigmoid-based indicator approximation (section B).

- **EAT$_{\text{cubic}}$**: MC estimation with cubic-based indicator approximation (Eq. (43)).

- **GSM**: MC estimation with adaptive Gumbel-Softmax relaxation (section C).

- **Exact**: Exact gradient computation enabled by the linear decoder assumption (section E).

- **Score**: Score function gradient estimator (REINFORCE, Williams (1992)).

All relaxation methods (EAT$_{\text{sigmoid}}$, EAT$_{\text{cubic}}$, GSM) require providing a temperature hyperparameter $\tau$. For these methods, we varied $\tau \in \{0.02, 0.05, 0.1, 0.2, 0.5\}$. We also experimented with annealing the temperature from a high value $\tau_{\text{start}}$ at the beginning of training to a final value $\tau$, same as above (so we had conditions anneal = True/False). The Exact and Score methods are temperature-independent by design, so their results vary only by seed. In total, we trained 32 unique configurations with 3 seeds each, yielding 96 total models.

**Dataset.** We used whitened van Hateren natural image patches following the preprocessing protocol from the original $\mathcal{P}$-VAE paper (Vafaii et al., 2024). The dataset consists of $16 \times 16$ pixel grayscale image patches extracted from the van Hateren natural image scenes dataset (Van Hateren & van der Schaaf, 1998). The preprocessing pipeline includes:

1. Random extraction of $16 \times 16$ patches from natural images

2. Whitening filter: $R(f) = f \cdot \exp((f/f_0)^n)$ with $f_0 = 0.5$ and $n = 4$

3. Local contrast normalization with kernel size 13 and $\sigma = 0.5$

4. Z-score normalization across spatial dimensions

This preprocessing ensures that the input statistics approximate natural scene statistics while removing low-frequency correlations, encouraging the model to learn sparse, localized features. For additional details on the dataset preparation and statistics, we refer readers to the original $\mathcal{P}$-VAE paper (Vafaii et al., 2024).

**Model architecture.** In this paper, we only trained linear $\mathcal{P}$-VAE models:

**Encoder.**

- Input $\boldsymbol{x}$: $16 \times 16 = 256$ dimensional flattened image patches

- Single linear layer: $\mathbf{W}_{\mathrm{enc}} \in \mathbb{R}^{512 \times 256}$

- Output: $512$ dimensional Poisson log-rates $\boldsymbol{u} = \mathbf{W}_{\mathrm{enc}}\boldsymbol{x}$

- No bias term, no nonlinearity

**Decoder.**

- Input $\boldsymbol{z}$: $512$ discrete latent variables $\boldsymbol{z} \sim \mathcal{P}\mathrm{ois}(\boldsymbol{z}; \exp(\boldsymbol{u}))$

- Single linear layer: $\boldsymbol{\Phi} \in \mathbb{R}^{256 \times 512}$

- Output: $256$ dimensional reconstruction $\hat{\boldsymbol{x}} = \boldsymbol{\Phi}\boldsymbol{z}$

- No bias term, no nonlinearity

**Prior distribution.** The prior over latent codes is a learnable parameter vector, initialized as a uniform distribution in log-space, which corresponds to a scale-invariant Jeffrey's prior. This choice encourages sparsity in the latent representations while remaining agnostic to the absolute scale of neural activity.

**Training configuration.**

- **Optimizer**: Adamax

- **Learning rate**: $\eta = 0.005$

- **Batch size**: 1000 samples per batch

- **Epochs**: 3000 training epochs + 5 warmup epochs

- **Weight decay**: 0.0 (no explicit L2 regularization)

- **Gradient clipping**: Maximum gradient norm of 500

- **Learning rate schedule**: Cosine annealing over all training epochs

- **Upperbound** $M$: Selected adaptively based on maximum Poisson rate during training (see section G.3 below)

**Temperature Annealing.** The temperature $\tau$ is a crucial hyperparameter that controls the sharpness of the continuous relaxation used in gradient estimation. Higher temperatures ($\tau$ close to 1) provide smoother, more diffuse gradients, while lower temperatures ($\tau$ close to 0) approximate true discrete sampling but with higher gradient variance.

To ensure stability during training, we annealed the temperature with the following schedule:

- **Initial temperature**: $\tau_{\mathrm{start}} = 1.0$ (always)

- **Final temperature**: $\tau_{\mathrm{stop}} \equiv \tau \in \{0.02, 0.05, 0.1, 0.2, 0.5\}$ (swept parameter)

- **Annealing type**: Linear decay

- **Annealing duration**: First 1500 epochs (50% of training)

- **Constant phase**: Remaining 1500 epochs at $\tau_{\mathrm{stop}}$

We also experimented with temperature annealing turned off, but did not find substantial differences in results.

**Evaluation protocol.**    All models, regardless of their training temperature or annealing schedule, are evaluated at temperature $\tau = 0$ (true Poisson) to ensure fair comparison. This evaluation protocol is critical because temperature acts as a smoothing parameter during training but should not influence the final discrete latent representations used for inference.

Evaluating at $\tau = 0$ corresponds to using true discrete Poisson sampling without any continuous relaxation:

$$\boldsymbol{z} \sim \mathcal{P}\mathrm{ois}(\boldsymbol{z}; \boldsymbol{\lambda}), \quad \boldsymbol{\lambda} = \exp(\mathbf{W}_{\mathrm{enc}}\boldsymbol{x}).$$

At $\tau = 0$, the gradient estimators converge to their discrete counterparts, eliminating the bias introduced by continuous relaxations. This ensures that our evaluation metrics (ELBO, reconstruction quality, sparsity) reflect the true performance of the learned discrete latent representation rather than artifacts of the training procedure.

**Evaluation metrics.**    For each model, we compute the following metrics on the validation set:

- **ELBO**: the primary objective (Eq. (2)).

- $R^2$ **score**: Coefficient of determination measuring reconstruction quality.

- **Proportion zeros**: Fraction of zero-valued latent activations, measuring sparsity.

- **Sparse coding performance**: Combined metric quantifying the joint quality of reconstruction and sparsity.

**Compute details.**    All experiments were conducted on NVIDIA RTX 6000 Ada Generation GPUs without mixed-precision training to ensure numerical stability. Training a single model takes approximately 2-3 hours. The full experimental sweep required training 96 models in total, which was parallelized across multiple GPUs using a custom sweep runner that manages GPU memory allocation and job scheduling.

**Score-based baselines.**    Our baseline score function estimator operated only on the reconstruction term of the ELBO objective ($\mathcal{L}$), since the KL divergence term took on a closed form and could thus be differentiated deterministically. For variance reduction, we used a baseline consisting of an exponential moving average (with momentum $\alpha = 0.9$) over the average reconstruction error with respect to Monte Carlo samples $k \in [1, K]$ and batch items $b \in [1, B]$. The resulting estimator at timestep $t$ consisted of a baseline reconstruction error times the score function of the variational distribution

$$\bar{\ell}_t := (1 - \alpha)\bar{\ell}_{t-1} + \alpha \frac{1}{B}\sum_{b=1}^{B}\frac{1}{K}\sum_{k=1}^{K}\ln p(\boldsymbol{x}|\boldsymbol{z}; \boldsymbol{\theta})$$

$$\nabla_{\mathbf{W}_{\mathrm{enc}}} \approx \frac{1}{B}\sum_{b=1}^{B}\frac{1}{K}\sum_{k=1}^{K}\left[\ln p(\boldsymbol{x}|\boldsymbol{z}; \boldsymbol{\theta}) - \bar{\ell}_t\right] \cdot \nabla_{\mathbf{W}_{\mathrm{enc}}}\ln q(\boldsymbol{z}|\mathbf{W}_{\mathrm{enc}}\boldsymbol{x}) + \beta \cdot \nabla_{\mathbf{W}_{\mathrm{enc}}}D_{\mathrm{KL}}[q(\boldsymbol{z}|\mathbf{W}_{\mathrm{enc}}\boldsymbol{x})\|p(\boldsymbol{z})].$$

The above contrasts with the standard score function gradient estimator in variational autoencoders, which takes the form

$$\nabla_{\mathbf{W}_{\mathrm{enc}}} \approx \frac{1}{B}\sum_{b=1}^{B}\frac{1}{K}\sum_{k=1}^{K}\left[\ln \frac{p(\boldsymbol{x}, \boldsymbol{z}; \boldsymbol{\theta})}{q(\boldsymbol{z}|\mathbf{W}_{\mathrm{enc}}\boldsymbol{x};)}\right] \cdot \nabla_{\mathbf{W}_{\mathrm{enc}}}\ln q(\boldsymbol{z}|\mathbf{W}_{\mathrm{enc}}\boldsymbol{x}).$$

### G.2. Partially observed generalized linear model (POGLM) methods

We evaluate the proposed relaxation methods on a Partially Observable Generalized Linear Model (POGLM) with an adaptive upperbound, following Li et al. (2024a). The POGLM setting provides a controlled benchmark where ground-truth parameters are known, allowing us to directly attribute performance differences to gradient estimation quality rather than confounding factors.

**Synthetic dataset.**    We generate synthetic spike train data from 10 distinct parameter sets. For each set, the connection weights $w_{n \leftarrow n'}$ and biases $b_n$ are sampled i.i.d. from $\mathrm{Uniform}(-2, 2)$ and $\mathrm{Uniform}(-0.5, 0.5)$, respectively. The network consists of $N = 5$ neurons: 3 visible and 2 hidden. For each parameter set, we generate 40 non-overlapping spike sequences for training and 20 for testing, with each sequence spanning $T = 100$ time bins.

**Model architecture.** The POGLM consists of a variational encoder and a generative decoder:

- **Encoder:** The input (visible neuron spike trains) is passed through a 1D convolutional layer ($C_{\text{in}} = 3$, $C_{\text{out}} = 2$, kernel size $k = 7$, padding $p = 3$) to predict the variational posterior parameters for the 2 hidden neurons. We employ an adaptive upperbound strategy to constrain the support of the distribution. For GSM and score function methods, the output is mapped to logit probabilities via a linear transformation. For $\text{EAT}_{\text{sigmoid}}$ and $\text{EAT}_{\text{cubic}}$, the firing rates are computed directly via a forward pass through the POGLM. All outputs are processed through a softplus activation with numerical clipping.

- **Decoder:** The input consists of spike trains from all 5 neurons (visible and hidden). This is processed by a 1D convolutional layer ($C_{\text{in}} = 5$, $C_{\text{out}} = 5$, kernel size $k = 3$) to predict firing rates for all neurons. As with the encoder, we apply softplus activation and numerical clipping.

Both encoder and decoder are implemented in PyTorch Lightning.

**Training details.** To ensure fair comparison, we maintain identical training settings across all methods:

- **Optimizer:** AdamW with learning rate $\text{lr} = 1 \times 10^{-3}$

- **Epochs:** 100

- **Initialization:** Weights and biases drawn from $\mathcal{N}(0, 0.5^2)$

- **Monte Carlo samples:** $N_{\text{mc}} \in \{1, 2, 5, 10\}$

- **Temperatures:** $\tau \in \{0.02, 0.05, 0.1, 0.15, 0.2, 0.25, 0.3, 0.35, 0.4, 0.45, 0.5\}$

- **Random seeds:** 3 per configuration for statistical reliability

In total, this yields 176 model configurations $\times$ 3 seeds = 528 independent training runs. Experiments were conducted on high-performance CPU nodes (dual Intel Xeon Gold 6226 @ 2.7GHz, 192 GB RAM), completing in approximately one day.

### G.3. Adaptive selection of the upperbound hyperparameter $M$

Both EAT and GSM methods require providing a hyperparameter $M$, corresponding to the number of exponential samples in EAT (Algorithm 1), and a truncation upperbound in GSM (Algorithm 2).

Rather than setting a globally large enough upperbound for accurate distribution approximation, we can dynamically set the upperbound for each data batch. Specifically, given a significance level $\alpha \in (0, 1)$, we seek the upperbound:

$$M = \lceil \text{invCDF}_{\max(\boldsymbol{\lambda})}(1 - \alpha) \rceil, \tag{79}$$

where $\text{invCDF}_{\max(\boldsymbol{\lambda})}$ is the inverse cumulative distribution function of the Poisson distribution whose rate parameter $\lambda$ is the maximum of the batch, since the largest $\lambda$ requires the highest upperbound.

Throughout our experiments, we set $\alpha = 10^{-3}$.

### G.4. Gradient analysis: additional methodological details

To rigorously quantify the quality of gradient estimators, we isolated the gradient estimation step from the optimization loop. This allowed us to compare estimators against a ground truth without the confounding factors of optimization trajectory or changing model parameters.

**Experimental setup.** We extracted the decoder weights $\boldsymbol{\Phi}$ from a fully trained linear $\mathcal{P}$-VAE model. We selected a fixed batch of $B = 90$ input samples $\boldsymbol{x}$ from the validation set (whitened natural image patches). Rather than using the predicted rates from the encoder, we manually fixed the latent firing rates to constant values, sweeping $\lambda \in \{0.1, 0.5, 1, 2, 5, 8, 10, 15, 20, 30, 40, 70, 100\}$, across all $K = 512$ dimensions and all batch items. This controlled setting allows us to study estimator performance as a function of the firing rate regime, isolated from encoder variations.

**Ground truth computation.**    Because the decoder is linear, the reconstruction loss and its derivatives are analytically tractable. For every batch item, we computed the exact gradient $\boldsymbol{g}^*$ and the exact Hessian $\mathbf{H}$ with respect to the log-rates $\boldsymbol{u} = \ln \boldsymbol{\lambda}$ using the closed-form derivations in section E (Eqs. (64) and (69)). Note that for a linear decoder, the Hessian is block-diagonal with respect to the batch dimension (samples are independent), allowing efficient per-sample computation.

**Gradient Estimation.**    For each condition (rate $\lambda$ and temperature $\tau$), we drew $N = 100$ Monte Carlo samples to form stochastic gradient estimates $\hat{\boldsymbol{g}}_1, \ldots, \hat{\boldsymbol{g}}_N$.

- **EAT and GSM methods:** Gradients were computed via automatic differentiation.

- **Score function:** We employed a batch-average baseline reduction. The baseline $b_k$ for latent dimension $k$ was calculated as the mean loss across the batch, reducing variance without introducing bias.

**Metric Calculation.**    All metrics were computed *per batch item $i$* and then averaged.

- **Bias and Variance:** The empirical bias $\boldsymbol{b} = \bar{\boldsymbol{g}} - \boldsymbol{g}^*$ and variance $\boldsymbol{\Sigma}$ were computed over the $N$ samples.

- **Cosine Similarity:** We computed the cosine similarity of the expected gradient ($\mathrm{CosMean} = \cos(\bar{\boldsymbol{g}}, \boldsymbol{g}^*)$) and the average cosine similarity of individual samples ($\mathrm{CosSample} = \mathbb{E}[\cos(\hat{\boldsymbol{g}}, \boldsymbol{g}^*)]$).

- **Curvature-Weighted Energy:** As motivated in section F, we projected the bias and noise into the geometry of the loss landscape using the exact Hessian $\mathbf{H}_i$ (per batch item $i$). To make these metrics comparable across different firing rates (where gradients' magnitudes vary naturally), we normalized them by the *Signal Energy* of the true gradient:

$$\mathrm{NormalizedBiasEnergy} = \frac{\boldsymbol{b}^\top \mathbf{H}_i \boldsymbol{b}}{\boldsymbol{g}^{*\top} \mathbf{H}_i \boldsymbol{g}^*}, \quad \mathrm{NormalizedNoiseEnergy} = \frac{\mathrm{Tr}(\mathbf{H}_i \boldsymbol{\Sigma})}{\boldsymbol{g}^{*\top} \mathbf{H}_i \boldsymbol{g}^*}. \tag{80}$$

This normalization provides the interpretable scale used in Fig. 3, where a value of $\sim 1.0$ implies that the error energy is equivalent to the signal energy driving the optimization.

# H. Supplementary results

In this section, we provide additional results, supporting and extending those reported in the main paper:

- Section H.1: Explores the possibility of auto-tuning the temperature. The results indicate that this is highly case-dependent, and therefore, it is difficult to come up with a one-size-fits-all rule.

- Section H.3: Additional figures, exploring a wider range of rates $\lambda$ or temperatures $\tau$, for distributional fidelity, gradient analysis, and $\mathcal{P}$-VAE/ POGLM results.

- Section H.2 Explores the generalizability of GSM in the over/under dispersed regimes.

### H.1. Automatic selection of optimal temperature is case-dependent

In the main paper, we saw that EAT and GSM methods have different sensitivities to the temperature. While EAT$_{\mathsf{cubic}}$ was robust, both EAT$_{\mathsf{sigmoid}}$ and GSM require careful tuning, which is not ideal. Here, we explore whether it is possible to automatically select the temperature hyperparameter for these methods to mitigate this issue.

**Existence of the optimal temperature.**    Similar to the upperbound (section G.3), we hope to find a clever, adaptive scheme for temperature selection. Therefore, we evaluate the relationship between the optimal $\tau$ (that produces the most accurate gradient estimation) and the Poisson rate $\lambda$ for different methods under various target functions.

To isolate the performance of gradient estimators from ELBO optimization, we established a benchmark that treats the gradient estimation for the Poisson rate parameter $\lambda$ as a regression task. Given a family of test functions $\{f_k(z)\}_{k=1}^K$, the objective is to estimate the gradient of the expectation with respect to $\lambda$:

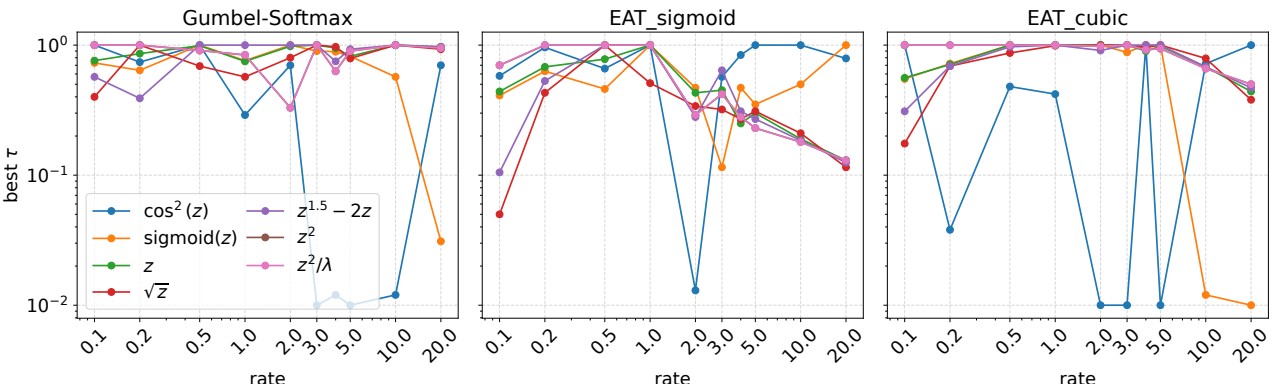

*Figure 9.* **Optimal temperature analysis across relaxation methods.** We compare the optimal temperature $\tau^*$ that minimizes MAE and the input rate $\lambda$. The optimal $\tau$ for all relaxation methods remains within the range $[0.1, 1]$. While many test functions show similar trends, most of them are monotonic. However, specific trends vary by relaxation method. The $\cos^2(z)$ curve (skyblue) is the only periodic function. Its behavior diverges significantly from the others. The function $z^2/\lambda$ (pink) depends explicitly on the rate. It also shows a unique pattern. Its optimal $\tau$ frequently hits the ceiling of the search range across all three relaxation methods. We conducted experiments within an expanded search space ($\tau \in [0.01, 20]$, $\lambda \in [0.01, 50]$). In this setting, this function still has a much larger optimal $\tau$ than others. These observations suggest that the relationship between optimal $\tau$ and the Poisson rate primarily depends on the relaxation method and the specific approximation task. Consequently, there is no uniform fitting for determining an optimal $\tau$ across various target functions.

$$\nabla_\lambda \mathcal{L} = \nabla_\lambda \mathbb{E}_{z \sim \text{Poisson}(\lambda)}[f_k(z)] \tag{81}$$

Here, $\lambda$ is the learnable variable. For an input $\lambda$ and temperature $\tau$, we compute the gradient estimate value $\hat{g}$ using GSM and the EAT Sigmoid/Cubic methods.

We quantify estimation quality by comparing the estimated value $\hat{g}$ with the theoretical ground truth $g_{true}$. In the Poisson distribution, the theoretical gradient of any objective function can be expanded as:

$$\nabla_\lambda \mathcal{L}(\lambda) = \sum_{z=0}^{\infty} f(z) \cdot p(z|\lambda) \cdot \left(\frac{z}{\lambda} - 1\right) \tag{82}$$

To compute this, we use an upperbound for approximation, reducing the infinite to a finite sum $\sum_{z=0}^{K}$. In our experiments, we set this upperbound to $K = \lfloor \lambda \rfloor + 20$, which is sufficient to cover most of the Poisson probability mass. The resulting value is treated as the Exact Gradient $g_{true}$. Finally, we adopt the Mean Absolute Error (MAE) as our evaluation metric.

Experimental Settings: For test functions, we selected seven functions $f(z)$ exhibiting distinct properties (linear, polynomial, non-linear, trigonometric, and parameter-dependent) to evaluate estimator robustness. The polynomials are $z$, $\sqrt{z}$, $z^2$, $z^{1.5} - 2z$. Non-linear/trigonometric functions are $\cos^2(z)$, sigmoid$(z)$. The above functions only depend on samples. The parameter-dependent function $z^2/\lambda$, this function tests whether the estimator correctly back-propagates gradients when the loss function explicitly depends on the optimization parameter $\lambda$.

Hyperparameters: Rate ($\lambda$) is 0.1 to 20 (total 10). Temperature $\tau$: 0.01 to 10 (total 156). Monte Carlo Samples ($N_{mc}$): 1 to 500 (total 9). Adaptive upperbound precision: 0.01.

We tested the gradient estimation error with temperature $\tau$ for each test function. Then we extracted the optimal temperature $\tau^*$ that minimizes MAE and plotted the relationship between $\tau^*$ and the input rate $\lambda$. Figure 9 illustrates the relationship between the optimal temperature $\tau$ on the Poisson rate $\lambda$ across three relaxation methods. Test functions are represented by different colors.

As shown in the figure, we can see that for each test function, the variation of $\tau^*$ with $\lambda$ exhibits a non-linear relationship and shows distinct trends across different functions $f(z)$. For most sample-dependent functions (e.g., $z^2$), the trends of $\tau^*$ share certain similarities. However, for the parameter-dependent function $f(z) = z^2/\lambda$ and the periodic function $f(z) = \cos^2(z)$,

the pattern of $\tau^*$ diverges significantly.

These results indicate that the optimal temperature $\tau^*$ is highly dependent on the specific estimation objective and the structure of the loss function. Consequently, a universal adaptive temperature formula does not exist. This provides the necessity of empirical tuning for temperature $\tau$ in our subsequent applications.

## H.2. Generalizability of GSM in the over/under dispersed regimes

While the main text focuses on the Poisson distribution, the GSM method can be applied to arbitrary discrete distributions supported on $\mathbb{N}_{\geqslant 0}$. In this section, we describe the formulation for applying our relaxation methods to the Negative Binomial (over-dispersed) and COM-Poisson (under-dispersed) distributions.

**Definition of dispersion.** Dispersion quantifies the variability of a count variable $z$ relative to its mean. It is defined as the ratio $F = \text{Var}(z)/\mathbb{E}[z]$, i.e., the Fano factor. Distributions are categorized as:

- **Equal-dispersion** ($F = 1$): Variance equals the mean (e.g., Poisson).

- **Over-dispersion** ($F > 1$): Variance exceeds the mean (e.g., Negative Binomial).

- **Under-dispersion** ($F < 1$): Variance is less than the mean (e.g., Binomial, COM-Poisson).

**Dispersion in the Poisson distribution.** For $z \sim \mathcal{P}\text{ois}(\lambda)$, we have $\mathbb{E}[z] = \lambda$ and $\text{Var}(z) = \lambda$, yielding $D = 1$. While standard, this strictly equal dispersion is often insufficient for modeling neural data, which frequently exhibits time-varying Fano factors distinct from unity (Goris et al., 2014).

**Modeling Over-dispersion: Negative Binomial.** To model over-dispersed data, we utilize the Negative Binomial (NB) distribution, parameterized by shape $r > 0$ and success probability $\mu \in (0, 1)$. The mean is $\mathbb{E}[z] = r(1 - \mu)/\mu$ and the variance is $\text{Var}(z) = r(1 - \mu)/\mu^2$. The dispersion is $F = 1/\mu$. Since $p < 1$, we always have $F > 1$.

**Method 1: Two-step relaxation (Gamma-Poisson Mixture).** The NB distribution can be represented as a Gamma-Poisson mixture (Zhang et al., 2025). If $z \sim \text{NB}(r, \mu)$, it is equivalent to the hierarchical sampling procedure:

$$\lambda \sim \text{Gamma}\left(r, \frac{\mu}{1 - \mu}\right), \quad z \sim \mathcal{P}\text{ois}(\lambda). \tag{83}$$

This representation allows us to apply the EAT estimator by differentiating through the Gamma sample (using standard pathwise derivatives for continuous variables) and then applying EAT to the conditional Poisson step. We summarize this in algorithm 3.

---

**Algorithm 3** Two-step relaxation of the NB distribution (Mixture)

---

**Input:** NB parameters $r, \mu$
**Output:** Relaxed event count $z \in \mathbb{R}_{\geqslant 0}$

$\beta \leftarrow \frac{\mu}{1-\mu}$     {Calculate rate parameter for Gamma}
$\lambda \leftarrow \text{Gamma}(r, \beta).\,\text{rsample}()$     {Sample continuous rate $\lambda$ (reparameterized)}
$z \leftarrow \text{EAT}(\lambda)$     {Sample soft Poisson counts given $\lambda$}

**Return** $z$

---

**Method 2: Direct truncation with GSM.** The GSM method does not require the mixture representation. For any discrete distribution (including NB) with support on $\mathbb{N}_{\geqslant 0}$, we can truncate the distribution at a sufficiently large upper bound $M$ and apply the Gumbel-Softmax relaxation directly to the log-probabilities. This bypasses the intermediate Gamma sampling step.

**Modeling under-dispersion: COM-Poisson.** Standard Poisson processes (and thus the standard EAT algorithm) cannot model under-dispersion ($F < 1$) without modifying the inter-arrival distribution (e.g., to an Erlang distribution), which complicates the inverse-CDF derivation. In contrast, GSM handles under-dispersion natively by defining the probability mass function directly. We utilize the Conway-Maxwell-Poisson (COM-Poisson) distribution, defined by rate $\lambda$ and the inverse dispersion parameter $\nu = \frac{1}{F}$:

$$P(z; \lambda, \nu) = \frac{\lambda^z}{(z!)^\nu Z(\lambda, \nu)}, \tag{84}$$

where $Z(\lambda, \nu) = \sum_{j=0}^{\infty} \frac{\lambda^j}{(j!)^\nu}$ is the partition function. Because GSM is invariant to normalization constants (due to the softmax), we can sample from the COM-Poisson without explicitly computing $Z(\lambda, \nu)$, provided we truncate at $M$. The logits are simply:

$$\ln \pi_m = m \ln \lambda - \nu \ln(m!), \quad \forall m \in \{0, \dots, M-1\}. \tag{85}$$

These logits are passed directly to the Gumbel-Softmax sampler.

## H.3. Additional empirical results supporting and extending the main paper

In this section, we report additional empirical results, organized by the main-text subsection they support: section 3.4 (ELBO performance on $\mathcal{P}$-VAE and POGLM), section 3.2 (distributional fidelity), and section 3.3 (gradient analysis). We begin with the $\mathcal{P}$-VAE and POGLM extensions, since these speak most directly to the main-paper claims, and end with the wider-range distributional and gradient figures, which broaden but do not change the conclusions of the main paper.

### H.3.1. SCALING $\mathcal{P}$-VAE TO $40 \times 40$ IMAGENET PATCHES

To test the estimators at a larger scale, we trained $\mathcal{P}$-VAE on $40 \times 40$ cropped ImageNet patches, roughly $6\times$ more pixels per sample than the $16 \times 16$ patches used in the main paper. At this scale, EAT$_{\text{cubic}}$ is the only relaxation that closes the gap to exact gradients:

- *Exact gradients:* ELBO of $-718.92$ (upperbound).

- EAT$_{cubic}$: ELBO of $-720.27$ at $\tau = 0.02$ (near-exact).

- EAT$_{sigmoid}$: ELBO of $-755.40$ at its best temperature, $\tau = 0.2$.

A gap of $\sim 35$ nats or more persists for EAT$_{\text{sigmoid}}$ across all temperatures we tested. The takeaway is that the larger-scale advantage of EAT$_{\text{cubic}}$ is not recoverable through temperature tuning of the sigmoid variant: the gap is structural, not a hyperparameter artifact.

### H.3.2. GENERALIZABILITY BEYOND POISSON: NEGATIVE BINOMIAL $\mathcal{P}$-VAE

We tested whether our findings transfer to overdispersed regimes by training Negative Binomial $\mathcal{P}$-VAE (NB-$\mathcal{P}$-VAE) on $16 \times 16$ natural image patches with fitted dispersion. For the EAT methods, we used a two-step Gamma-Poisson relaxation (algorithm 3), and swept the temperature $\tau$ from $10^{-6}$ to $0.5$.

The exact-gradient baseline reaches an ELBO of $-169.37$. EAT$_{\text{cubic}}$ maintains near-exact performance across the practical temperature range, with a best ELBO of $-169.39$; EAT$_{\text{sigmoid}}$ degrades at the extremes, dropping to $-177.28$ at $\tau = 0.5$. At the lower bound $\tau = 10^{-6}$, both methods suffer from numerical instability (EAT$_{\text{sigmoid}}$: $-176.19$; EAT$_{\text{cubic}}$: $-186.65$), but both recover to near-exact performance by $\tau = 10^{-3}$. The temperature robustness of EAT$_{\text{cubic}}$ therefore transfers cleanly to the Negative Binomial distribution, suggesting that the main-paper conclusions are not specific to the Poisson case.

### H.3.3. MACAQUE V1 LINEAR READOUT

To evaluate whether $\mathcal{P}$-VAE features transfer to a downstream neural-prediction task, we used $\mathcal{P}$-VAE models pretrained on natural image patches as fixed feature extractors and fit a linear readout to predict macaque V1 responses (Cadena et al., 2019). Following Cadena et al. (2019), we report the proportion of explainable variance (Sahani & Linden, 2002).

Both relaxations outperform the exact-gradient baseline:

- *Exact gradients:* $\sim 0.19$ explainable variance.

- $\text{EAT}_{\textit{sigmoid}}$: $0.24 \pm 0.01$.

- $\text{EAT}_{\textit{cubic}}$: $0.22 \pm 0.01$.

Temperature robustness does not cleanly propagate to the linear readout performance, so this is not a result about $\tau$-sensitivity. The fact that both relaxations beat exact gradients is, however, consistent with the implicit-regularization story of the next section (section H.3.4): the continuous relaxation seems to produce features that generalize better to a downstream task than features learned with the exact gradient.

### H.3.4. LIMITED-DATA REGIME (10,000 TRAINING SAMPLES)

Data-limited settings are common in neuroscience, where collecting natural-stimulus responses is expensive. To probe this regime, we trained $\mathcal{P}$-VAE on a restricted subset of 10,000 natural image patches (compared to the $\sim 107{,}000$ used in the main paper) and evaluated on the standard test set of $\sim 36{,}000$ samples.

In this regime, the exact-gradient baseline overfits: its ELBO improves early in training but then degrades as training proceeds. The EAT estimators, in contrast, remain stable. $\text{EAT}_{\text{cubic}}$ reaches an ELBO of $-170.11$ at $\tau = 0.05$, matching its full-dataset performance. The continuous relaxation appears to act as an implicit regularizer that benefits low-data settings.

### H.3.5. LEARNED DECODER FEATURES

Figure 10 visualizes the learned $\mathcal{P}$-VAE decoder weights $\Phi$ across relaxation methods. Here, $\text{EAT}_{\text{sigmoid}}$ produces the cleanest, most Gabor-like features — visibly closer to the exact-gradient baseline than $\text{EAT}_{\text{cubic}}$. We attribute this to the smoothness of the underlying transition function: the sigmoid is $C^{\infty}$, while the cubic smoothstep is only $C^1$ (Eq. (43)), so the sigmoid yields better-behaved gradients in the regime where decoder weights are being shaped.

This is the one regime in our experiments where $\text{EAT}_{\text{sigmoid}}$ clearly beats $\text{EAT}_{\text{cubic}}$, and it makes the case for keeping the sigmoid variant available when interpretable features matter more than ELBO optimization or temperature robustness.

### H.3.6. EXTENDED POGLM RESULTS

We complement the synthetic POGLM panel of Fig. 4 and the RGC results of Fig. 5 with three additional analyses, covering weight recovery, sensitivity to Monte Carlo sample size, and additional hidden-neuron counts on real data.

Figure 11 reports six evaluation metrics — ELBO, marginal log-likelihood, conditional log-likelihood, hidden log-likelihood, and weight and bias recovery error — across a denser $\tau$ sweep on the synthetic POGLM task. The pattern in Fig. 4 carries over to every metric: $\text{EAT}_{\text{cubic}}$ remains stable across the full range of $\tau$, while $\text{EAT}_{\text{sigmoid}}$ and GSM degrade at higher temperatures. The weight and bias recovery panels are particularly informative: they show that the temperature robustness of $\text{EAT}_{\text{cubic}}$ is not just a property of the ELBO but also of the parameters recovered by the model.

Figure 12 then selects the best $\tau$ per method and plots the resulting performance against the number of Monte Carlo samples used during training. Apart from the score function baseline (which improves with MC size as expected), all pathwise methods are essentially flat: MC sample size does not meaningfully affect performance once $\tau$ is well chosen.

Finally, Fig. 13 reproduces the RGC results of Fig. 5 for $H = 1$ and $H = 2$ hidden neurons. The pattern is consistent with the $H = 3$ case shown in the main text: $\text{EAT}_{\text{cubic}}$ maintains near-optimal performance across the full temperature range, while $\text{EAT}_{\text{sigmoid}}$ and GSM degrade once $\tau > 0.2$. The conclusion from the main paper — that $\text{EAT}_{\text{cubic}}$ is the most reliable choice across temperatures — holds for every hidden-neuron count we tested.

### H.3.7. DISTRIBUTIONAL FIDELITY ACROSS FIRING RATES

The main paper presented distributional fidelity results at three representative rates $\lambda \in \{2, 20, 100\}$. Here, we extend those results to a wider range, $\lambda \in [0.1, 100]$, and verify that the patterns hold throughout. Figure 14 extends Fig. 1 (mean and variance ratios), and Fig. 15 extends Fig. 2 (Wasserstein distances). In both cases, the rankings established in the main paper hold: $\text{EAT}_{\text{cubic}}$ remains the most faithful to true Poisson moments and to the Poisson distribution as a whole across all tested rates and temperatures.

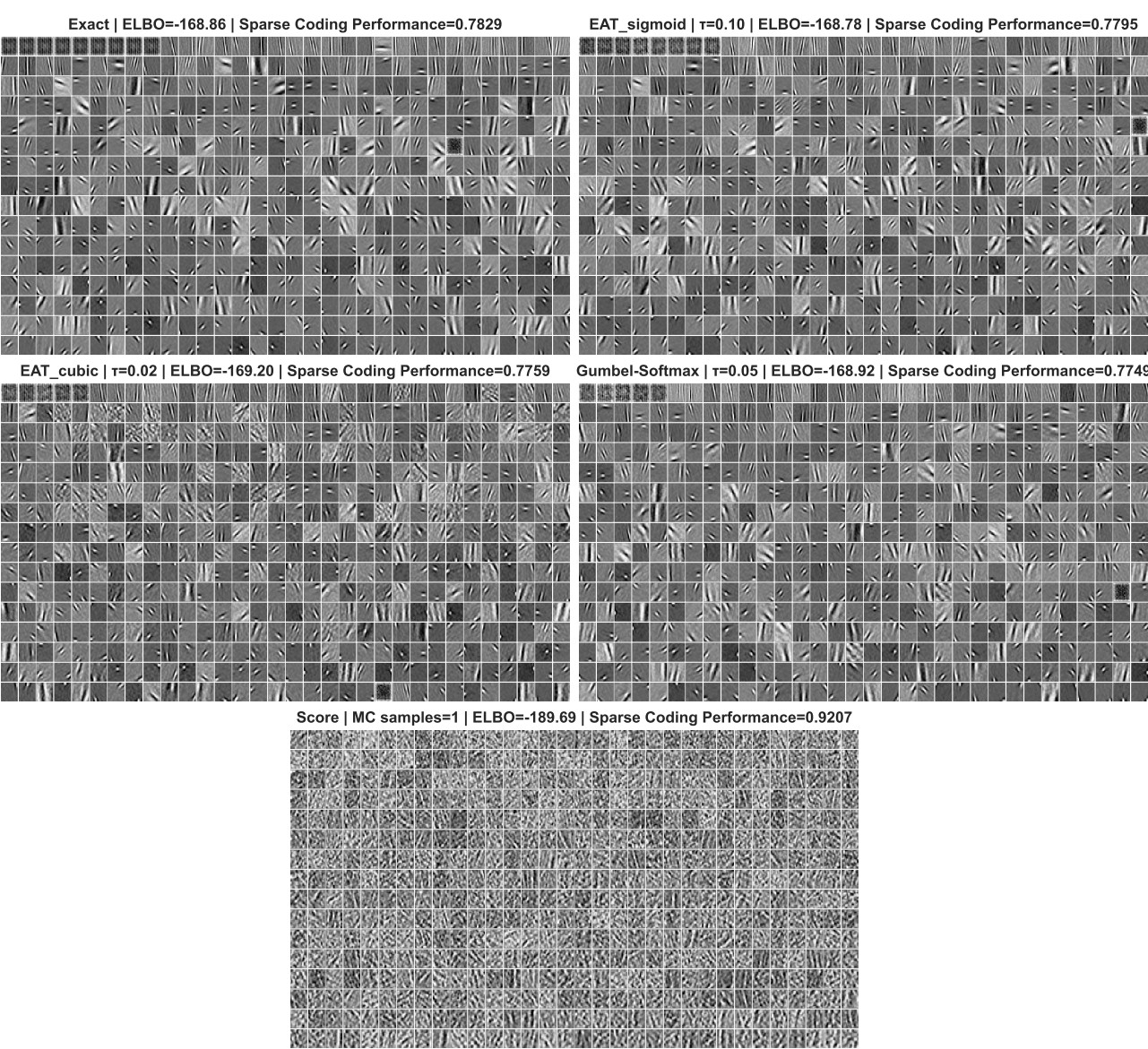

*Figure 10.* $\mathcal{P}$-**VAE learned features.** These show the decoder weights, $\boldsymbol{\Phi} \in \mathbb{R}^{M \times K}$, where $M = 256$ is the input dimension (number of pixels: $16 \times 16 = 256$), and $K = 512$ is the latent dimensionality. We see that EAT$_{\text{sigmoid}}$ learns high-quality features, almost as good as the case where exact gradients are known (top left). GSM also learns good features, but the quality of EAT$_{\text{cubic}}$ features is lower. This is likely due to the fact that sigmoid is $C^{\infty}$, while the cubic smoothstep is $C^1$ (Eq. (43)). Finally, the score function baseline is visibly worse.

### H.3.8. GRADIENT QUALITY ACROSS FIRING RATES

The main paper reported gradient-quality metrics at a single rate $\lambda = 20$ (Fig. 3), using curvature-weighted bias and noise energies. Here, we extend the analysis along two axes: a wider range of rates, and the raw (un-weighted) bias and variance.

Figure 16 extends Fig. 3 across rates $\lambda \in [0.1, 100]$. The qualitative pattern from the main paper holds: both EAT methods achieve near-perfect directional alignment and low bias/noise energy across temperatures, while GSM shows elevated bias at low temperatures and elevated noise throughout.

Figure 17 shows the raw bias and variance of the gradient estimators at $\lambda = 20$, complementing the curvature-weighted metrics in Fig. 3. The raw metrics reproduce the same conclusions: EAT$_{\text{cubic}}$ maintains low bias across temperatures, EAT$_{\text{sigmoid}}$ bias grows with $\tau$, and GSM has very high variance at low $\tau$ that decays toward 1 as $\tau$ grows.

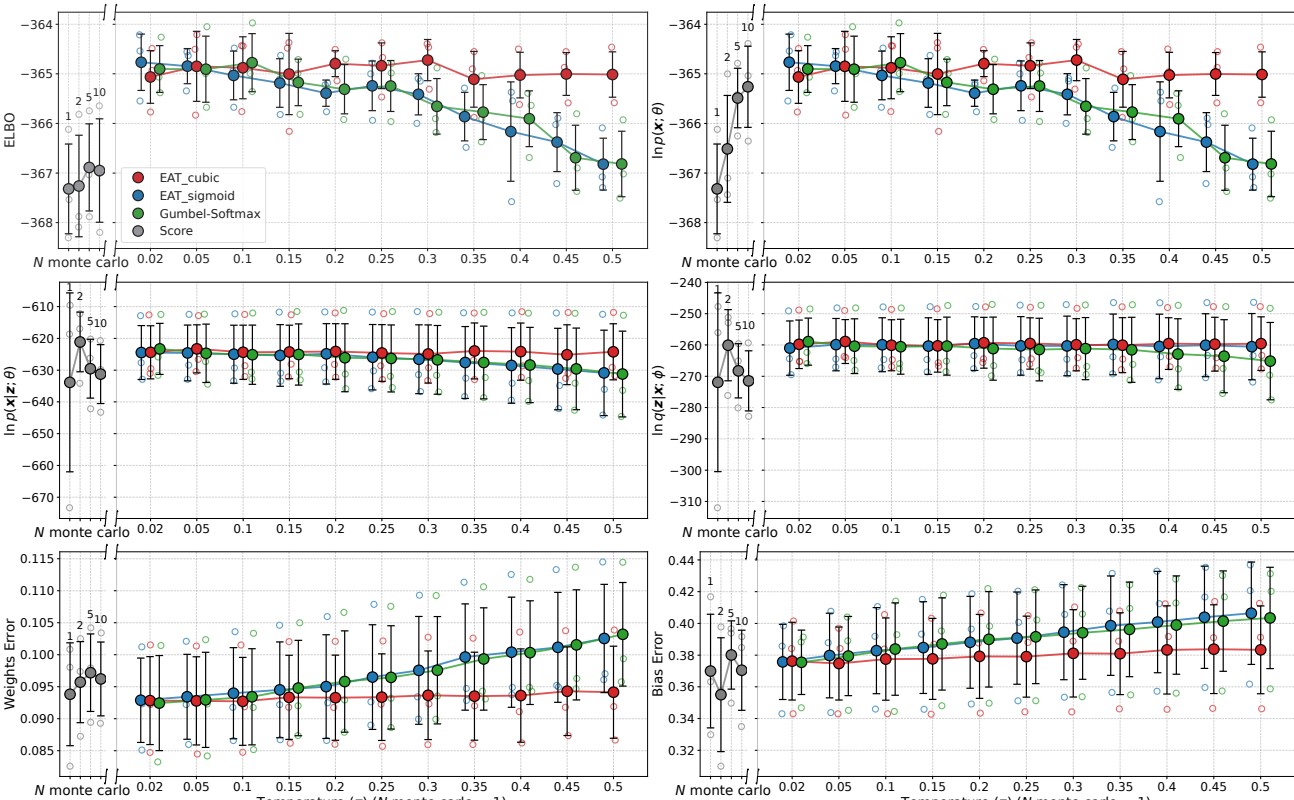

*Figure 11.* **Detailed POGLM results across different relaxation methods.** We use six evaluation metrics to analyze four POGLM architectures (score function, $\text{EAT}_{\text{cubic}}$, $\text{EAT}_{\text{sigmoid}}$, and $\text{GSM}$). The metrics are: *ELBO* (larger is better); *marginal log-likelihood*, MLL, $\ln p(\boldsymbol{x}; \theta)$, means how well the model explains the data (larger is better); *conditional log-likelihood*, CLL, $\ln p(\boldsymbol{x}|\boldsymbol{z}; \theta)$, which reflects the reconstruction ability of the generative model (larger is better); *hidden log-likelihood*, HLL, $\ln q(\boldsymbol{z}|\boldsymbol{x}; \phi)$, which measures how well the inference network fits the latent distribution (larger is better); *weights_error* (smaller is better); and *bias_error* (smaller is better). We separate the score function results from the other three methods and place them on the left side of the figure, since the score function does not include $\tau$. We can observe that as the Monte Carlo sample size increases, the score function method improves accordingly. This matches its mathematical behavior. We also see a marginal effect when the MC size becomes large. For the comparison among the other methods (right side of the figure), the MC sample size is 1. For ELBO, we see that as the temperature $\tau$ increases, the performance of $\text{EAT}_{\text{sigmoid}}$ and $\text{GSM}$ decreases, while $\text{EAT}_{\text{cubic}}$ remains robust. For MLL, the trend is similar to ELBO. For the weights and bias, we observe that the $\text{EAT}_{\text{sigmoid}}$ and $\text{GSM}$ increase with temperature, while $\text{EAT}_{\text{cubic}}$ stays almost unchanged. This again proves the robustness of $\text{EAT}_{\text{cubic}}$.

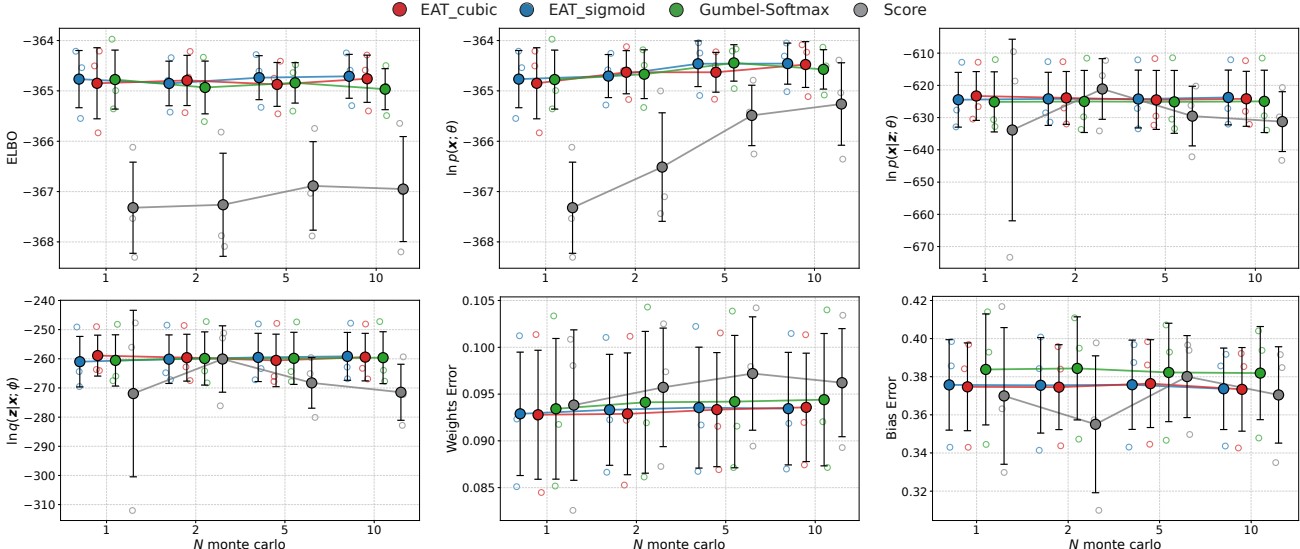

*Figure 12.* **Performance of our relaxation methods under optimal temperature $\tau$ across different Monte Carlo sample sizes.** Based on Fig. 11, we take one more step. For each relaxation method, we select the $\tau$ that gives the best mean ELBO. Then we plot the results under different Monte Carlo sample sizes. There's a clear pattern that, except for the score function method, all other relaxation methods are insensitive to the MC sample size across all evaluation metrics.

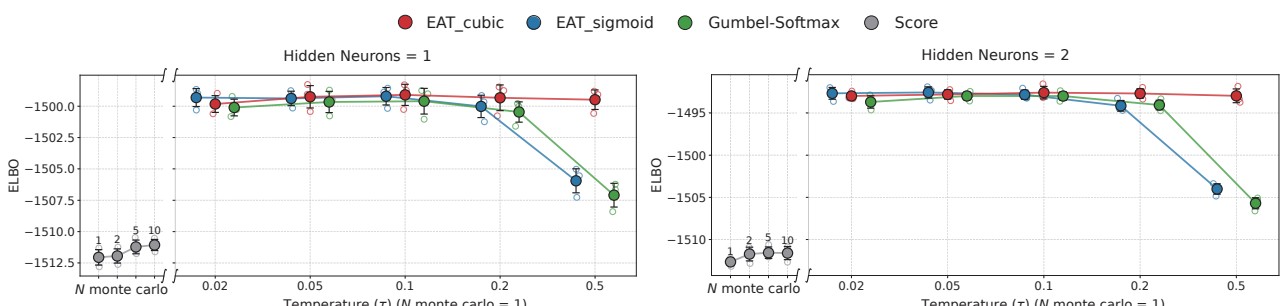

*Figure 13.* **POGLM validation ELBO on the retinal ganglion cells (RGC) dataset, while assuming one and two hidden neurons.** The experimental settings and layout in this figure are the same as those in Fig. 5. The resulting pattern is consistent with Fig. 5: across the full temperature range, EAT$_{\text{cubic}}$ (red) maintains near-optimal performance, while EAT$_{\text{sigmoid}}$ (blue) and GSM (green) begin to degrade once $\tau > 0.2$. EAT$_{\text{cubic}}$ exhibits the most stable performance.

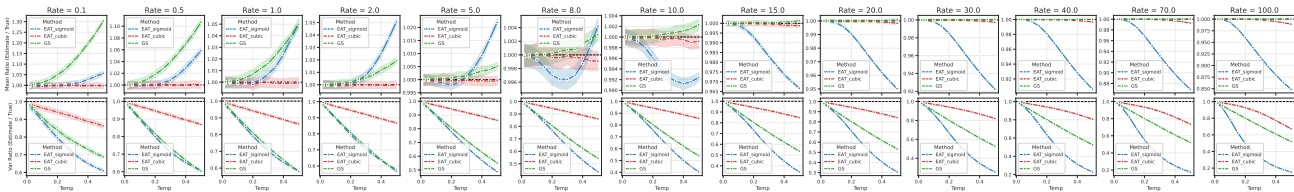

*Figure 14.* **Distributional fidelity across a wide range of firing rates.** Extending analysis of Fig. 1, showing the ratio of empirical moments to true Poisson moments across varying rates $\lambda \in [0.1, 100]$ (columns) and temperatures $\tau \in [0.02. 0.5]$ (x-axis). **Top (Mean):** EAT$_{\text{cubic}}$ (red) maintains an unbiased estimate of the mean across all rates and temperatures. GSM (green) achieves unbiased means only at high rates ($\lambda \gtrsim 20$). EAT$_{\text{sigmoid}}$ (blue) exhibits consistent mean inflation across all conditions, substantially worsening as rates increase. **Bottom (Variance):** EAT$_{\text{cubic}}$ deviates the least from the true variance across all temperatures and rates. EAT$_{\text{sigmoid}}$ shows severe variance collapse. Dashed lines indicate ideal Poisson fidelity (ratio $= 1$).

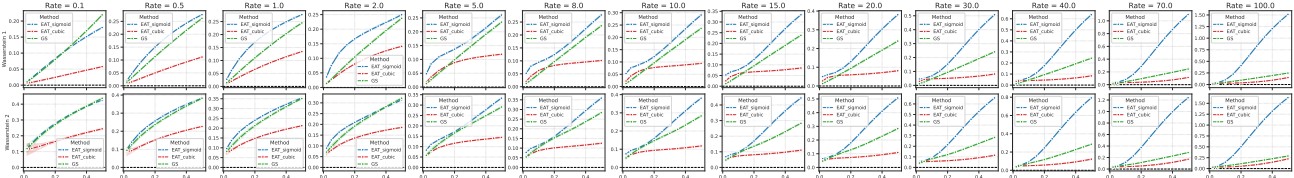

*Figure 15.* **Wasserstein distances across firing rates.** Extending analysis of Fig. 2, comparing Wasserstein-1 ($W_1$) and Wasserstein-2 ($W_2$) distances from the true Poisson distribution. EAT$_{cubic}$ (red) consistently achieves the lowest $W_1$ and $W_2$ distances, remaining relatively stable across all rates and temperatures. In contrast, EAT$_{sigmoid}$ (blue) deviates heavily from both metrics, particularly as the firing rate increases. GSM (green) remains relatively constant across rates but generally performs worse than EAT$_{cubic}$ across temperatures.

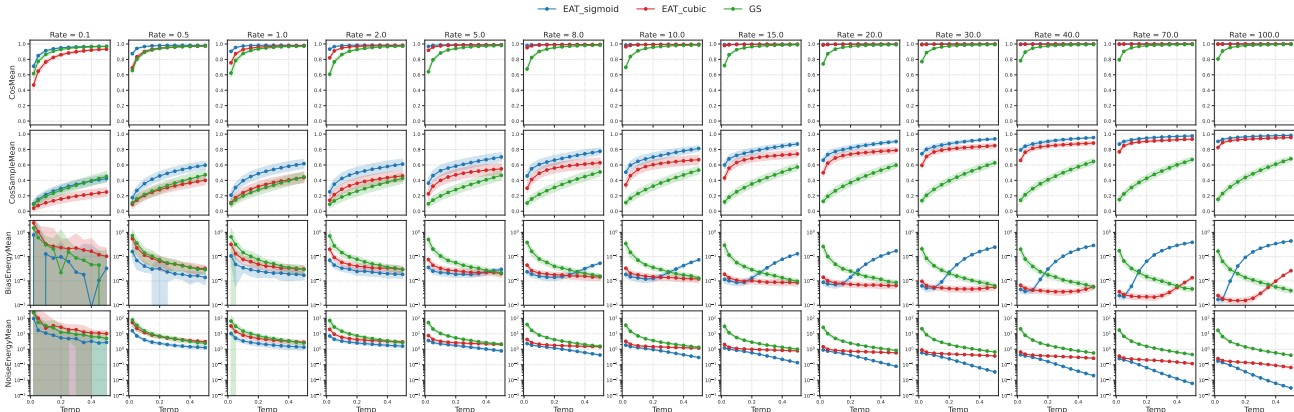

*Figure 16.* **Gradient quality metrics across firing rates.** Extended analysis of Fig. 3. **Row 1** (CosMean): at low temperatures and low rates, all methods deviate from perfect alignment (1.0). At higher rates, all methods except GSM become relatively invariant to temperature. GSM remains lower across all rates at low temperatures. **Row 2** (CosSample): At low rates, all methods show low sample alignment, but this improves with temperature. As the rate increases, EAT$_{sigmoid}$ and EAT$_{cubic}$ approach 1.0, while GSM lags behind. **Row 3** (BiasEnergy): Generally decreases with temperature, except for EAT$_{sigmoid}$ at high rates ($> 8$), where bias energy increases with temperature. **Row 4** (NoiseEnergy): consistently decreases as temperature increases across all rates for all methods. Thus, we see that the raw metrics reproduced the patterns exhibited by the curvature-aware metrics in Fig. 3.

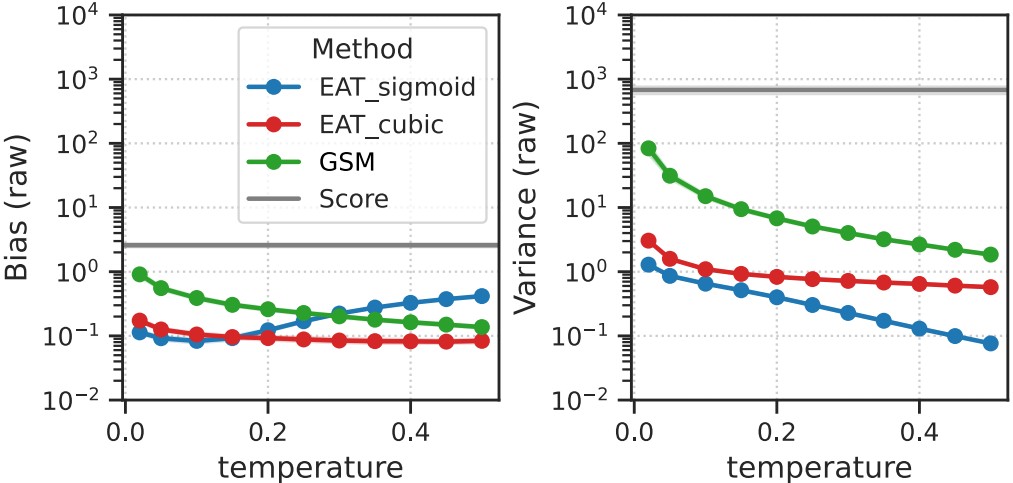

*Figure 17.* **Raw bias and variance of gradient estimators.** In Fig. 3, we visualized the curvature-weighted bias and noise energy metrics, for $\lambda = 20$. Here, we plot the raw quantities. **Left (Bias):** EAT$_{cubic}$ (red) maintains consistently low bias across all temperatures. EAT$_{sigmoid}$ (blue) bias increases with temperature, while GSM (green) bias decreases. **Right (Variance):** For all models, variance decreases as temperature increases. EAT$_{cubic}$ variance is stable (close to 1). GSM variance starts extremely high ($\sim 100$) at low temperatures and approaches 1 at high temperatures. EAT$_{sigmoid}$ starts near 1 but collapses to $\sim 0.1$ at high temperatures.

