# OpenReview forum: "A hitchhiker's guide to Poisson gradient estimation"
_ICML.cc/2026/Conference — ICML 2026 regular_

### Official Review · Reviewer_wBby · 2026-03-08

**Soundness:** 3
**Presentation:** 4
**Significance:** 1
**Originality:** 3
**Overall Recommendation:** 4
**Confidence:** 4

**Summary:**

Inspired by NeuroAI applications, this paper provides an exposition of two methods for estimating gradients in Poisson-based models. Authors then extend one method based on a sensitivity analysis regarding a temperature parameter. The three methods are comprehensively discussed and evaluated through comparisons of distributional fit, gradient quality, and downstream ELBO optimization with respect to temperature.

**Compliance With Llm Reviewing Policy:**

Affirmed.

**Final Justification:**

The authors addressed the limitations with many additional experiments for the rebuttal, most importantly including analysis on real neural data as well as generalization to NB-VAE.

**Key Questions For Authors:**

1. Why did you choose temperature range [0.02, 0.5]? Please sweep over significantly smaller temperatures, e.g. 1e-6 to 1e-2, for an analysis you think would break using small temperatures. This will show the practical lower limit on temperature choice and provide more justification for why selecting temperature is a significant problem.
2. A primary issue in neural datasets is limited size, e.g. at most a couple hundred trials per subject per session. Can you perform an analysis similar to Figure 4 but for increasing dataset sizes realistic to what you would see in neuroscience (e.g., 100, 500, and 1000 trials)? Also, please state the dataset size and dimensionality used for ELBO performance in Sec 3.4. This analysis will help highlight if EAT_cubic offers advantages in estimation quality for small datasets.

**Limitations:**

yes

**Strengths And Weaknesses:**

Strengths: The exposition is excellent, including clear, detailed derivations and explanations of the methods considered. The authors directly tackle the question of choosing between models/hyperparameters for practitioners, which is often sidelined in methods papers. The analysis is thorough and ranges from directly testing their observations from Sec. 3.1 to downstream evaluation, of which the gradient analysis was especially interesting. Code is also included, which is great for reproducibility.

Weaknesses:
- The primary weakness of this paper is its limited scope (comparison of three specific methods, of which two are highly similar, for specific NeuroAI applications under specific model assumptions for one specific parameter). In particular, the singular focus on performance with respect to only temperature is quite limiting. Based on the description of EAT (around line 149-152), it seems obvious for EAT-based methods to choose small values of tau so that the relaxation is more realistic, and reasons/tradeoffs for choosing larger tau are not highlighted in the paper. Overall, it is not clear that the studied phenomenon can be meaningfully generalized into other models or domains.
- The authors lean heavily on neuroscientific facts to motivate the analysis of these methods. However, the authors only discuss results in the context of Poisson variables and homogeneous Poisson processes. While analysis of inhomogeneous processes is more challenging, bursty and inhomogenous spiking behavior is also a well-known neuroscientific fact. Thus, exclusion of this phenomenon, even a brief discussion, seems unusual given the emphasis on neuroscientific accuracy.
- Relatedly, although the authors are highly motivated by neural applications, they do not include any analysis of real spiking data, of which there are many open-source datasets. While I acknowledge and appreciate the rigor of quantitative metrics such as ELBO and distribution similarity, I understand the ultimate intention of EAT_cubic is to be applied to neural data, so such an analysis should also be included.
- The plots in Appendix H.2 are too small. Consider increasing number of rows for better visibility.
- General note that the proposed method (EAT_cubic) should be the first entry in the legend.

---

> ### Author Rebuttal · Authors · 2026-03-31
>
> We thank Reviewer `wBby` for the thorough evaluation and for recognizing the "excellent exposition" and "especially interesting" gradient analysis. The concerns about scope and neural data are fair, and we address them with substantial new results.
>
> ## **Scope: generalization beyond Poisson**
>
> > The primary weakness of this paper is its limited scope...
>
> We ran new experiments that directly address this.
>
> **1. Negative Binomial VAE.** We trained NB-VAEs on 16×16 natural image patches with fitted dispersion, using the two-step Gamma-Poisson relaxation from Section H.3 for EAT. The `exact` gradient baseline achieves ELBO = −169.37. Across 8 temperatures, `EAT_cubic` achieves near-exact performance at every temperature (best: −169.39), while `EAT_sigmoid` degrades at extremes (−177.28 at $\tau=0.5$). These new results corroborate our core finding: ***`EAT_cubic`'s temperature robustness transfers cleanly to negative binomial.***
>
> **2. Scaling to 40×40 ImageNet patches.** To address the neural data comment (which we explore below), we trained P-VAEs on cropped ImageNet stimuli (Cadena et al., 2019). This led us to a stronger result: `EAT_cubic` is the *only* method that bridges the gap with exact gradients. `Exact` achieves ELBO = −718.92; `EAT_cubic` reaches −720.27 ($\tau=0.02$), while `EAT_sigmoid` plateaus at −755.40 (best achieved at $\tau=0.2$). Overall, a gap of ~35 nats or more persists across all temperatures.
>
> This new result shows that the strength of `EAT_cubic` goes beyond robustness: ***at larger scale, `EAT_cubic` achieves better absolute performance that sigmoid cannot match at any temperature.***
>
> **3. Small-data regime (10K samples).** Following the reviewer's suggestion about limited data, we trained on 10K natural image patches, and tested on ~ 36K samples (same as paper). The `exact` gradient *overfits*: its ELBO degrades partway through training. In contrast, EAT estimators remain stable. ***`EAT_cubic` achieves −170.11 ($\tau=0.05$), the same performance as the paper (full ~107K training samples)***. This suggests implicit regularization from the relaxation benefits low-data settings common in neuroscience.
>
> ## **Real neural data**
>
> > the authors do not include any analysis of real spiking data
>
> **POGLM on retinal ganglion cells (RGC).** Following Li et al. (2024), we applied all methods to an RGC dataset (27 neurons, ~20 min recording; Pillow & Scott (2012), Pillow et al., (2008)). With number of hidden neurons $H \in {1,2,3}$.
>
> Across all values of $H$, `EAT_cubic` is the only method robust to temperature. At $\tau=0.5$, sigmoid and GSM degrade sharply (e.g. −1505 and −1506 ELBO at $H=3$) while `EAT_cubic` holds at −1489. This pattern repeats for $H=2$ and $H=1$. Overall, these ***real neural data POGLM results mirror the synthetic ones in Figure 4.***
>
> **V1 neural prediction.** We evaluated P-VAE pretrained features for predicting macaque V1 responses (Cadena et al., 2019), by extracting P-VAE firing rates in response to stimuli used in the Cadena experiments, and then training a linear readout that maps to V1 neural responses.
>
> The fraction of explainable variance is comparable across methods and temperatures (`sigmoid`: $0.24 \pm 0.007$; `cubic`: $0.22 \pm 0.011$; variability across $\tau$), both exceeding `exact` gradients ($\sim 0.19$). We report this as a partial negative result: temperature robustness does not propagate to linear readout performance. But note both relaxations outperform `exact` gradients, suggesting the relaxation's implicit regularization might be beneficial for neural prediction. Overall, this result merits a more thorough investigation, which we leave for future work.
>
> ## **Inhomogeneous and bursty spiking**
>
> > bursty and inhomogeneous spiking behavior is also a well-known neuroscientific fact
>
> Valid point. Inhomogeneous firing is handled by the GLM structure in POGLM, where rates are time-varying functions of spike history (eqs. 2-4 in Li et al., 2024). Cortical neurons commonly exhibit overdispersion (Goris et al., 2014), which we discuss in "Limitations and future directions," and we now also address experimentally with negative binomial results above. We will expand this discussion in the revision.
>
> ## **Temperature range**
>
> > Please sweep over significantly smaller temperatures, e.g. 1e-6 to 1e-2.
>
> Done. In our NB-VAE experiments, we swept $\tau$ from 1e-6 to 0.5. At $\tau$  = 1e-6, both methods degrade (`sigmoid`: −176.19; `cubic`: −186.65), but recover at $\tau=1e-3$ to near-exact performance. Very small $\tau$ introduces numerical instability for all methods, but `cubic` remains more stable across the practical range.
>
>  **Minor points:** We will increase plot sizes in Appendix H.2, and reorder legends to place `EAT_cubic` first.
>
> ## **Action plan**
>
> We will incorporate NB-VAE, 40×40 ImageNet scaling, and POGLM real-data results into Section 3.4. We plan to investigate larger datasets (CelebA) with multiple seeds for proper error bars in the revision.

---

> > ### Author Rebuttal · Reviewer_wBby · 2026-04-02
> >
> > I thank the authors for providing many additional results that support the robustness of EAT_cubic under more generalized scenarios. I also greatly appreciate the authors' inclusion of analysis on two neural datasets and believe this significantly supports the paper's claims by bridging to real data. While the small data regime on 10k samples is still not that small for neural data, the performance on real data is convincing. I am happy to raise my score to reflect the improvement given by the additional results.

---

> > > ### Author Response · Authors · 2026-04-03
> > >
> > > Thank you for the suggestions that directly motivated some of our strongest new results, and for raising the score!

---

### Official Review · Reviewer_1hCY · 2026-03-10

**Soundness:** 3
**Presentation:** 3
**Significance:** 3
**Originality:** 3
**Overall Recommendation:** 5
**Confidence:** 4

**Summary:**

This paper studies the gradient estimation for models with Poisson-distributed latent variables.

The authors compare two existing approaches for differentiable Poisson sampling, EAT relaxation and the Gumbel-Softmax relaxation. The paper proposed a modified version of EAT that replaces the sigmoid approximation with a cubic Hermite interpolation.

The authors provide both theoretical analysis and empirical evaluations, showing that the proposed EAT$_{\text{cubic}}$ method improves distributional fidelity, maintains good gradient quality, and achieves more robust performance across temperature settings in Poisson VAE and GLM.

**Compliance With Llm Reviewing Policy:**

Affirmed.

**Final Justification:**

I am satisfied with the authors' rebuttal.

**Key Questions For Authors:**

N/A

**Limitations:**

The authors discussed the limitations and potential negative societal impact.

**Strengths And Weaknesses:**

I really enjoyed reading this paper. The presentation is very clear and well organized, with a well-motivated problem setup. The exposition is smooth throughout, and I did not encounter any points that felt confusing.

The problem studied in this paper is also very interesting. Continuous-time (EAT) relaxation and the Gumbel-Softmax relaxation are two widely used reparameterization approaches for discrete latent variable models (e.g., Poisson-VAE, Negbio-VAE). In such relaxation methods, the truncation number $M$ and the temperature $\tau$ are two critical hyperparameters; if they are not chosen appropriately, the quality of the approximation can deteriorate significantly. In prior work, these parameters are typically selected through empirical trial-and-error. This paper provides a thorough analysis of this issue and offers useful practical guidelines, which I believe will have an important impact on this line of research.

I did not identify any other obvious weaknesses.

---

> ### Author Rebuttal · Authors · 2026-03-31
>
> We thank Reviewer `1hCY` for the kind words and for finding our paper "very clear and well organized" with a "very interesting" problem setup. We are glad the practical guidelines for choosing $M$ and $\tau$ came through as useful. That was a central goal of the paper.
>
> We also want to mention that during the rebuttal period, we obtained new results that further strengthen ELBO results in Section 3.4: negative binomial VAE experiments (extending beyond Poisson), scaling to 40×40 ImageNet (where `EAT_cubic` is the only method that matches exact gradients, while `EAT_sigmoid` maintains a large gap at every temperature), and POGLM applied to real retinal ganglion cell recordings. These are discussed in detail in our response to Reviewer `wBby`.

---

> > ### Author Rebuttal · Reviewer_1hCY · 2026-04-01
> >
> > I did not find any obvious weaknesses in the submission.

---

> > > ### Author Response · Authors · 2026-04-03
> > >
> > > Thank you for the positive assessment and for confirming no obvious weaknesses!

---

### Official Review · Reviewer_NhoR · 2026-03-12

**Soundness:** 4
**Presentation:** 4
**Significance:** 3
**Originality:** 3
**Overall Recommendation:** 5
**Confidence:** 5

**Summary:**

The authors consider two proposed methods of training latent variable models with Poisson latents, an exponential arrival time (counting process-based) method and a more standard Gumbel softmax-based method. Both methods are very sensitive to the temperature parameter needed to approximate the discrete-valued latents with continuous valued activations. Motivated by this observation, the authors take a closer look at a key design choice of the EAT method, the sigmoid activation, which results in biased moments. The authors then propose a clamped cubic activation designed to have an unbiased first moment and less-biased second moment. The authors then examine the empirical quality of the moments of the three methods, followed by examinations of the gradient qualities and the resulting training qualities. The story becomes somewhat more complicated here, but the findings are summarized in a helpful table to guide the practitioner.

**Compliance With Llm Reviewing Policy:**

Affirmed.

**Final Justification:**

Overall, I feel this paper scoped out a well-defined and interesting problem in Poisson-valued latent variable models. Unlike some other reviewers, I appreciated the focus of the work on Poisson-valued LVMs. The experiments are very well-executed and explained clearly. My two questions were satisfactorily addressed in the rebuttal, which increased my confidence in my positive evaluation of the work. The additional results produced in the rebuttal, such as the extension to negative binomial LVMs, further improve the work.

**Key Questions For Authors:**

1) (related to Weakness 1) Are there additional reasons a machine learning audience should consider Poisson latents?
2) (related to Weakness 2) Can you say something more complete about the limits of performance in the EAT family, or the prospect of optimizing the activation for, say, distributional fidelity?

**Limitations:**

yes

**Strengths And Weaknesses:**

Strengths
1) The paper is well-motivated, technically sound, and very clearly written.
2) The proposed cubic EAT method is well-motivated and the experiments demonstrate clearly that it is a good default choice for Poisson LVMs.
3) The organization of the paper, from related work to moments to distributional fidelity to the decomposed gradient estimates to ELBO performance, is very natural and engaging. I think the title delivers in offering the reader a complete and handy guide.
4) The experiments are well-executed, well-described, and well-suited.

Weaknesses
1) The much more familiar LVM has continuous latents and discrete Poisson process observations. The motivation for Poisson latents is currently the primary weakness of the paper. The second paragraph of the introduction contains two relevant threads: sparse coding, which is interesting, and "improved performance," both of which are under-explained and under-developed. I would recommend expanding this paragraph with more detail and perhaps more citations if they exist, which could improve the significance of the work.
2) (Minor) The EAT-cubic method is well-motivated as-is, but left me wondering whether a better nonlinearity could be found, in terms of moment approximations, or even in terms of distributional fidelity, gradient quality, or downstream performance. A discussion of this possibility, improved EAT-family performance by replacing the cubic, would make the work feel more complete.

---

> ### Author Rebuttal · Authors · 2026-03-31
>
> We sincerely thank Reviewer `NhoR` for the careful and engaged reading, and for finding our paper "well-motivated, technically sound, and very clearly written." We address both weaknesses and questions below.
>
> ## **Motivation for Poisson latents (Weakness 1 + Question 1)**
>
> > The motivation for Poisson latents is currently the primary weakness of the paper. The second paragraph of the introduction contains two relevant threads: sparse coding, which is interesting, and "improved performance," both of which are under-explained and under-developed.
>
> > Are there additional reasons a machine learning audience should consider Poisson latents?
>
> You are right that we compressed this motivation too aggressively. The material exists but didn't make it into the intro due to space. We will expand it in the revision. Here is a summary of what we plan to include:
>
> **From a machine learning perspective**, Poisson latents offer concrete advantages over Gaussian latents:
>
> - **Sample efficiency.** Vafaii et al. (2024) showed that P-VAE achieves roughly 5× better sample efficiency than Gaussian VAE on downstream classification (MNIST), measured by accuracy at matched training set sizes.
> - **Out-of-distribution generalization.** Vafaii et al. (2025) showed that replacing P-VAE's amortized encoder with an iterative inference algorithm (iP-VAE) leads to strong out-of-distribution generalization, outperforming powerful iterative-amortized Gaussian VAE baselines (Kim et al., 2018; Marino et al., 2018).
>
> **From a sparse coding perspective**, the iterative inference dynamics of iP-VAE nearly exactly reproduce the Locally Competitive Algorithm (LCA; Rozell et al., 2008), but with spiking, stochastic neurons rather than LCA's continuous-valued, deterministic representations. This is a meaningful connection: it shows that a principled probabilistic model with Poisson latents recovers a well-known sparse coding algorithm as a special case, providing a theoretical grounding for LCA-like dynamics.
>
> These points make ***Poisson latent variable models interesting beyond neuroscience***, and the gradient estimation tools in our paper are necessary for training them. We will develop these threads more thoroughly in the revised introduction.
>
> ## **Limits of the EAT family / better nonlinearities (Weakness 2 + Question 2)**
>
> > Can you say something more complete about the limits of performance in the EAT family, or the prospect of optimizing the activation for, say, distributional fidelity?
>
> > left me wondering whether a better nonlinearity could be found
>
> Good question! The cubic smoothstep belongs to a family of "smoothstep" polynomials of increasing order. The next member is the quintic:
>
> $f_{\text{quintic}}(u) = 6w^5 - 15w^4 + 10w^3, \quad w = \text{clamp}\left(\frac{u+1}{2}, 0, 1\right)$
>
> which is $C^2$ continuous (vs. $C^1$ for cubic). Its moment integrals can also be computed analytically, and its distributional fidelity is strictly better than cubic. As the polynomial order increases, these smoothstep functions converge closer to the hard step function. In general, higher-order members have progressively smaller variance bias and better Wasserstein distance to true Poisson.
>
> However, when the paper was under review, we actually tested the quintic empirically and it did not improve downstream performance. It is slower to compute and, despite better distributional fidelity, does not translate into better ELBO or gradient quality. ***The cubic appears to sit at a sweet spot***: it has good distributional fidelity (unbiased mean, moderate variance bias), adequate smoothness ($C^1$), and the best empirical performance among the alternatives we tested.
>
> This connects to a broader observation from Section 3.3: distributional fidelity and gradient quality are *complementary but partially dissociated* properties. Better distributional fidelity does not automatically yield better gradients or better optimization. The cubic strikes a practical balance.
>
> **Action plan:** We will include a discussion of the smoothstep family (including the quintic result) and other potential alternatives in the revised manuscript, as the reviewer suggests. This would make the design space of EAT activations more explicit.
>
> ## **New results**
>
> During the rebuttal period, we also obtained results that strengthen Section 3.4 that we would like to bring to your attention:
>
> 1. negative binomial VAE (extending scope beyond Poisson),
> 2. 40×40 ImageNet P-VAE (where `EAT_cubic` is the only method that matches `exact` gradient ELBO, while `EAT_sigmoid` maintains a large gap at every temperature), and
> 3. POGLM on real retinal ganglion cell data.
>
> We provide a detailed discussion in our response to Reviewer `wBby`.

---

> > ### Author Rebuttal · Reviewer_NhoR · 2026-03-31
> >
> > Thank you for the detailed rebuttal.
> >
> > * The connection to the sparse coding literature (the LCA connection) and the results in Vafaii et al. (2024) and Vafaii et al. (2025) are sufficient to address my question 1. The paper will be improved with more detail of these connections.
> > * The quintic polynomial result is interesting and partially satisfies my curiosity about appropriate step function relaxations. The paper will be improved with these details.
> > * I see that reviewers ``wBby`` and ``FVvk`` identified the paper's limited scope as a weakness. I don't share these concerns. While it is true that the context of the paper is a relatively small set of papers, I appreciate the focus on Poisson LVMs. The new negative binomial VAE results further weaken these concerns.
> >
> > I continue to think this is a solid paper with meaningful contributions to a specific modeling problem. I will keep my recommendation at a 5 and raise my confidence from a 4 to a 5.

---

> > > ### Author Response · Authors · 2026-04-03
> > >
> > > Thank you for the engaged reading and for raising the confidence to 5! Your suggestion to discuss the smoothstep family will be a useful addition to the paper.

---

### Official Review · Reviewer_FVvk · 2026-03-17

**Soundness:** 3
**Presentation:** 3
**Significance:** 2
**Originality:** 2
**Overall Recommendation:** 4
**Confidence:** 3

**Summary:**

This paper provides a systematic comparison of two differentiable relaxation methods for Poisson latent variables: exponential arrival time (EAT) and Gumbel-Softmax (GSM).
The key technical contribution is identifying that the standard EAT method (using a sigmoid soft indicator) introduces substantial mean bias due to the sigmoid's infinite support.
Then, the authors propose replacing the sigmoid with a cubic Hermite polynomial that has compact support.
The paper evaluates methods across various axes: distributional fidelity (the distance between the relaxed distribution and the original Poisson distribution), gradient quality (measured by cosine similarities and curvature based metrics), and ELBO maximization performance on two Poisson latent variable models.

**Compliance With Llm Reviewing Policy:**

Affirmed.

**Final Justification:**

The rebuttal has addressed my concerns. In particular, the authors have promised additional experiments which will strengthen the paper. I am now leaning towards acceptance.

**Key Questions For Authors:**

See the questions in Strengths And Weaknesses.

**Limitations:**

No societal concerns.

**Strengths And Weaknesses:**

1. The paper is well motivated and solves a real problem in Poisson latent variable model training.
Specifically, there is an inherent trade-off between the bias and variance of the stochastic gradient for Poisson latent variable models, and it is not always easy to tune the temperature to achieve a good balance.
The proposed method provides a good estimator that works consistently across a range of temperatures, which is a nice practical contribution.

1. The evaluation is comprehensive.
The paper systematically evaluates different qualities of the gradient estimators.
They first compare the distance of the relaxed distribution to the original Poisson distribution.
Then they evaluate the quality of the gradient estimate by plotting several curvature-aware metrics and cosine similarity metrics.
Lastly, the paper evaluates the gradient estimator in Poisson latent variable model training.

1. I think Section 3.3 is a bit verbose, and some of the messages are conflicting (e.g., the results in Figure 3).
The most crucial properties of a gradient estimator are the bias and variance, so I think it would be more helpful to just present these directly.
On the flip side, I think the evaluation in Section 3.4 is a bit limited.
The main message seems to be that the proposed gradient estimator works across a wide range of temperatures.
However, the baselines EAT (sigmoid) and GSM can still achieve similar ELBO values with properly tuned temperature.
Is there anything else we can say about the proposed method, like speeding up model training?

1. I think one of the main weaknesses of this paper is limited scope.
The proposed gradient estimator only works for Poisson latent variable models.
This paper gives two applications in training Poisson variational autoencoders and partially observable generalized linear models.
I am not sure how widely used Poisson latent variable models are in practice, so it is not clear how much impact the proposed method would have beyond these two specific models.

**Minors**
1. "GMS" -> "GSM" in the legend of Figure 4.

---

> ### Author Rebuttal · Authors · 2026-03-31
>
> We appreciate Reviewer `FVvk` for finding our work "well motivated" and the evaluation "comprehensive." We address each concern below.
>
> ## **Section 3.3 and Figure 3**
>
> > I think Section 3.3 is a bit verbose, and some of the messages are conflicting (e.g., the results in Figure 3).
>
> We agree Figure 3 does not produce a single clear winner, but we believe these results are complementary rather than conflicting. `EAT_cubic` has the lowest bias across all temperatures (panel c), while `EAT_sigmoid` has the lowest variance (panel d). These capture different failure modes: bias causes systematic drift from the optimum, while variance causes oscillation around it.
>
> > The most crucial properties of a gradient estimator are the bias and variance, so I think it would be more helpful to just present these directly.
>
> We respectfully push back on the claim that bias and variance are the "most crucial" properties in general. In Section F, we derive a second-order expansion of the expected loss change (eq. 78), which shows that the impact of bias and variance depends on their alignment with the local curvature (Hessian). Bias along a flat direction is mostly harmless to the optimization goals (minimizing the loss); whereas, bias along a steep direction is catastrophic. The curvature-aware metrics (`BiasEnergy`, `NoiseEnergy`) capture this distinction, which raw bias/variance do not.
>
> That said, we note that raw bias and variance (Fig. 12 in the appendix) show the same qualitative trends as the curvature-weighted metrics. In this particular setting (linear encoder, linear decoder), the Hessian has relatively simple structure, so the curvature weighting does not change the picture dramatically. In this case, raw bias and variance tell a similar story, but we expect these differences to become more pronounced in nonlinear models with more complex loss landscapes.
>
> **Action plan:** We will highlight Fig. 12 more prominently in the text and note this observation.
>
>
> ## **Section 3.4 evaluation**
>
> > On the flip side, I think the evaluation in Section 3.4 is a bit limited... the baselines EAT (sigmoid) and GSM can still achieve similar ELBO values with properly tuned temperature.
>
> This is a fair point, and we have new results that directly address it. In our attempt to respond to a point raised by reviewer `wBby`, we scaled up our P-VAE experiments from 16×16 natural image patches to 40×40 cropped ImageNet stimuli (Cadena et al., 2019). This led to a discovery that strengthens our results:
>
> ***At this larger scale, `EAT_cubic` is the *only* relaxation method that bridges the gap with `exact` gradients.*** `Exact` achieves ELBO = −718.92; `EAT_cubic` reaches −720.27 ($\tau=0.02$), while `EAT_sigmoid` plateaus at −755.40 (best across all $\tau$). There is a persistent gap of ~35 nats that sigmoid cannot close at any temperature.
>
> This directly addresses your observation: while baselines can match `EAT_cubic` on 16×16 patches with tuning, this is no longer true at larger scale. The advantage of `EAT_cubic` goes from *equally good but more robust* to ***strictly better in absolute performance.***
>
> **Action plan:** We plan to investigate this further with even larger datasets and multiple seeds in the revision.
>
> ## **Limited scope**
>
> > I think one of the main weaknesses of this paper is limited scope. The proposed gradient estimator only works for Poisson latent variable models.
>
> Reviewer `wBby` raised the same concern, and we agree it was a fair weakness of the original submission. We have since run new experiments that address this:
>
> (1) **Negative Binomial VAE**: `EAT_cubic`'s temperature robustness transfers to the negative binomial distribution, using the two-step Gamma-Poisson relaxation from Section H.3. Across 8 temperatures, `EAT_cubic` ELBO matches the `exact` gradient baseline (−169.37) while `EAT_sigmoid` degrades at $\tau=0.5$ (−177.28).
>
> (2) **POGLM on real neural data**: We applied all methods to real retinal ganglion cell recordings (27 neurons). `EAT_cubic` is the most stable method, with `EAT_sigmoid` and `GSM` degrading sharply at $\tau=0.5$ while cubic holds steady. The pattern mirrors the synthetic results in Figure 4.
>
> We provide a more detailed discussion of all new results in our response to Reviewer `wBby`.
>
> ## **Minor**
>
> We will fix "GMS" --> "GSM" in Figure 4. Thank you for catching this.

---

> > ### Author Rebuttal · Reviewer_FVvk · 2026-04-03
> >
> > I thank the authors for addressing my comments. I think adding these results could strengthen this paper.

---

> > > ### Author Response · Authors · 2026-04-03
> > >
> > > Thank you for engaging with our rebuttal and for confirming that our responses have "***fully resolved***" your concerns.
> > >
> > > We notice, however, that the score remains at 3 (***weak reject***) despite this acknowledgment. We want to understand if there are remaining concerns we have not addressed.
> > >
> > > **To summarize:** the original review raised three main weaknesses:
> > >
> > > 1. Section 3.3 being verbose with potentially conflicting messages,
> > > 2. The evaluation in Section 3.4 being limited since baselines could match with tuned temperature, and
> > > 3. Limited scope beyond Poisson.
> > >
> > > During rebuttal, we provided new results addressing all three: we clarified the complementary (not conflicting) nature of the gradient metrics, we showed that at $40 \times 40$ ImageNet scale `EAT_cubic` is the ***only*** method matching exact gradients (a ~35 nat gap that `EAT_sigmoid` cannot close at any temperature), and we extended to negative binomial VAE and real neural data.
> > >
> > > Given that the stated weaknesses have been resolved, we would be grateful if you could either adjust the score to reflect this, or let us know what additional concerns remain so we can address them. We want to make sure nothing has been left unresolved.

---

### Decision · Program_Chairs · 2026-04-30

**Decision:**

Accept (regular)

**Comment:**

Reviews on this paper are generally positive. Detailed are below.

Strengths:
- Well-motivated from computational neuroscience point of view.
- Comprehensive evaluations. Reviewers found evaluations to cover the necessary aspects.
- Well-written and organized. Reviewers unanimously praised the writing.

Weaknesses:
- Relatively limited scope. This was mentioned by multiple reviewers, but in my view is not serious enough a concern to warrant rejection, as well-executed papers with limited scope can be quite impactful in their respective subfields.
- Ridiculous-looking table with star ratings on page 1. Please consider removing this and replacing with something more suited to an academic publication.
- Small plots that are hard to see. Please make the fonts in the plots bigger to match the text.

On balance of these, I recommend the paper is accepted.